# Personalized whole-brain neural mass models reveal combined Aβ and tau hyperexcitable influences in Alzheimer's disease
Lazaro M. Sanchez-Rodriguez[1,2,3], Gleb Bezgin [1,2,3,4], Felix Carbonell[5], Joseph Therriault[1,2,4], Jaime Fernandez-Arias[1,2,4], Stijn Servaes[1,2,4], Nesrine Rahmouni[1,2,4], Cécile Tissot[1,2,4,6], Jenna Stevenson[1,2,4], Thomas K. Karikari [7,8], Nicholas J. Ashton[7,9,10,11], Andréa L. Benedet[7], Henrik Zetterberg [7,12,13,14,15,16], Kaj Blennow [7,16], Gallen Triana-Baltzer[17], Hartmuth C. Kolb[17], Pedro Rosa-Neto [1,4] & Yasser Iturria-Medina [1,2,3] ✉

Neuronal dysfunction and cognitive deterioration in Alzheimer's disease (AD) are likely caused by multiple pathophysiological factors. However, mechanistic evidence in humans remains scarce, requiring improved non-invasive techniques and integrative models. We introduce personalized AD computational models built on whole-brain Wilson-Cowan oscillators and incorporating resting-state functional MRI, amyloid-β (Aβ) and tau-PET from 132 individuals in the AD spectrum to evaluate the direct impact of toxic protein deposition on neuronal activity. This subject-specific approach uncovers key patho-mechanistic interactions, including synergistic Aβ and tau effects on cognitive impairment and neuronal excitability increases with disease progression. The data-derived neuronal excitability values strongly predict clinically relevant AD plasma biomarker concentrations (p-tau217, p-tau231, p-tau181, GFAP) and grey matter atrophy obtained through voxel-based morphometry. Furthermore, reconstructed EEG proxy quantities show the hallmark AD electrophysiological alterations (theta band activity enhancement and alpha reductions) which occur with Aβ-positivity and after limbic tau involvement. Microglial activation influences on neuronal activity are less definitive, potentially due to neuroimaging limitations in mapping neuroprotective vs detrimental activation phenotypes. Mechanistic brain activity models can further clarify intricate neurodegenerative processes and accelerate preventive/treatment interventions.

Alzheimer's disease (AD) is defined by synaptic and neuronal degeneration and loss accompanied by amyloid beta (Aβ) plaques and tau neurofibrillary tangles (NFTs)[1–3]. In vivo animal experiments indicate that both Aβ and tau pathologies synergistically interact to impair neuronal circuits[4]. For example, the hypersynchronous epileptiform activity observed in over 60% of AD cases[5] may be generated by surrounding Aβ and/or tau deposition yielding neuronal network hyperactivity[5,6]. Cortical and hippocampal network hyperexcitability precedes memory impairment in AD models[7,8]. In an apparent feedback loop, endogenous neuronal activity, in turn, regulates Aβ aggregation, in both animal models and computational simulations[9,10]. Multiple other factors involved in AD pathogenesis -remarkably, neuroinflammatory dysregulations- also seemingly influence neuronal firing and act on hypo/hyperexcitation patterns[11–13]. Thus, mounting evidence suggest that neuronal excitability changes are a key mechanistic event appearing early in AD and a tentative therapeutic target to reverse disease symptoms[3,4,7,14]. However, the exact patterns of Aβ, tau and other disease factors' neuronal activity alterations in AD's neurodegenerative progression are unclear as in vivo and non-invasive measuring of neuronal excitability in human subjects remains impractical.

---

Brain imaging and electrophysiological monitoring constitute a reliable readout for brain network degeneration likely associating with AD's neuro-functional alterations[3,15–18]. Patients present distinct resting-state blood-oxygen-level-dependent (BOLD) signal content in the low frequency fluctuations range (0.01–0.08 Hz)[16,19]. These differences increase with disease progression, from cognitively unimpaired (CU) controls to mild cognitive impairment (MCI) to AD, correlating with performance on cognitive tests[16]. Another characteristic functional change is the slowing of the electro-(magneto-) encephalogram (E/MEG), with the signal shifting towards low frequency bands[15,18]. Electrophysiological spectral changes associate with brain atrophy and with losing connections to hub regions including the hippocampus, occipital and posterior areas of the default mode network[20]. All these damages are known to occur in parallel with cognitive impairment[20]. Disease processes also manifest differently given subject-specific genetic and environmental conditions[1,21]. Models of multiple pathological markers and physiology represent a promising avenue for revealing the connection between individual AD fingerprints and cognitive deficits[3,18,22].

In effect, large-scale neuronal dynamical models of brain re-organization have been used to test disease-specific hypotheses by focusing on the corresponding causal mechanisms[23–25]. By considering brain topology (the structural connectome[18]) and regional profiles of a pathological agent[24], it is possible to recreate how a disorder develops, providing supportive or conflicting evidence on the validity of a hypothesis[23]. Generative models follow average activity in relatively large groups of excitatory and inhibitory neurons (neural masses), with large-scale interactions generating E/MEG signals and/or functional MRI observations[26]. Through neural mass modeling, personalized virtual brains were built to describe Aβ pathology effects on AD-related EEG slowing[25] and several hypotheses for neuronal hyperactivation have been tested[27]. Simulated resting-state functional MRI across the AD spectrum was used to estimate biophysical parameters associated with cognitive deterioration[28]. In addition, different intervention strategies to counter neuronal hyperactivity in AD have been tested[10,22]. Notably, comprehensive computational approaches combining pathophysiological patterns and functional network alterations allow the quantification of non-observable biological parameters[29] like neuronal excitability values in a subject-specific basis[1,3,18,23,24], facilitating the design of personalized treatments targeting the root cause(s) of functional alterations in AD.

Here, we develop a personalized whole-brain neural mass model integrating multilevel, multifactorial AD pathophysiological profiles to clarify their causal impact on neuronal activity alterations. Using individual in vivo functional MRI together with Aβ- and tau- positron emission tomography (PET), we infer and quantify the combined influence of these proteinopathies on human neuronal excitability. Additionally, we investigate the associations between the obtained subject-specific pathophysiological neuronal activity affectations and clinically applicable blood-plasma biomarkers (p-tau217, p-tau231, p-tau181, glial fibrillary acidic protein), gray matter atrophy, as well as cognitive integrity measured in the same patient cohort. Finally, we identify the critical toxic protein accumulation stages that typically accompany hallmark AD electrophysiological (E/MEG) alterations. Overall, our results expand previous understandings of neuro-pathological impact on AD, namely the emergence of neuronal hyperactivity[3,4,7,14], slowing of the E/MEG signals[15,18] and the existence of synergistic multifactorial interactions[1,4]. These findings support the premise of using integrative neural mass models to decode multilevel mechanisms in complex neurological disorders.

## Results

### Inferring pathophysiological impacts on whole-brain neuronal activity

Figure 1 presents the proposed personalized generative framework to study the combined pathophysiological effect of Aβ and tau on neuronal activity (see *Methods*). Cognitively unimpaired, mild cognitive impairment and Alzheimer's disease participants (N = 132, Supplementary Table 1) underwent structural and resting-state functional MRI and Aβ ([18]F-NAV4694)-, tau ([18]F-MK-6240)- and microglial activation ([11]C-PBR28)-PET. Individuals were also cognitively profiled and had measures of plasma p-tau and glial fibrillary acidic protein (GFAP). From the fMRI signals, regional fractional amplitudes of low-frequency fluctuations (fALFF) values were

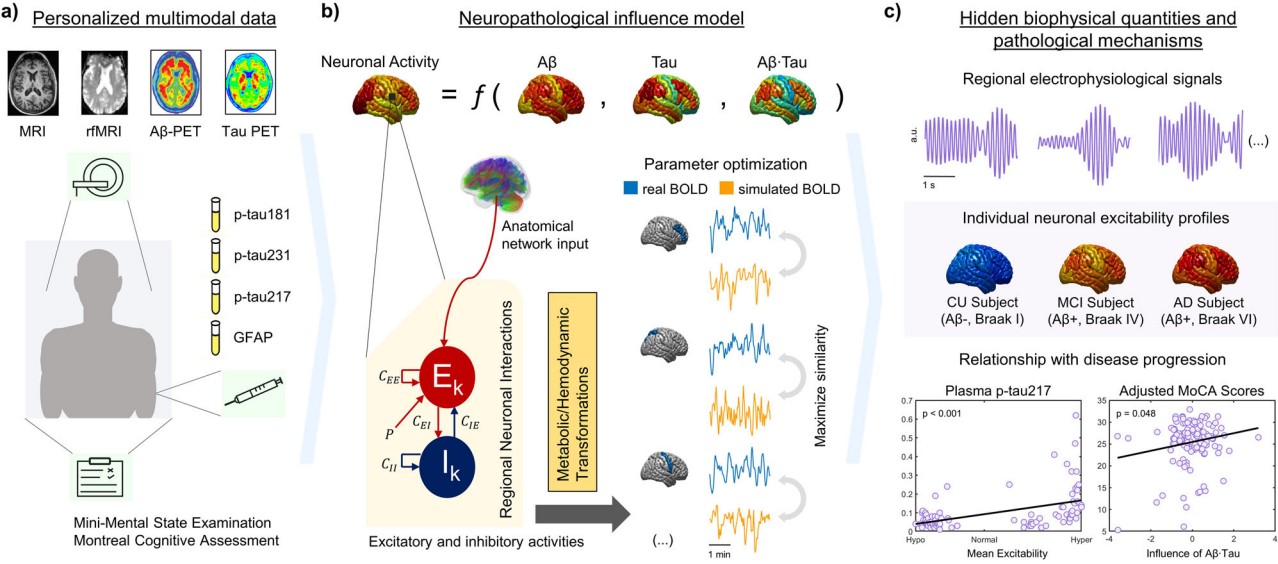

**Fig. 1 | Schematic AD personalized pathophysiological whole-brain models.**
**a** Individuals underwent a multimodal assessment including structural and resting-state functional MRI, Aβ and tau-PET, clinically relevant plasma biomarkers, and cognitive evaluations. **b** In the Alzheimer's disease model, the subject's neuronal excitability profile is defined as a function of Aβ, tau and the synergistic interaction of Aβ and tau. Regional excitatory and inhibitory firing rates are influenced by the local pathophysiological profiles and the signals coming from other regions via an average anatomical connectome. The regional neuronal signals generate BOLD indicators

through metabolic/hemodynamic transformations. By maximizing the similarity between the generated and observed BOLD data, the set of subject-specific influences of the pathophysiological Aβ, tau and Aβ·tau factors on neuronal activity are quantified. **c** These estimated pathophysiological influences serve to recover electrophysiological activity producing the real individual BOLD signals, and to study individual excitability profiles and their relationship with independent AD (plasma) markers and cognitive deterioration. Volumetric brain views in the figure were generated with SurfStat (https://www.math.mcgill.ca/keith/surfstat/).

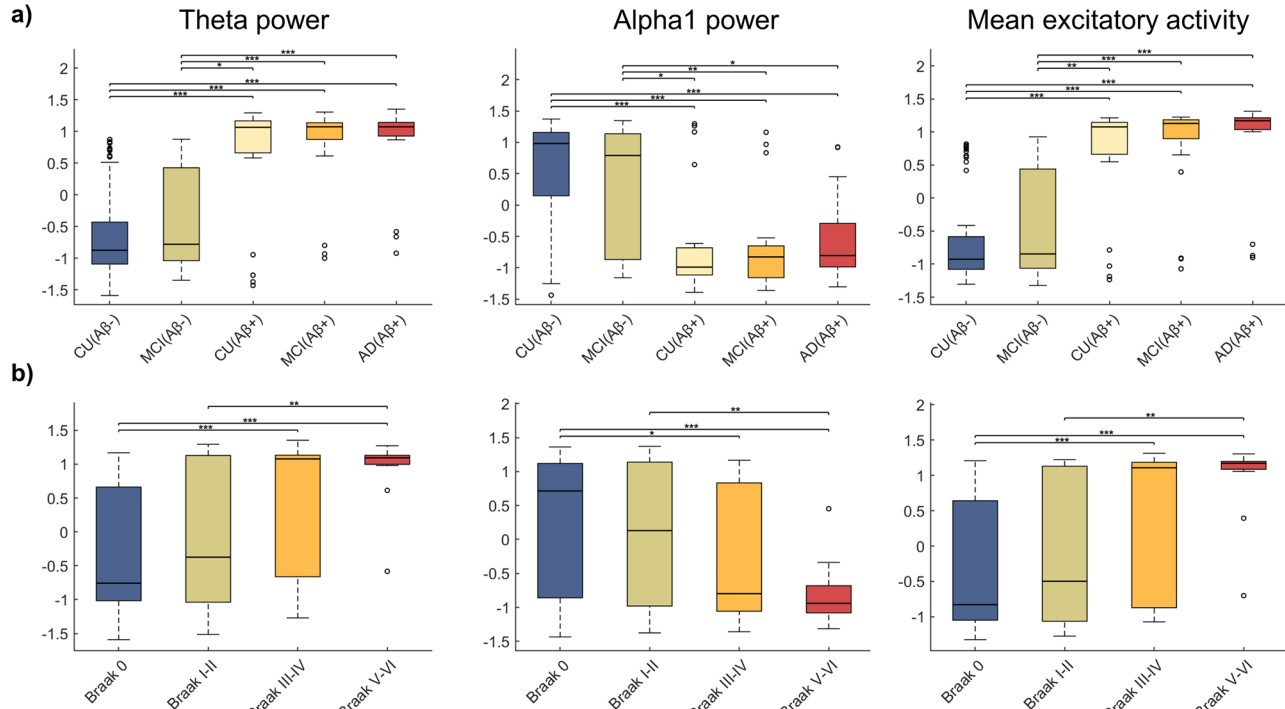

**Fig. 2 | Behavior of the inferred electrophysiological quantities of interest with Aβ and tau deposition levels.** From left to right: ratio of power in the theta band (4–8 Hz) of the regional excitatory input currents (the E/MEG is proportional to the excitatory input current), ratio of power in the alpha1 band (8–10 Hz) and mean excitatory firings (over all regions and time points). Each of the quantities was standardized using the mean and standard deviation from all subjects, for visualizing general trends. Participants were then grouped according to clinical diagnosis and Aβ-positivity (**a**) and Braak stages (**b**). In the box-and-whisker plot, the central lines indicate the group medians, with the bottom and top edges of each box denoting the 25th and 75th percentiles, respectively. Whiskers extend to the maximum and minimum values while data points that are deemed outliers for the group are plotted individually with circles. The results of ANCOVA post-hoc t-tests for the above-mentioned groups, with the corresponding electrophysiological quantity as response variable and age and sex as covariates are also shown. * represents significance level $p < 0.05$, ** means significance level $p < 0.01$ and *** is $p < 0.001$.

obtained for 66 bilateral regions of interest (DKT parcellation[30]), a measure consistently identified as a reliable neuronal activity biomarker of AD's progression[1,16,19] and compatible with structurally-defined brain parcellations[31]. We assume that at each brain region, the excitability properties of the excitatory neuronal populations are potentially mediated by the local pathophysiological burden, i.e., the participant's PET-measured accumulation of Aβ plaques, tau tangles and the combined Aβ and tau deposition (their synergistic interaction, Aβ·tau)[3–7,22]. Neuronal activities generated in this manner via interconnected Wilson–Cowan oscillators[32–36] are transformed into BOLD signals by a hemodynamic-metabolic module. Lastly, the individual model parameters quantifying the brain-wide subject-specific influence of each neuropathological factor (or their synergistic interaction) on neuronal excitability are identified by retaining the set maximizing the similarity between the simulated BOLD data and the subject's real BOLD indicators.

In Supplementary Fig. 1, we report the obtained likely Aβ, tau and Aβ·tau relative contributions for all participants. The contributions of each factor to the estimated neuronal excitability are subject-dependent but different trends exist within the clinical groups. For example, 13 out 16 AD participants (81.3%) had an estimated Aβ effect favoring hyper-excitation –disregarding the magnitude–, 10 (62.5%) had hyperexcitable tau influences, and 9 (56.3%), Aβ·tau. In the CU group, the majority of subjects also had hyperexcitable Aβ effects (71.6%), while hypoexcitable tau and Aβ·tau contributions were predominant (59.3% and 54.3%, respectively). The average correlation between observed fALFF markers and the best-fit in silico analogous quantities was 0.44 across participants (standard deviation = 0.10). For all regions and subjects (8712 data points), real and in silico neuronal activity indicators generated through our pathological influence model follow a linear relationship (Supplementary Fig. 2).

Subsequently, we investigated the pathophysiological mechanisms that give rise to the observed impacts on neuronal activity. We performed statistical tests on several quantities of interest that were computed after individual parameter identification with the goal of better understanding Aβ and tau's combined neuronal activity effects across the AD spectrum. The participants were, for statistical analysis, separated into groups (Supplementary Table 2) according to their clinical diagnosis (CU, MCI, AD) and Aβ-positivity or in vivo Braak staging[37,38]. In the next subsections, we study reconstructed hidden electrophysiological signals, neuronal excitability spatial profiles, and additive relationships with plasma biomarkers and cognitive integrity.

**Reproducing hallmark electrophysiological alterations in AD progression**

A desired attribute of biologically-defined modeling tools in clinical applications is to reproduce and mechanistically clarify reported pathophysiological observations. Through the inferred pathophysiological influence parameters, we reconstructed proxy quantities for electro-(magneto)encephalographic (E/MEG) sources in each brain region and subject (E/MEGs were not recorded for participants in the TRIAD cohort). We tested whether the AD pathophysiological whole-brain estimations recreated reported spectral changes in AD, i.e., increases of theta band power (4–8 Hz) and decreases of power in the lower alpha band (alpha1, 8–10 Hz)[15,18,22], as the disease progresses. Among the quantities contributing to the E/MEG model output, we also closely studied excitatory firings and changes to their magnitude given the influence of the toxic protein depositions.

We observed that the standardized ratio of power in the theta band (4–8 Hz) was higher for Aβ+ groups than for Aβ- (Fig. 2a and Supplementary Table 3). Conversely, the alpha1 power (8–10 Hz) decreased with Aβ-positivity. Finally, the average excitatory firings were generally higher for

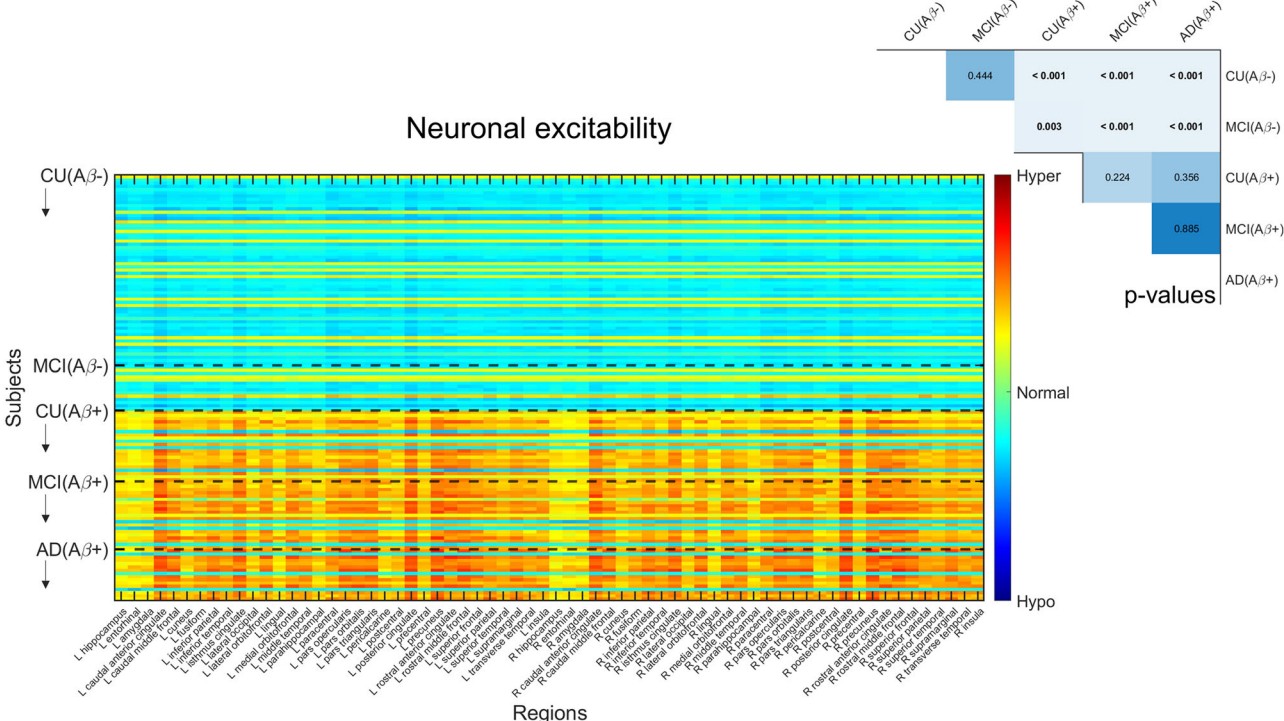

**Fig. 3 | Neuronal excitabilities under the influence of Aβ, tau and Aβ·tau.** Inferred neuronal excitability values for the brain regions of interest ("y"-axis) and all subjects ("x"-axis). Participants were grouped according to clinical diagnosis and Aβ-positivity in this figure, to understand Aβ's contribution to the individually estimated biological profiles (see Supplementary Fig. 3 for tau's effect). Within a group, subjects appear according to their existing ordering in the anonymized database. Warm colors represent hyperexcitability of the region in the subject's brain and cool colors denote hypoexcitable states (the color-bar extends to the extremes of the optimization interval). Results of ANCOVA post-hoc t-tests for the above-mentioned groups, with the average intra-brain excitability values as response variable and age and sex as covariates appear in the upper right. P-values in bold fonts represent differences at a 5% significance level or lower.

Aβ+ subjects. Similar results were observed across Braak stages (Fig. 2b). Differences in all, theta and alpha1 power and mean excitatory activity, were observed for subjects in Braak 0 (non-significant tau neurofibrillary tangle involvement) and the advanced limbic (Braak III-IV) and isocortical stages (Braak V-VI) and, furthermore, for Braak I-II (transentorhinal) and Braak V-VI subjects.

## The estimated neuronal excitabilities increase with clinical states and disease progression

Seeking to find mechanisms underlying the observed electrophysiological patterns, we studied the biophysical quantity that is influenced by the pathophysiological factors in our model: neuronal excitability. Figure 3 and Supplementary Fig. 3 show excitability values for all brain regions of interest and subjects. The combined action of the pathological factors either increases ("hyper") or decreases ("hypo") regional excitability around a certain baseline normal value.

We found significant differences of neuronal excitability due to Aβ positivity and Braak stages. Firstly, we observed significant discrimination between all Aβ− and Aβ+ groups (Fig. 3), i.e.: CU(Aβ−) and CU(Aβ+), MCI(Aβ+), AD(Aβ+) ($p < 0.001$, sex and age adjusted); MCI(Aβ−) and CU(Aβ+), MCI(Aβ+), AD(Aβ+) ($p < 0.05$, sex and age adjusted). Additionally, we discovered similar differences between Braak 0 participants and those in all later stages, and for Braak I-II, and Braak V-VI (Supplementary Fig. 3). Subjects in advanced disease stages generally presented hyperexcitability profiles, while most of the Aβ− and Braak 0, I-II participants were largely characterized by a slight hypoexcitability.

## Neuronal hyperexcitability relates to high levels of plasma AD biomarkers and gray matter atrophy

We proceeded to investigate the relationship between the obtained individual excitability values and biomarkers of AD-associated neurodegeneration. We utilized blood biomarkers of AD pathophysiology, which constitute accessible alternatives to neuroimaging indicators[39–42] and brain tissue atrophy assessed via voxel-based morphometry (VBM)[41,43]. The analyses sought to determine if the results of the computational estimations aligned with independent measurements for neurodegeneration that were not considered in the participants' whole-brain Aβ and tau effects models.

Figure 4a–d show the behavior of the average intra-brain excitabilities with the plasma biomarkers p-tau181, p-tau231 and p-tau217 (phosphorylated tau indicators) and GFAP (a measure of reactive astrogliosis and neuronal damage[42]), respectively. In Fig. 4e, we examine the relationship between average intra-brain excitabilities and VBM values, while Fig. 4f–h present results of the correlation analyses at specific regions (parahippocampal gyrus, fusiform gyrus and amygdala) with consistent gray matter alterations in AD based on VBM[44], across all the subjects. The other region-specific statistically significant results all appeared in areas with documented reductions in gray matter volume (Supplementary Table 3), including the hippocampus, entorhinal cortex and posterior cingulate gyrus, bilaterally[44]. Notably, we observed that high levels of the plasma biomarkers and reduced gray matter volume significantly relate to the participants' neuronal hyperactivation. Such subjects are typically Aβ- and tau-positive (Supplementary Fig. 4), underscoring that the model-obtained excitabilities reflect Aβ and tau pathology together with generalized neurodegeneration.

## Synergistic Aβ and tau interactions strongly relate to cognitive performance

To conclude the post-hoc investigation of the relevant quantities identified through the AD pathophysiological neural activity models, we proceeded to assess the pathophysiological factors' effects on cognitive impairment. For this purpose, we utilized the individual Aβ, tau and Aβ·tau weights contributing to the obtained neuronal excitability (Supplementary Fig. 1) as predictors in regression models with response variables MMSE and MoCA

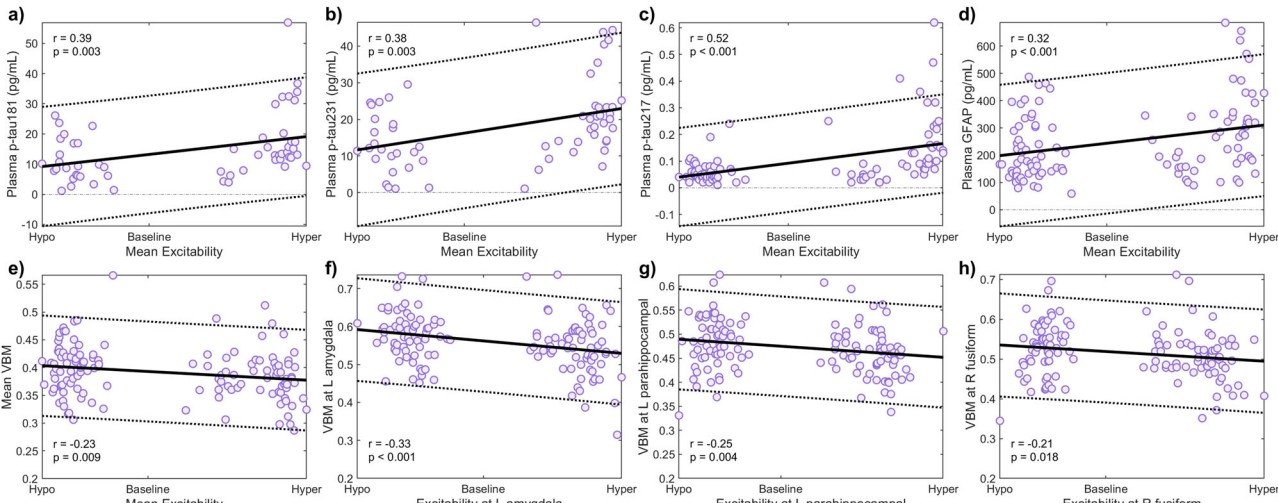

**Fig. 4 | Relationship between the inferred neuronal excitability values and independent AD biomarkers.** Spearman's correlation analyses for the associations between the participants' estimated average intra-brain excitabilities and the plasma biomarkers p-tau181 (**a**), p-tau231 (**b**), p-tau217 (**c**) and GFAP (**d**), and the average gray matter volume measured via voxel-based morphometry (VBM) (**e**). The relationships between local excitability values and the regional volumes are shown for the left amygdala (**f**), left parahippocampal gyrus (**g**), and right fusiform gyrus (**h**). The error bands denote 95% confidence intervals.

scores, respectively, while adjusting for age, sex and education. We observed (Table 1) that both Aβ's solo influence on neuronal activity and the estimated Aβ·tau synergistic interaction term were significant predictors of MMSE and MoCA evaluations ($p < 0.05$). The coefficients of these terms in the regression models were positive in all cases, namely, the lower the pathophysiological influence parameter is, the lower the cognitive score. Negative additive effects of the studied factors yield higher firing rates at a given input current in our pathophysiological influence model (*Methods, Electrophysiological model*). Thus, neuronal hyperexcitability seems to be associated with cognitive deterioration, according to our calculations.

## Discussion

We developed an integrative biophysical framework to map pathophysiological influences on neuronal activity, with application to AD. Highly data-driven models (not intended to replicate neuronal activity features) have been used in the past to individually characterize multifactorial dynamic interactions propagating through anatomical and vascular networks in the AD spectrum[1,17,45,46]. On the other hand, mechanistic investigations have assessed the emergence of pathological electrophysiological activity in generative models that consider the influence of isolated biological factors, such as Aβ plaques[25], tau tangles[47] or in several possible AD synaptic dysfunction scenarios[10,22]. Despite the high computational value of these works, realistic biological information could have been estimated from the data under certain constraints, to validate the mechanistic simulations. A recent work by Ranasinghe et al.[48] obtained parameters in computational models and correlated the results with Aβ and tau PET SUVRs. Instead of assessing associations, we intended to characterize the direct role that Aβ and tau have in the generation of pathological neuronal activity in AD. In neural mass models, causal effects are straightforwardly measured by perturbing its relevant biophysical parameters and observing the dynamical changes that occur to the neuronal signals generated under the perturbation[1,23,29,49]. The subject-specific influences by Aβ and tau (imaged through in vivo PET) on neuronal firing were computationally identified from fMRI indicators in this work. Altogether, we observed increased neuronal excitability with AD progression, which also predicted increased plasma biomarkers concentrations, accelerated gray matter atrophy and cognitive impairment.

Our findings confirm previous observations[3–7,15] and cast new light on pathological processes that are inaccessible to in vivo human neuroimaging, as relationships with neuronal excitability in AD. Through considering the influence of multiple pathophysiological factors, we have retrieved the AD

electrophysiological hallmark: enhancement of theta band activity together with alpha decreases, as disease progresses[15,18], from BOLD signals. Our results also indicate that CU(Aβ+) and/or Braak III-IV are the stages from which these electrophysiological biomarkers become abnormal. These groups contain subjects who are not cognitively impaired but present significant Aβ deposition[50] and/or have widespread temporal and parietal tau aggregation detectable by tau PET[38]. A recent study, also on subjects from the TRIAD cohort, found reduced, clinically significant delayed recall and recognition memory tests performance at Braak III and IV stages as well[51]. Additionally, multicenter research has shown that CU(Aβ+) subjects, independently of tau status, present substantially increased risk of short-term (3–5 years) conversion to mild cognitive impairment, compared to CU(Aβ−)[52]. Our personalized estimations of the pathophysiological impacts on neuronal activity reaffirm this evidence. Aβ+ and post-Braak II individuals may be, on average, the most likely candidates to benefit from early disease interventions modifying the cognitive decline that associates with patho-electrophysiological activity[15,18,20].

Although proxy measurements[53], post-mortem studies[14] and animal models[4] have suggested neuronal hyperactivity mechanisms in AD, no direct quantification of in vivo neuronal excitability existed thus far in humans. In this study, by assuming a toxic protein influence model (Aβ, tau, Aβ·tau) we inferred neuronal excitability values from the individual PET-functional MRI datasets. The progression towards hyperexcitation with disease worsening was equally evident for a simplified pathophysiological influence model with separate contributions by Aβ and tau only (Supplementary Figs. 5 and 6). Increased excitability was also associated with high levels of plasma biomarkers (blood phosphorylated tau and GFAP) which are sensitive to incipient AD pathology[40–42,54,55] and disease progression, especially p-tau217[39,40]. Additionally, the obtained excitability values also correlated with VBM measures –neuronal firing increases with decreased gray matter volume, particularly at brain regions that most prominently showcase neurodegeneration in AD[44]. Our correlation analyses of the relationship between estimated excitability values in the whole-brain models and these independent AD biomarkers may suggest that functional readjustments are attempted in parallel to Aβ and tau pathological spread and the loss of neurons in the human brain. Finally, we also observed that the more hyperactive the existing excitatory neuronal populations of a subject were (i.e., presenting negative influence values of the significant factors in our model), the greater the participant's cognitive dysfunction, thus supporting a direct link among neuronal excitability, pathophysiological burden, and cognitive integrity.

**Table 1 | Multiple linear regression analysis investigating the pathological effects on neuronal activity as predictors of MMSE and MoCA scores**

| MMSE scores | | | | |
|---|---|---|---|---|
| | β | 95% CI of β | *t*-Statistic | *p* |
| Intercept | 22.463 | [15.591 29.335] | 6.471 | <0.001 |
| $\theta^{A\beta}$ | 0.791 | [0.092 1.489] | 2.241 | 0.027 |
| $\theta^{Tau}$ | 0.138 | [−0.581 0.857] | 0.378 | 0.705 |
| $\theta^{A\beta \cdot Tau}$ | 1.040 | [0.371 1.710] | 3.077 | 0.002 |
| Sex | −1.505 | [−2.872 −0.139] | −2.180 | 0.031 |
| Age | 0.064 | [−0.025 0.152] | 1.426 | 0.156 |
| Education | 0.095 | [−0.096 0.286] | 0.987 | 0.326 |
| **MoCA scores** | | | | |
| | β | 95% CI of β | *t*-Statistic | *p* |
| Intercept | 18.702 | [8.656 28.748] | 3.686 | <0.001 |
| $\theta^{A\beta}$ | 1.598 | [0.618 2.579] | 3.229 | 0.002 |
| $\theta^{Tau}$ | −0.020 | [−0.973 0.932] | −0.042 | 0.967 |
| $\theta^{A\beta \cdot Tau}$ | 1.006 | [0.009 2.004] | 1.997 | 0.048 |
| Sex | −2.379 | [−4.298 −0.460] | −2.454 | 0.015 |
| Age | 0.096 | [−0.034 0.226] | 1.455 | 0.148 |
| Education | 0.046 | [−0.222 0.314] | 0.340 | 0.735 |

The influences of Aβ plaques ($\theta^{A\beta}$), tau tangles ($\theta^{Tau}$) and the interaction of Aβ and tau ($\theta^{A\beta \cdot Tau}$) on neuronal activity, sex, age and education were considered as predictors. Reported values are obtained coefficients (β), the 95% confidence intervals (CI) and the p-values for the t-statistic of the two-sided hypothesis tests. MMSE: dfe = 123; $R^2$ = 0.18, $p < 0.001$, normally distributed residuals (two-sided one sample Kolmogorov–Smirnov test, $p < 0.001$), $\theta^{A\beta}_{Cohen's-d}$ = 0.202, $\theta^{Tau}_{Cohen's-d}$ = 0.034, $\theta^{A\beta \cdot Tau}_{Cohen's-d}$ = 0.277; MoCA: dfe = 120; $R^2$ = 0.19, $p < 0.001$, normally distributed residuals (two-sided one sample Kolmogorov–Smirnov test, $p < 0.001$), $\theta^{A\beta}_{Cohen's-d}$ = 0.295, $\theta^{Tau}_{Cohen's-d}$ = −0.004, $\theta^{A\beta \cdot Tau}_{Cohen's-d}$ = 0.182. The pairwise correlation coefficients between the estimated neuronal activity influences were $r(\theta^{A\beta}, \theta^{Tau})$ = 0.35, $r(\theta^{A\beta}, \theta^{A\beta \cdot Tau})$ = 0.23 and $r(\theta^{Tau}, \theta^{A\beta \cdot Tau})$ = 0.19.
*MMSE* Mini-Mental State examination, *MoCA* Montreal Cognitive Assessment.

The above interpretation of the results in this article relies on the assumptions of the pathophysiological influence model (see Eq. (1)) and previous evidence of Aβ and tau's effects in the AD brain. In our individual dynamical models, the subject's Aβ and tau accumulations are the sole source of information and permitted influences on the generated neuronal activities. We confirmed that data fitting by adding the individual Aβ, tau and Aβ·tau brain maps (mean Akaike information criterion = −171.8 across subjects) outperformed the models without these variables (mean Akaike information criterion = −169.4). In addition, Supplementary Fig. 7 shows the improvement in each participant's model fit through the model with Aβ, tau and Aβ·tau neuronal excitability parameters compared to the simpler one with neuronal excitability not influenced by Aβ and tau. The subjects' F-statistics increase with Aβ- and tau-positivity and are significant for 81.8% of the subjects ($p < 0.05$), with the remaining subjects being predominantly Aβ− and/or Braak 0. This analysis further demonstrates the fundamental contribution of Aβ and tau loads to explain neuro-functional alterations observed in AD. Aβ and tau's toxic accumulations are believed to play key roles in the processes leading to neuronal degeneration and loss[1–3]. Their respective progression patterns (measured in vivo by PET uptake) are different, however, with Aβ plaques generalizing to many cortical areas early in the disease, while NFT spreading increases rapidly in temporal and parietal regions only[56]. Previous research also suggested that Aβ-induced hyperexcitability precluded tau accumulation[57,58]. AD participants, naturally, present higher levels than their counterparts who are Aβ− and tau-negative and may not be diagnosed as such[59]. Based on these facts, one may explain why Aβ's separate contribution and Aβ and tau's synergistic interaction –and not tau alone, with less whole-brain involvement than Aβ– were the most significant factors influencing aberrant neuronal activity and

cognitive symptomatology in our cohort (Table 1). Likewise, one would have expected a certain separation of neuronal excitability according to Aβ and tau statuses (Fig. 3, Supplementary Fig. 3). Existing computational models that assume parameter perturbations by relevant factors as serotonin receptor maps in neuropsychiatric disorders yield equivalent results[24]. Our objective was to detect the trends in such possible separation conditioned by the underlying Aβ and tau AD data and, by doing so, to better characterize in vivo human disease mechanisms. Unequivocal evidence across analyses indicates the existence of a significant neuronal excitability (and functional) change in AD that relates to the disease's physical progression: the more pathology there is, the more the neuronal populations fire.

Beyond AD-related protein deposition, our method can also investigate the influence of other critical factors. It has been hypothesized, and to some extent observed[7,12,13], that microglial activation (a probable marker for neuroinflammation[60,61]) affects excitability and neuronal activity in AD. Consequently, we performed a set of complimentary experiments where we recreated the obtained results in a model that also considers deviations to neuronal excitability due to microglial activation –measured with 18kDa Translocator Protein PET. Despite the slightly better fit in terms of resembling the real resting state fMRI indicators (0.50 ± 0.07 vs 0.44 ± 0.10 correlation in the model without the microglial activation term), we did not find substantial neuronal excitability or spectral electrophysiological separation between clinical groups when the microglial activation factor was considered, nor the estimations were confirmed by the participants' plasma and gray matter atrophy markers (Supplementary Figs. 8–10). Moreover, the synergistic interaction of Aβ and tau was the factor that better predicted cognitive impairment, with no significant effect by the microglial activation term (Supplementary Table 4). We attribute these results to model overfitting and/or technical limitations associated with the acquisition of microglial activation. Unlike the Aβ and tau PET SUVRs data, which showed extended statistically significant differences across all brain regions for CU and AD participants (ANCOVA post-hoc t-tests with age and sex as covariates, $p < 0.05$), microglial activation images exhibited differences in only 30 regions (i.e., approx. 45%; Supplementary Table 5). Microglial activation is thought to have a neuroprotective character (M2-phenotype) at early disease stages[12,13]. On the other hand, excessive activation of microglia seemingly becomes detrimental in clinical AD (M1-phenotype) by releasing pro-inflammatory cytokines that may exacerbate AD progression[12,13,61]. Nevertheless, modern neuroinflammation PET tracers are not specific to these two different phenotypes as no consistent targets have been discovered[13]. Thus, our extended results albeit being relatively uninformative in terms of AD-affectations to neuronal excitability, capture intrinsic microglial activation PET mapping insufficiencies[60].

Our methodology also has limitations. Although we used state-of-the-art fMRI experiments in this study (TR = 681 ms, spatial resolution = 2.5 × 2.5 × 2.5 mm³), more detailed spatiotemporal dynamics could be captured with novel ultra high-resolution functional imaging techniques[62]. On the other hand, by using average anatomical connectivity, we have singled-out the mechanisms by which toxic protein deposition and neuroinflammation are associated with pathological neuronal activity. Personalized therapeutic interventions[1] would require precise individual profiles for increased efficiency. In such applications, including the connectomes' individual variability may be beneficial. Regarding the neurophysical model for the influence of pathophysiological factors, two aspects should be considered in future work. Firstly, extending the intra-regional neuronal interactions with additional excitatory and inhibitory populations, pursuing a finer descriptive scale, will also enable us to account for additional significant disease factors such as neuronal atrophy[11]. Secondly, the effects on inhibitory firings should be explored separately as well. Pyramidal (excitatory) neurons greatly outnumber any other neuronal population, making them the most likely proteinopathies target[3]. However, inhibitory populations are key in maintaining healthy firing balances[3] and interacting with glial cells[63]. To generate plausible signals and compare results across participants/disease states, the individual calculations were run under equal

experimental conditions. These constraints yielded stringent parameter optimization bounds ([−0.05, 0.05] for each pathophysiological influence and their combined effect). It is possible that new optima existed outside of these intervals in different conditions. Finally, the focus of this study was limited to capturing abnormalities in AD by Aβ and tau's combined action. The model inputs will require modifications to measure neuronal excitability contributions in other neurodegenerative conditions given their characteristic neuropathological factors. For example, dopamine transporter (DaT) $^{123}$I–FP-CIT scans can be used to quantify dopaminergic deficiency consistent with Parkinsonism and associated disorders[64]. Ongoing efforts pursue developing alpha-synuclein protein PET radiotracers that do not also bind to Aβ[65]. Replacing the AD- pathophysiology with such quantified maps in our framework may well help advance the characterization of neuronal excitability dysfunction in the Parkinsonian circuit[66].

The study was undertaken with several computational challenges. First, the dynamical system is nonlinear and highly dimensional (660 variables, for 66 regions). Simulations of such systems are time- and memory-consuming. Parameter identification, in turn, is a much more complex process as many simulations of the system are required as the optimization algorithm evaluates possible solutions[67]. Although it would have been exceptionally informative to obtain voxel-wise neuronal excitability perturbations by Aβ and tau, unfortunately such a task was computationally prohibitive at the present. Inter-individual variability was lost by averaging the pathology and brain function descriptors in relatively coarse brain regions[68]. Furthermore, we selected a surrogate optimization algorithm for its advantages to deal with intensive parameter identification problems[67], imposed constraints based on the biophysical properties of the model[69] and evaluated several random trial points samples for each subject, to increase the chances of finding the global optima. These aiding maneuvers came with additional cost, requiring the utilization of a computing cluster to perform the optimizations in a reasonable time. On the other hand, the initial conditions of generative neural models are generally unknown and were not estimated in this work. To bypass related issues, the initial transient simulation segments were dropped[25,70], and the analyses focused on the comparison of the underlying parameters and signals, which were obtained by assuming equal non-relevant parameters and minimizing an objective function that was built with frequency-domain (fALFF) indicators[1,16,19]. Finally, we must reiterate that fALFFs were preferred to construct the optimization objective function over other widely-spread indicators as functional connectivity due to their unambiguous and straightforward definition in structural inter-connected regions (as opposed to functional connectivity being strictly correct for functional parcellations only and having a myriad of possibly informative –but not definitive– network and node-specific features[31,71]), yet discriminating disease states from a functional standpoint[1,16,19].

Our approach has major implications to disease hypothesis testing. Generative models[23] in works by Iturria-Medina et al.[1,17], Deco et al.[24,72], Sotero et al.[18,70], de Haan et al.[10,22] among others, focus on better comprehending neurological conditions. The models considered in the present study reflect plausible biophysical mechanisms potentially determining neuronal activity abnormalities in the AD spectrum[3,4,7,12]. Critical mechanistic information on the underlying activity-generating processes is obtained, as well as about their relationship with clinical and cognitive profiles, as all these disease-informative variables are tracked in our comprehensive methodology. A critical methodological contribution is the capacity to resolve complex biological processes hidden to current non-invasive imaging and electrophysiological monitoring techniques, e.g., the neural masses' firing excitabilities. For future work, we aim to further clarify the specific molecular features responsible for the differences in excitability values across clinical stages. By doing so, we expect to gain additional insights into AD pathophysiology that could boost diagnostic accuracy and preclinical applications. This AD pathophysiological model is equally applicable to other intricate multifactorial neurological disorders by considering their relevant disease factors. Computational disease modeling may further unveil the complex mechanisms of neurodegeneration and aid providing efficient treatment at a personalized level.

## Methods

### Participants
We selected individuals from the Translational Biomarkers in Aging and Dementia (TRIAD) cohort (https://triad.tnl-mcgill.com/). The study was approved by the McGill University PET Working Committee and the Douglas Mental Institute Research Ethics Boards and all participants gave informed written consent. All ethical regulations relevant to human research participants were followed. All subjects underwent T1-weighted MRI, resting-state fMRI, Aβ ($^{18}$F-NAV4694)-, tau ($^{18}$F-MK-6240)- and translocator protein microglial activation ($^{11}$C-PBR28)- PET scans, together with a complete cognitive evaluation, including the Mini-Mental State Examination (MMSE) and the Montreal Cognitive Assessment (MoCA). We chose baseline assessments in all cases. Only participants with "cognitively unimpaired" ($N = 81$), "mild cognitive impairment" ($N = 35$), or "Alzheimer's disease" ($N = 16$) clinical and pathophysiological diagnoses were considered[73] (see also Supplementary Table 1).

### Image processing
**MRI.** Brain structural T1-weighted 3D images were acquired for all subjects on a 3T Siemens Magnetom scanner using a standard head coil. T1 space sequence was performed in sagittal plane in 1 mm isotropic resolution; TE = 2.96 ms, TR = 2300 ms, slice thickness = 1 mm, flip angle = 9 deg, FOV = 256 mm, 192 slices per slab. The images were processed following a standard voxel-based morphometry pipeline[1,41,43], including non-uniformity correction using the N3 algorithm and segmentation into gray matter (GM), white matter (WM) and cerebrospinal fluid probabilistic maps (SPM12, www.fil.ion.ucl.ac.uk/spm). Each GM and WM map was non-linearly registered (with modulation) to the MNI space[74] using the DARTEL tool[75] and smoothed with a Gaussian kernel of full width half maximum (FWHM) of 8 mm[41,43]. All images were visually inspected to ensure proper alignment to the MNI template. We selected 66 (bilateral) cortical regions in the Desikian–Killiany–Touriner (DKT)[30] atlas (Supplementary Table 5). Subcortical regions, e.g., in the basal ganglia, were not considered given their tendency to present PET off-target binding[76,77].

**fMRI.** The resting-state fMRI acquisition parameters were: Siemens Magnetom Prisma, echo planar imaging, 860 time points, TR = 681 ms, TE = 32.0 ms, flip angle = 50 deg, number of slices = 54, slice thickness = 2.5 mm, spatial resolution = $2.5 \times 2.5 \times 2.5$ mm$^3$, EPI factor = 88. We applied a minimal preprocessing pipeline[1] including motion correction and spatial normalization to the MNI space[74] using the registration parameters obtained for the structural T1 image, and removal of the linear trend. We calculated the fractional amplitude of low-frequency fluctuations (fALFF)[16], a regional proxy indicator for neuronal activity that has shown high sensibility to disease progression. Briefly, we transformed the signals for each voxel to the frequency domain and computed the ratio of the power in the low-frequency range (0.01–0.08 Hz) to that of the entire BOLD frequency range (0–0.25 Hz) with code from the RESTplus toolbox[78]. The fALFF values were ultimately averaged over the voxels according to their belonging to brain regions.

**Diffusion weighted MRI (DW-MRI).** High angular resolution diffusion imaging (HARDI) data was acquired for $N = 128$ cognitively unimpaired subjects in the Alzheimer's Disease Neuroimaging Initiative (ADNI) (adni.loni.usc.edu). The authors obtained approval from the ADNI Data Sharing and Publications Committee for data use and publication, see documents http://adni.loni.usc.edu/wp-content/uploads/how_to_apply/ADNI_Data_Use_Agreement.pdf and http://adni.loni.usc.edu/wp-content/uploads/how_to_apply/ADNI_Manuscript_Citations.pdf, respectively. For each diffusion scan, 46 separate images were acquired,

with 5 $b_0$ images (no diffusion sensitization) and 41 diffusion-weighted images (b = 1000 s/mm$^2$). ADNI aligned all raw volumes to the average $b_0$ image, corrected head motion and eddy current distortions. Region-to-region anatomical connection density matrices were obtained using a fully automated fiber tractography algorithm[79] and intravoxel fiber distribution reconstruction[80]. For any subject and pair of regions $k$ and $l$, the $C_{lk}$ measure ($0 \leq C_{lk} \leq 1$, $C_{lk} = C_{kl}$) reflects the fraction of the region's surface involved in the axonal connection with respect to the total surface of both regions. More details can be found in a previous publication where ADNI's DW-MRI was utilized[1]. We averaged the ADNI subject-specific connectivity matrices[1,81] to utilize a single, representative anatomical network across our calculations on the TRIAD dataset.

**PET**. Study participants had Aβ ($^{18}$F-NAV4694), tau ($^{18}$F-MK-6240) and translocator protein microglial activation ($^{11}$C-PBR28) PET imaging in a Siemens high-resolution research tomograph. A bolus injection of $^{18}$F-NAV4694 was administered to each participant and brain PET imaging scans were acquired approximately 40–70 min post-injection. The images were reconstructed using an ordered subset expectation maximization (OSEM) algorithm on a 4D volume with three frames ($3 \times 600$ s)[50]. $^{18}$F-MK-6240 PET scans of 20 min ($4 \times 300$ s) were acquired at 90–110 min after the intravenous bolus injection of the radiotracer[82]. $^{11}$C-PBR28 images were acquired at 60–90 min after tracer injection and reconstructed using the OSEM algorithm on a 4D volume with 6 frames ($6 \times 300$ s)[61]. Images were preprocessed according to four main steps[83]: 1) dynamic co-registration (separate frames were co-registered to one another lessening the effects of patient motion), 2) across time averaging, 3) re-sampling and reorientation from native space to a standard voxel image grid space ("AC-PC" space), and 4) spatial smoothing to produce images of a uniform isotropic resolution of 8 mm FWHM. Using the linear and nonlinear registration parameters obtained for the participants' structural T1 images, all PET images were spatially normalized to the MNI space. $^{18}$F-MK-6240 images were meninges-striped in native space before performing any transformations to minimize the influence of meningeal spillover. SUVR values for the DKT gray matter regions were calculated using the cerebellar gray matter as the reference region.

The DKT atlas was separately used to define the ROIs for tau-PET Braak stage-segmentation[37,38] which consisted of: Braak I (pathology confined to the transentorhinal region of the brain), Braak II (entorhinal and hippocampus), Braak III (amygdala, parahippocampal gyrus, fusiform gyrus and lingual gyrus), Braak IV (insula, inferior temporal, lateral temporal, posterior cingulate and inferior parietal), Braak V (orbitofrontal, superior temporal, inferior frontal, cuneus, anterior cingulate, supramarginal gyrus, lateral occipital, precuneus, superior parietal, superior frontal and rostromedial frontal) and Braak VI (paracentral, postcentral, precentral and pericalcarine)[84]. All image processing was performed in MATLAB 2021b (The MathWorks Inc., Natick, MA, USA) with the aid of the specific tools and algorithms specified above.

### Plasma biomarkers
Blood biomarkers were quantified with Single molecule array (Simoa) assays (Quanterix, Billerica, MA). These measurements included tau phosphorylated at threonine 181 (p-tau181)[41], tau phosphorylated at threonine 231 (p-tau231)[40], tau phosphorylated at threonine 217 (p-tau217)[39,85] and glial fibrillary acidic protein (GFAP)[42] and have been previously reported.

### Personalized integrative AD neuronal activity model
**Electrophysiological model.** Following the specialized literature[22,32–36], we utilized coupled neural masses to model electrophysiological brain activity (with personalized model corrections accounting for the pathophysiological AD effects, see below). Neural masses represent the average dynamic behavior of similar neurons within a given spatial domain, i.e., brain regions[18,36,86]. In the seminal Wilson–Cowan (WC) model[36], excitatory and inhibitory populations are locally coupled. These neuronal populations are described by their firing rates, $E(t)$ and $I(t)$, respectively.

Additionally, the excitatory population is further stimulated by unspecific local inputs ($P$) and cortico-cortical interactions with other WC modules in the brain network (Supplementary Fig. 11a)[33,34,70]. In effect, each $l$ region influences the dynamics of the $k$ region by the quantity $\frac{\eta}{N} C_{lk} E_l$, where $\eta$ is a global scaling coupling strength and $N$ is the total number of regions in our considered parcellation ($N = 66$). We performed a dynamical system analysis[34–36,86,87] and obtained $P$ and $\eta$ values that simulate plausible electrophysiological oscillations and BOLD signals within the considered range of pathophysiological affectations (Supplementary Fig. 12). All other model parameters were set at generic WC values[32–36,88] (Supplementary Table 6).

To investigate the in vivo neuronal excitability affectations by AD pathophysiology[3–7,22] in the human brain, perturbations to the model's excitability parameter by Aβ, tau and their synergistic interaction are quantified, for each individual. In the neural mass framework, the integration of all inputs received by the neuronal population is achieved by means of a sigmoidal activation function[36], $S(x) = \frac{1}{1+\exp[-a(x-\theta)]} - \frac{1}{1+\exp[a\theta]}$, where $\theta$ is the firing threshold and $x$ is the input current (synthetic EEG signals are proportional to the regional excitatory input current[32]). Compared to "baseline" firings, regional excitability can be higher (hyperexcitability) or lower (hypoexcitability) depending on whether the firing rate function is shifted to lower or higher input current values, respectively (Supplementary Fig. 11b)[22,34]. The neural masses' activation functions (thus, their firing properties) are determined by the threshold parameters. We suppose that the effective regional firing threshold values are mediated by the following disease factors: Aβ plaques (with a subject-specific contribution weight given by $\theta_j^{A\beta}$), tau tangles ($\theta_j^{Tau}$) and the interaction of amyloid and tau ($\theta_j^{A\beta \cdot Tau}$)[3–7,22]. Based on the much larger excitatory prevalence in the cortex[3,14], we also assume that the regional excitability profiles are quantified through the excitatory firing threshold ($\theta_E$) only. The pathophysiological effects are simplistically written as linear fluctuations from the normal baseline value due to the participant's regional accumulation of each AD-relevant factor (Supplementary Fig. 11c), with each PET modality's SUVRs normalized to the [0,1] interval (by dividing by the maximum value across subjects and regions[89], Supplementary Fig. 13), to preserve the dynamical properties of the generated signals and compare values across subjects and conditions:

$$\theta_{j,k} = \theta_0 + \theta_j^{A\beta} \cdot A\beta_k + \theta_j^{Tau} \cdot Tau_k + \theta_j^{A\beta \cdot Tau} \cdot A\beta_k \cdot Tau_k \quad (1)$$

where, as above, the index $k$ denotes the brain region, and $j$ is used to identify the participant. A negative contribution by a factor ($\theta_j^{A\beta}$, $\theta_j^{Tau}$ or $\theta_j^{A\beta \cdot Tau}$) means that the pathological accumulation of such a biomarker tends to decrease the firing threshold thus yielding hyperexcitability. Given the inverse relationship existing between firing thresholds and effective firing rates[22], we define excitability as $1/\theta_{j,k}$.

The evolution of the average firing rates $E(t)$ and $I(t)$ is given by the following set of differential equations[34–36]:

$$\dot{E}_k = \frac{1}{\tau_E} \left[ -E_k + S(x_{E,k}) \right] \quad (2)$$

$$\dot{I}_k = \frac{1}{\tau_I} \left[ -I_k + S(x_{I,k}) \right]$$

$$x_{E,k} = C_{EE}E_k - C_{IE}I_k + P + \frac{\eta}{N} \sum_{l=1, l \neq k}^{N} C_{lk}E_l$$

$$x_{I,k} = C_{EI}E_k - C_{II}I_k$$

where we have dropped the subject's tag, $j$, for readability purposes. The participant's regional BOLD signals are consequently generated via a metabolic and hemodynamic model (MHM) by Sotero et al.[49,69,70,90]. For the sake of completeness, the corresponding transformations are provided below. The specific parameter values and their interpretation can be found in Supplementary Table 7 and the references therein.

In the MHM, the total action potential arriving to the neuronal populations from other local and external populations ($S(x_{E,k})$ and $S(x_{I,k})$) reflect the role that excitatory and inhibitory activities play in generating the BOLD signal[70,91]. All variables are normalized to baseline values. Thus, the neuronal inputs in region $k$ are computed as $\xi_{E,k} = \frac{S_{E,k}}{S_{E,k}^0}$ and $\xi_{I,k} = \frac{S_{I,k}}{S_{I,k}^0}$, where the scaling constants denote values at rest[49,69,70,90].

Changes in glucose consumption ($g_{E,k}$ and $g_{I,k}$) are linked to the excitatory and inhibitory inputs, specifically:

$$\dot{g}_{E,k} = z_{E,k}$$

$$\dot{z}_{E,k} = \frac{-2}{\kappa_E} z_{E,k} - \frac{1}{\kappa_E^2}\left(g_{E,k} - 1\right) + \frac{h_E}{\kappa_E}\left(\xi_{E,k} - 1\right)$$

$$\dot{g}_{I,k} = z_{I,k}$$

$$\dot{z}_{I,k} = \frac{-2}{\kappa_I} z_{I,k} - \frac{1}{\kappa_I^2}\left(g_{I,k} - 1\right) + \frac{h_I}{\kappa_I}\left(\xi_{I,k} - 1\right)$$

The metabolic rates of oxygen for excitatory ($m_{E,k}$) and inhibitory ($m_{I,k}$) activities, and the total oxygen consumption, $m_k$, are obtained from the glucose variables:

$$m_{E,k}(t) = \frac{2 - x(t)}{2 - x_0} g_{E,k}(t)$$

$$m_{I,k}(t) = g_{I,k}(t)$$

$$m_k(t) = \frac{\gamma m_{E,k}(t) + m_{I,k}(t)}{\gamma + 1}$$

$$x(t) = \frac{1}{1 + \exp\left[c\left(d - g_{E,k}(t)\right)\right]}$$

Next, CBF dynamics ($f_k$) is modeled as follows[92], assuming that CBF is coupled to excitatory activity:

$$\dot{f}_k = y_k$$

$$\dot{y}_k = \frac{-2}{\kappa_f} y_k - \frac{1}{\kappa_f^2}\left(f_k - 1\right) + \mu\left(\xi_{E,k} - 1\right)$$

The outputs of the metabolic and vascular modules are converted to normalized cerebral blood volume ($b_k$) and deoxy-hemoglobin ($q_k$) content through the Balloon model[93]:

$$\dot{b}_k = \frac{1}{\kappa_0}\left(f_k - f_{out}\right)$$

$$\dot{q}_k = \frac{1}{\kappa_0}\left(m_k - f_{out}\frac{q_k}{b_k}\right)$$

$$f_{out} = b_k^{\frac{1}{\xi}}$$

The BOLD signal is then obtained by using a linear observation equation:

$$BOLD_k(t) = V_0\left(a_1\left(1 - q_k\right) - a_2\left(1 - b_k\right)\right)$$

where $a_1 = 4.3\Upsilon_0 E_0 \cdot TE + \varepsilon r_0 E_0 \cdot TE$ and $a_2 = \varepsilon r_0 E_0 \cdot TE + \varepsilon - 1$ are parameters that depend on the experimental conditions (field strength, $TE$)[24,94–96]. The sets of equations above were solved, for each individual dataset, with an explicit Runge-Kutta (4,5) method, ode45, as implemented in MATLAB 2021b (The MathWorks Inc., Natick, MA, USA) and a timestep of 0.001 s. The first 20 s of all simulations were discarded to avoid transient behavior[25,70].

**Parameter estimation.** The personalized estimation of the optimal pathological influences set ($\theta_j^{A\beta}, \theta_j^{Tau}, \theta_j^{A\beta\cdot Tau}$) was performed via surrogate optimization (MATLAB 2021b's surrogateopt). This parameter optimization method performs few objective function evaluations hence it is well-suited for computationally expensive cost functions as it is the case of our high-dimensional BOLD-simulating dynamical system. For each participant, we identified the set of parameters ($\theta_j^{A\beta}, \theta_j^{Tau}, \theta_j^{A\beta\cdot Tau}$) minimizing the correlation distance $[1 - corr(fALFF_{simulated}, fALFF_{real})]$ between the regional fALFF values of the in silico pathological BOLD signals and the subject's real BOLD indicators (Supplementary Fig. 11d). The individual and combined effects of ($\theta_j^{A\beta}, \theta_j^{Tau}, \theta_j^{A\beta\cdot Tau}$) on regional excitability (Eq. (1)) were constrained to $[-0.05, 0.05]$ to preserve the dynamical properties of the signals (see also Supplementary Fig. 12, Supplementary Table 6) and to compare results across subjects and disease states, as all Aβ and tau SUVRs were normalized to same interval. Optimization iterations were performed until surrogateopt found a point satisfying the constraints and too few new feasible points were found to continue (exitflag = 3). This occurred in less than 2000 iterations for all subjects. Several surrogate optimization random trial points initializations were run for each subject (20 series of evaluated points or more, see below). The global optimum was selected as the parameter set with the smallest objective function value amongst all runs for the participant, as it is unlikely to obtain a perfect similarity (correlation distance = 0) in a problem with real data. All optimizations run in the platforms of the Digital Research Alliance of Canada due to their high computational requirements. Around 10% of the subjects were arbitrarily chosen and had 20 additional random trial points surrogateopt evaluations in a desktop computer, all producing the previously identified ($\theta_j^{A\beta}, \theta_j^{Tau}, \theta_j^{A\beta\cdot Tau}$) set for the given participant.

**Interpreting the pathophysiological effects on neuronal activity.** The obtained pathological influences ($\theta_j^{A\beta}, \theta_j^{Tau}, \theta_j^{A\beta\cdot Tau}$) describe subject-specific interactions determining brain activity. We use these weights to reconstruct otherwise hidden electrophysiological quantities of interest. Individual neuronal excitability patterns[22] are mapped through Eq. (1) and can be related to separate measurements like plasma biomarkers for AD[40–42]. Grand average excitatory activities are found by averaging the firing rates $E_k(t)$ over the regions and time points[22], for every subject. Likewise, the excitatory input currents of Eq. (2) are used as proxy measures for cortical sources of resting-state EEG[32]. We perform a Fast Fourier Transformation power analysis of the neural masses' signals and obtain the relative power of the traditional rhythms, in particular: theta (4–8 Hz) and alpha1 (8–10 Hz) frequency band oscillations[22]. Additionally, we investigate the relationship of the obtained pathophysiological influences with cognition[97,98].

**Statistics and reproducibility**
Clinical diagnosis and PET-imaging Aβ status (determined visually by consensus of two neurologists blinded to the diagnosis) were used to divide the cohort for analyses of the results. Separately, we employed another division based on the conventional unambiguous Braak grouping[38] of I-II (transentorhinal stages), III–IV (limbic) and V–VI (isocortical), to assess trends in terms of intracellular tau neurofibrillary changes (see also

Supplementary Tables 2 and 3). Group-differences in the electrophysiological quantities of interest (average intra-brain theta and alpha1 power, excitatory firing activity and excitability) were evaluated with ANCOVA post-hoc t-tests, i.e., we looked at the effects of the clinical groups and Aβ positivity/Braak stages on the corresponding quantity, accounting for age and sex. The average theta and alpha1 power and excitatory firing activity were box-cox and z-score transformed across subjects. The associations between excitability and plasma biomarkers/gray matter atrophy were tested using Spearman's Rho correlation (large-sample approximation). In addition, to assess the relationship between the pathophysiological factors and cognitive integrity we fitted multiple linear regression models using the following specifications: $MMSE\ score \sim 1 + \theta^{A\beta} + \theta^{Tau} + \theta^{A\beta \cdot Tau} + sex + age + education$ and $MoCA\ score \sim 1 + \theta^{A\beta} + \theta^{Tau} + \theta^{A\beta \cdot Tau} + sex + age + education$. Each of the pathophysiological neuronal activity effects were standardized using the mean and standard deviation from all subjects. Statistical model comparison (baseline excitability vs regional Aβ, tau and Aβ·tau influences on excitability) accounting for the difference in model size was evaluated via subject-specific $F$-tests (dfe$_1$ = 65 and dfe$_2$ = 62). Additionally, the Akaike information criterion was calculated as $AIC_j = N \cdot \ln[\frac{RSS_j}{N}] + 2 \cdot dfe$, where $N = 66$ and $RSS_j$ is the residual sum of squares for each subject $j$ under a given model.

## Reporting summary

Further information on research design is available in the Nature Portfolio Reporting Summary linked to this article.

## Data availability

The main source data supporting the findings of this study are available by submitting a data share request via https://triad.tnl-mcgill.com/contact-us/. All the data collected under the TRIAD cohort is governed by the policies set by the Research Ethics Board Office of the McGill University, Montreal and the Douglas Research Center, Verdun. Other data and sources are available from the corresponding author on reasonable request. The source data behind the graphs in the paper can be found in Supplementary Data 1.

## Code availability

The code utilized in this article for the neuronal activity simulations and quantification of the pathological effects can be accessed at the Neuroinformatics for Personalized Medicine lab's website (NeuroPM, https://www.neuropm-lab.com/publication-codes.html) and is freely available and documented on the Zenodo repository[99]. The algorithm is detailed in Supplementary Note 1.

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

## Acknowledgements

LSR was partially supported by funding from the Fonds de recherche du Québec—Santé and the Healthy Brains for Healthy Lives (HBHL) initiative. This project was undertaken thanks in part to the following funding awarded to YIM: the Canada Research Chair tier-2, the CIHR Project Grant 2020, and the Weston Family Foundation's AD Rapid Response 2018 and Transformational Research in AD 2020. In addition, we used the computational infrastructure of the McConnell Brain Imaging Center at the Montreal Neurological Institute, supported in part by the Brain Canada Foundation, through the Canada Brain Research Fund, with the financial support of Health Canada and sponsors. P.R.N., G.B., J.T. and J.F. are supported by the Canadian Institutes of Health Research (CIHR) [MOP-11-51-31; RFN 152985, 159815, 162303], Canadian Consortium of Neurodegeneration and Aging (CCNA; MOP-11-51-31 -team 1), Weston Brain Institute, the Alzheimer's Association [NIRG-12-92090, NIRP-12-259245], Brain Canada Foundation (CFI Project 34874; 33397), the Fonds de Recherche du Québec—Santé (FRQS; Chercheur Boursier, 2020-VICO-279314) and the Colin J. Adair Charitable Foundation. H.Z. is a Wallenberg Scholar supported by grants from the Swedish Research Council (#2022-01018), the European Union's Horizon Europe research and innovation programme under grant agreement No 101053962, Swedish State Support for Clinical Research (#ALFGBG-71320), the Alzheimer Drug Discovery Foundation (ADDF), USA (#201809-2016862), the AD Strategic Fund and the Alzheimer's Association (#ADSF-21-831376-C, #ADSF-21-831381-C, and #ADSF-21-831377-C), the Bluefield Project, the Olav Thon Foundation, the Erling-Persson Family Foundation, Stiftelsen för Gamla Tjänarinnor, Hjärn-fonden, Sweden (#FO2022-0270), the European Union's Horizon 2020 research and innovation programme under the Marie Skłodowska-Curie grant agreement No 860197 (MIRIADE), the European Union Joint Pro-gramme—Neurodegenerative Disease Research (JPND2021-00694), and the UK Dementia Research Institute at UCL (UKDRI-1003). K.B. is supported by the Swedish Research Council (#2017-00915 and #2022-00732), the Alzheimer Drug Discovery Foundation (ADDF), USA (#RDAPB-201809-2016615), the Swedish Alzheimer Foundation (#AF-930351, #AF-939721 and #AF-968270), Hjärnfonden, Sweden (#FO2017-0243 and #ALZ2022-0006), the Swedish state under the agreement between the Swedish gov-ernment and the County Councils, the ALF-agreement (#ALFGBG-715986 and #ALFGBG-965240), the European Union Joint Program for Neurode-generative Disorders (JPND2019-466-236), the National Institute of Health (NIH), USA, (grant #1R01AG068398-01), the Alzheimer's Association 2021 Zenith Award (ZEN-21-848495), and the Alzheimer's Association 2022-2025 Grant (SG-23-1038904 QC). T.K.K. was funded by the Swedish Research Council (Vetenskapsrådet #2021-03244), the Alzheimer's Association

Research Fellowship (#AARF-21-850325), the Swedish Alzheimer Foundation (Alzheimerfonden), the Aina (Ann) Wallströms and Mary-Ann Sjöbloms stiftelsen, and the Emil och Wera Cornells stiftelsen.

## Author contributions

L.S.R. and Y.I.M. conceived the theoretical formalism. P.R.N., S.S., N.R., C.T., J.S., T.K., N.A., A.B., H.Z., K.B., G.T., H.K. collected the data. G.B. and L.S.R. processed the data. L.S.R. implemented the algorithm and performed the computations. Y.I.M., F.C., G.B., J.T. and J.F.A. contributed to the analyses. L.S.R. wrote the manuscript draft. YIM supervised the project. All authors discussed the results and contributed to the final manuscript.

## Competing interests

H.Z. has served at scientific advisory boards and/or as a consultant for Abbvie, Acumen, Alector, Alzinova, ALZPath, Annexon, Apellis, Artery Therapeutics, AZTherapies, CogRx, Denali, Eisai, Nervgen, Novo Nordisk, Optoceutics, Passage Bio, Pinteon Therapeutics, Prothena, Red Abbey Labs, reMYND, Roche, Samumed, Siemens Healthineers, Triplet Therapeutics, and Wave, has given lectures in symposia sponsored by Cellectricon, Fujirebio, Alzecure, Biogen, and Roche, and is a co-founder of Brain Biomarker Solutions in Gothenburg AB (BBS), which is a part of the GU Ventures Incubator Program (outside submitted work). K.B. has served as a consultant, at advisory boards, or at data monitoring committees for Acumen, ALZPath, BioArctic, Biogen, Eisai, Julius Clinical, Lilly, Novartis, Ono Pharma, Prothena, Roche Diagnostics, and Siemens Healthineers, and is a co-founder of Brain Biomarker Solutions in Gothenburg AB (BBS), which is a part of the GU Ventures Incubator Program, outside the work presented in this paper. The other authors declare no competing interests.

## Additional information

¹Department of Neurology and Neurosurgery, McGill University, Montreal, QC, Canada. ²McConnell Brain Imaging Centre, Montreal Neurological Institute, Montreal, QC, Canada. ³Ludmer Centre for Neuroinformatics & Mental Health, Montreal, QC, Canada. ⁴McGill University Research Centre for Studies in Aging, Douglas Research Centre, Montreal, QC, Canada. ⁵Biospective Inc., Montreal, QC, Canada. ⁶Lawrence Berkeley National Laboratory, Berkeley, USA. ⁷Department of Psychiatry and Neurochemistry, Institute of Neuroscience and Physiology, The Sahlgrenska Academy at the University of Gothenburg, Mölndal, Sweden. ⁸Department of Psychiatry, School of Medicine, University of Pittsburgh, Pittsburgh, PA, USA. ⁹King's College London, Institute of Psychiatry, Psychology and Neuroscience Maurice Wohl Institute Clinical Neuroscience Institute, London, UK. ¹⁰NIHR Biomedical Research Centre for Mental Health and Biomedical Research Unit for Dementia at South London and Maudsley NHS Foundation, London, UK. ¹¹Centre for Age-Related Medicine, Stavanger University Hospital, Stavanger, Norway. ¹²Department of Neurodegenerative Disease, UCL Institute of Neurology, Queen Square, London, UK. ¹³UK Dementia Research Institute at UCL, London, UK. ¹⁴Hong Kong Center for Neurodegenerative Diseases, Clear Water Bay, Hong Kong, China. ¹⁵Wisconsin Alzheimer's Disease Research Center, University of Wisconsin School of Medicine and Public Health, University of Wisconsin-Madison, Madison, WI, USA. ¹⁶Clinical Neurochemistry Laboratory, Sahlgrenska University Hospital, Mölndal, Sweden. ¹⁷Neuroscience Biomarkers, Janssen Research & Development, La Jolla, CA, USA. ✉e-mail: yasser.iturriamedina@mcgill.ca

