## [Peer Review File · Communications Biology]

Reviewers' comments:

Reviewer #1 (Remarks to the Author):

This paper analyzes personalized brain activity decoders incorporating data from resting-state functional MRI, amyloid- β ($A\beta$), and tau-PET scans from participants related to Alzheimer's disease (AD). By using this model, they aim to evaluate the direct impact of toxic protein deposition on neuronal activity. Their subject-specific approach has enabled them to uncover interactions in the disease mechanism, specifically how $A\beta$ and tau proteins synergistically impact cognitive impairment and how neuronal excitability increases with disease progression.

The work is interesting because it integrates a number of modalities with different initial temporal and spatial resolutions. However, the rigid hand-engineered feature extraction methods, such as averages over ROIs or power spectrum analysis, reduce the dimensionality and potential intra-subject variability considerably. The analyses are sound, but the claims/conclusions of "direct" or "causal" relations are not convincingly backed by the data. There is a lot of value in the data and the performed analyses, but I recommend some adjustments before publication.

1. The raw data is rich, versatile, and interesting. It should be clearly stated, that a lot of the variability in the data is lost when the features are computed and that the resulting descriptors of brain function and pathology are much reduced in complexity. This enhances a specific signal-to-noise ratio but may eradicate certain other inter-individual differences. Also, it should be noted that there are non-trivial correlations between features that cannot necessarily be dealt with easily.

2. The study participants were categorized according to various criteria such as dementia diagnosis (CN, MCI, AD), $A\beta$ status, etc. $A\beta$ has a very striking effect on excitability; it's almost binary. This should be further explored and discussed. Is it possible that there is a circularity here? In Figure 4, could the authors highlight the $A\beta$ status by color in scatter plots? What was the proportion of $A\beta+$ in the different Braak stages?

3. The statistical analysis with multiple linear regression could be more rigorous. For example, what was done to make sure the conditions of a linear system are met? How were the residuals distributed? What about multi-collinearity? What was the effect size in practical terms?

4. The main conclusion stating that the authors "...map pathophysiological influences on neuronal activity, ..." is not backed by data. Also the claim of "... cast new light on microscopical pathological processes that are inaccessible to traditional neuroimaging methods" is a stretch given the study data.

5. This is probably due to my lack of knowledge of this particular method, but I could not entirely follow the section "Personalized integrative AD neuronal activity decoder". The firing threshold of participant j in region k is a subject-specific linear function of the pathological markers. The relation between the

amount of pathology and the threshold is assumed to be the same across the brain within the same subject, but different across participants. While I can neither understand the details nor have a clear intuition, I don't feel competent to criticize the use of the method.

6. Are the authors sure that there are no circular arguments in their analyses? Consider the following thought experiment: All participants have a similar neural activation pattern. Some have, however, more A β across brain regions and are classified as A β +. Would you expect to see a group difference in Fig. 3?

Reviewer #2 (Remarks to the Author):

Dear authors,

Thank you for your paper, I hope that you find the following points useful.

The present paper presents a computational model fitted on an individual basis to multi-modal data (gathered and processed by the authors) of subjects in varying stages of Alzheimer's disease. The model in question is a network (connectome) of Wilson-Cowan oscillators and the fitted parameters correspond to each node's/region's excitability. The fitted parameters suggest that amyloid-positivity concurs with overall hyperexcitability. Linear regression analysis also shows that amyloid-beta (and amyloid-tau cross interaction) is significantly correlated with worse cognitive performance per MMSE scores.

The extent of multi-modality in the data used to fit the neural-mass whole-brain models is impressive and, to my knowledge, a novel aspect of the work. However, the use of whole-brain models to infer neuronal parameters from data is not novel, and is, at times, suggested to be novel by the authors (see below). The linear regression analysis showing a positive correlation between amyloid-induced hyperexcitability and cognitive decline is a novel addition (to my knowledge) to investigating excitability in Alzheimer's with whole-brain modeling. The parameter estimation aspect of the work is not discussed sufficiently for the reader to be able to deem the methodology reliable or not. The E/MEG results are also unclear.

In short, the amount and variety of data collection integrated with whole-brain modeling is impressive, however, the parameter estimation methodology needs to be improved and elaborated. The results are very interesting and will be a great addition to the literature. However, it is my opinion that additional work is needed on behalf of the authors to add sufficient rigor and reproducibility to the parameter estimation for the paper to be publishable.

1. The usage of the term: activity decoder. As far as I understand, the authors are introducing the term

activity decoder to refer to inferring neuronal activity by estimating parameters of a computational model. In lines 40-42, it appears to the reader that this is novel. In line 269, the activity decoder is described as a “hybrid approach”. In lines 364-366, the impression of the reader is that the activity decoder is a novel methodological contribution that can infer biological parameters. The fitting of computational models to infer real-world parameters is an established methodology in computational neuroscience and the computational modeling of Alzheimer’s disease. It is my strong suggestion that this term be avoided. It should be noted, however, that I commend the authors’ wish to communicate computational neuroscience to a wider audience.

2. The term “computational model” and “mathematical model” is absent in the paper although the paper deals with parameter estimation of a computational/mathematical model. I would prefer the content of the paper (parameter estimation of whole-brain model for data) to be clear from the title and abstract. Which model (whole-brain Wilson-Cowan) is used should also be clear from the abstract.

3. Here are three papers that investigate neuronal excitability in Alzheimer’s disease. I believe the results and methodology from the reviewed paper should be compared with these three papers, as they use a similar computational model to make statements on neuronal excitability in terms of amyloid-beta and tau:

Altered excitatory and inhibitory neuronal subpopulation parameters are distinctly associated with tau and amyloid in Alzheimer’s disease

Kamalini G Ranasinghe^{1*}, Parul Verma², Chang Cai², Xihe Xie², Kiwamu Kudo^{2,3}, Xiao Gao², Hannah Lerner¹, Danielle Mizuiri², Amelia Strom¹, Leonardo Iaccarino¹, Renaud La Joie¹, Bruce L Miller¹, Maria Luisa Gorno-Tempini¹, Katherine P Rankin¹, William J Jagust⁴, Keith Vossel^{1,5}, Gil D Rabinovici^{1,2}, Ashish Raj², Srikantan S Nagarajan²

A multi-scale model explains oscillatory slowing and neuronal hyperactivity in Alzheimer’s disease
Christoffer G. Alexandersen¹, Willem de Haan², Christian Bick^{1,3,4,†} and Alain Goriely^{1,†}

Tau Pathology induces Whole-Brain Subcritical Brain Dynamics across the spectrum of Alzheimer’s Disease

Xenia Kobeleva^{1,2} Petra Ritter³ Gustavo Patow⁴ Gustavo Deco⁵

4. In Fig. 3, it is not clear what “Neuronal Excitability” is with respect to the fitted parameters. It would be very interesting to see how tau and amyloid-beta and their interaction themselves affect the subjects, as opposed to their sum. Which is contributing to the hyperexcitability? Is tau inducing hypoexcitability or hyperexcitability? What about their cross-interaction?

5. In lines 290-291, it is suggested that the present paper is providing in-vivo evidence of hyperactivity. It is my opinion that inferring parameters with a computational model is not an “in-vivo” method but is more aptly thought of as “in-silico”. I do not believe the authors meant to say this, but the sentence can

be misleading.

6. In lines 301-302, the meaning of “biomarker capabilities of our estimated excitability parameters” is not sufficiently clear to me.

7. Line 469: Where are the generic Wilson-Cowan parameters taken from?

8. It should be clear from the Results Section that the excitability in question is the neuronal excitability of excitatory neurons and not inhibitory neurons.

9. In the Methods Section on the computational model, I recommend presenting the computational model first and then immediately explaining the variables and parameters afterward. This is common practice and is generally deemed to be clearer to read.

10. In line 466, it is unclear what kind of dynamical systems analysis was performed. What is meant by plausible electrophysiological oscillations and BOLD signals? In particular, what frequency are the Wilson-Cowan oscillators running at? Are the WC parameters set globally across all subjects?

11. In line 422, it should be made clear in the Results that the connectome used for simulations is averaged across subjects. Particularly, as the word “personalized” model is being used, it might mislead readers into thinking the connectomes are used per individual.

12. With regards to the E/MEG simulations: the Wilson-Cowan model typically operates at a single frequency (PSD will reveal a single peak). It should be clear what frequency an isolated Wilson-Cowan model operates at with the parameters set by the authors. Analyzing the frequency band outside of this frequency does not really make sense (unless it can be demonstrated that the PSD shows multiple peaks). As such, I am not confident in the electrophysiological simulations and the claim that the model captures both phenomena seen in theta and alpha bands before I see a more detailed analysis.

13. How the computational model is solved is not presented. It would be assuring to see a neural mass signal compared to a BOLD signal (it would also clarify what is meant by setting the parameters such that the signal is comparable to BOLD and MEG signals), to see that the integration has been done successfully. It should also be clear how the initial conditions have been set.

14. Parameter estimation. The objective function used to estimate the parameters using MATLAB's surrogate optimization method is not clear. A mathematical definition of the objective function should be provided.

15. Parameter estimation: Which parameters are being optimized over, is not entirely clear. Is θ_0 being optimized as well?

16. Assuming all θ (4 parameters) are being optimized per subject, it is not trivially true that any

optimization method can be successful in finding optimal parameters, and it depends particularly on the objective function. In line 546, the authors say that it is “guaranteed” that an optimal solution is found. No optimization method can guarantee the identification of a global maximum.

17. Previous work in inferring parameters of whole-brain models, typically will maximize the Pearson correlation between simulated and experimental fMRI resting-state functional connectivity. It is established in the literature that we should expect Pearson correlations of about 0.4. Without a discussion of how these optimization methods compare with previous work is it difficult to ascertain whether the global optimization methodology is trustworthy or not. (of course, MATLAB’s surrogate method works fine, but the question is whether the particular optimization problem can be solved to a satisfactory degree, and this has to be demonstrated, as it is not trivially true).

18. After reading the documentation on MATLAB’s surrogate method, it is not clear to me what is meant by lines 544-545 that “no new feasible points are found”. This should be elaborated, as it seems stopping criteria are usually defined by objective function values and total iterations per MATLAB’s documentation.

19. It should also be checked whether the surrogate method returns the same parameter values when rerun with different parameter settings (stricter stopping criteria) to demonstrate that it is converging (and converging to the same value per run). It should also be verified that the optimization method identifies the same optimal parameters for different initial conditions. Ideally, this should be demonstrated with a figure or, at least, be discussed in the text.

Reviewer #3 (Remarks to the Author):

The manuscript by Sanchez-Rodriguez and colleagues proposes a model that links amyloid-B and tau effects on functional MRI in Alzheimer’s disease. The authors use amyloid-B and tau PET data to inform the a large-scale computational model that generates simulated the fractional amplitudes of low-frequency fluctuations (fALFF) of BOLD signals. The main motivation of the manuscript and the idea of using PET maps to infer the alterations in excitability during AD progression is novel and highly interesting. Nevertheless, the manuscript does not provide convincing evidence that the model offers any meaningful information that explains the observed empirical data.

Major comment 1: The most important concern derives from the fact that the authors did not include any analyses regarding the PET SUVR values in the manuscript. It appears in Supplementary Figure 8 that after the normalization the subjects that are classified as A β -positive have substantially higher values than A β -negative ones, as expected. However, without knowing these data-driven results it is not possible to evaluate whether the proposed computational model adds any useful information to the observed differences in the empirical data. Being more specific it is very likely that the results regarding neuronal excitabilities, theta and alpha power, and plasma tau correlations simply reflects the substantial

differences in PET SUVR values. In such case we may not draw any conclusions that the model provides any useful information beyond the data-driven results. It is also clear in Supplementary Figure 8 that the differences in fALFF values, if any, are not as substantial as PET SUVR values. As a result any model or any model parameter would produce similar results to compensate for the differences in PET SUVR values if provided rather than explaining the subtle differences in the observable (i.e. fALFF). To test this the authors might replicate the results with other biophysically-relevant cortical maps or spatial-autocorrelation matched random maps with differences comparable to empirical PET SUVR values. For example, lack of positive results with microglial activation would also be explained simply by not having the same empirical differences in this particular measure. Similarly, other than excitability, any other model parameter such as unspecific local inputs or global coupling would manifest a similar contrast between groups if provided by the average PET SUVR values. Therefore, the only novel information provided by the model is that the changes in excitability simultaneously explains the alterations in theta and alpha power. However, that could be shown by using a generic model without a necessity to fit to the empirical data focusing on additional insights regarding the relationship between excitability and theta/alpha power. Therefore, to support the claims and the conclusions of the manuscript one would expect to see the group differences in raw PET SUVR scores which appears to be the main result of the study and the relationships between these PET SUVRs and fALFF (and preferably functional measure) profiles before and after fitting the model. Then systematically introduce the model that can explain the differences in empirically observed measures (i.e. fALFF and/or functional connectivity), preferably providing mechanistic insights. I believe the authors already have proper data with empirical results that could produce such study.

Major Comment 2: Similar to the first point, the results in figure 4 seems to show the group differences in PET SUVR values rather than excitability. In other words, the use of correlation coefficient to describe the relationship between excitability/SUVR and plasma AD biomarkers is misleading because the main effect is driven by the group differences. Nevertheless, the figure also suggests a positive relationship with AD biomarkers in the hyper-excitable subjects. The authors could demonstrate these relationships using a more adequate analyses.

Minor Comment 1: The rationale behind choosing fALFF to fit the model is not clear in the manuscript. In methods section authors claim that this indicator has shown high sensibility to disease progression. However, there is a much extensive literature on functional connectivity related to the disease progression. Furthermore, there might be differences in functional and fALFF measures regarding their involvement in AD as well as the model dynamics. Therefore authors should at least elaborate their rationale to choose this specific measure.

Minor Comment 2: The main text lacks a lot of relevant details to understand and interpret the results such as which parcellation is used, which computational model is used, how the model fit to the data, why fALFF is chosen...etc. Most of the information was found in methods section. However, as a convention it is better to very briefly introduce these details in the main text that are crucial to understand and interpret the results. Moreover, the manuscript lacks any details and results regarding

the model fitting procedure (such as the model fit values, number of iterations, whether there were random initializations...etc), which makes it difficult to evaluate the results.

Reviewers' comments:

General response: We thank the anonymous referees for the thorough revision of the paper and constructive comments. We appreciate the opportunity to clarify important aspects of the methods and results according to the provided feedback.

Reviewer #1 (Remarks to the Author):

This paper analyzes personalized brain activity decoders incorporating data from resting-state functional MRI, A β , and tau-PET scans from participants related to Alzheimer's disease (AD). By using this model, they aim to evaluate the direct impact of toxic protein deposition on neuronal activity. Their subject-specific approach has enabled them to uncover interactions in the disease mechanism, specifically how A β and tau proteins synergistically impact cognitive impairment and how neuronal excitability increases with disease progression.

The work is interesting because it integrates a number of modalities with different initial temporal and spatial resolutions. However, the rigid hand-engineered feature extraction methods, such as averages over ROIs or power spectrum analysis, reduce the dimensionality and potential intra-subject variability considerably. The analyses are sound, but the claims/conclusions of "direct" or "causal" relations are not convincingly backed by the data. There is a lot of value in the data and the performed analyses, but I recommend some adjustments before publication.

General response to Reviewer #1:

We appreciate the reviewer's valuable comments and the opportunity to further improve the manuscript based on this. In this revised manuscript version, we clarified many of the mentioned points and added them in the Results/Discussion sections. Complimentary confirmatory analyses were performed which resulted in an expanded Fig. 4, which now includes relationships between the obtained excitability values and grey matter atrophy, and an attached supplementary figure to show the A β positivity and Braak stages of the participants in the study of correlations with plasma biomarkers. Additionally, several parts of the Discussion were re-written. For instance, we further explain the rationale for using regional average values instead voxels. The comments on the interpretation of the results were also expanded to clarify that neuronal excitability underlying the individual observable activity indicators are higher in advanced disease stages due to the modeled A β and tau effects, i.e., their causal influence. We also re-state that each participant's optimization problem is solved independently from the others (i.e., the only inputs are the participant's A β tau and rs-fMRI-derived spatial distributions). In what follows, we provide a point-by-point response to the reviewer's concerns.

1. The raw data is rich, versatile, and interesting. It should be clearly stated, that a lot of the variability in the data is lost when the features are computed and that the resulting descriptors

of brain function and pathology are much reduced in complexity. This enhances a specific signal-to-noise ratio but may eradicate certain other inter-individual differences. Also, it should be noted that there are non-trivial correlations between features that cannot necessarily be dealt with easily.

Response:

Thank you. This is an excellent point and corresponding changes have been made to the Limitations section of the Discussion to highlight that variability is lost by averaging neuroimaging features over ROIs (lines 419-426). The underlying reason to do so is mainly computational. To estimate the likely excitability values behind each individual's data (excitability is not directly quantifiable with current neuroimaging techniques), it is necessary to simulate many times the full dynamical system within the parameter optimization algorithm. Originally, we considered to solve the optimization problem with about 324,000 gray matter voxels per subject, which resulted in 3,240,000 nonlinear differential equations. Although it would have been exceptionally informative to obtain voxel-wise neuronal excitability perturbations by $A\beta$ and tau, unfortunately such a task was computationally prohibitive. To deal with this issue, we used a brain parcellation, resulting in 660 variables (for 66 brain regions). Still, such optimization effort required 10GB of RAM for each participant, and calculations needed to be performed using High Performance Computing resources from the Digital Research Alliance of Canada. We expect to increase the resolution in the future. This type of computationally expensive problems would particularly benefit from quantum computing capabilities. We agree that the regional approximation could have produced loss of inter-individual and disease-progression information.

2. The study participants were categorized according to various criteria such as dementia diagnosis (CN, MCI, AD), $A\beta$ status, etc. $A\beta$ has a very striking effect on excitability; it's almost binary. This should be further explored and discussed. Is it possible that there is a circularity here? In Figure 4, could the authors highlight the $A\beta$ status by color in scatter plots? What was the proportion of $A\beta+$ in the different Braak stages?

Response:

Thank you for this important observation. This is, indeed, a very interesting effect. To further clarify its biological and technical causes, we generated figures during the preparation of the article that respond to the question and have now been added to the Supplementary Materials (please see Supplementary Fig 4). Additionally, the following numbers reflect the proportion of $A\beta+$ in each of the Braak stages: Braak 0: 0.1364, Braak I-II: 0.5152, Braak III-IV: 1, Braak V-VI: 1.

Biologically, due to A β 's high toxicity and aggressive spreading in the brain, its presence may indeed have a strong (close to binary) effect on neuronal activity. Observations in animal models suggest a notable A β effect on neuronal activity (Bero et al., 2011; Tok et al., 2022). For example, application of soluble A β (found around plaques) to neuronal animal in-vivo circuits induces hyperactivity, and suppression of A β production blocks this hyperactivation (Busche et al., 2012). Neuronal activity, conversely, may regulate A β aggregation (Bero et al., 2011). A β deposition also triggers epileptic form discharges (Tok et al., 2022). While specific research into neuronal activity in-vivo human mechanisms lags animal studies (the motivation for this work), indirect proxy measurements may indicate that neuro-hyperexcitability arises in AD. Experimental evidence also supports the long clinical tradition of separating subjects according to A β status. For instance, a recent multicenter study showed that CU(A β +) subjects, independently of tau status, present substantially increased risk of short-term (3-5 years) conversion to mild cognitive impairment, compared to CU(A β -) (Ossenkoppele et al., 2022). Longitudinal imaging studies validate that A β plaque build-up occurs primarily before the onset of cognitive deficits (Serrano-Pozo et al., 2011). Although some brain regions present an elevated tau deposition too, particularly towards the end of the AD spectrum, tau accumulation follows a different spreading pattern. Interestingly, research has suggested that neuronal hyperexcitability could be the driver of tau pathology propagating from entorhinal cortex to the hippocampus and cortex (Rodriguez et al., 2020).

In the previous version of the article, we had written "The relative preponderance of A β 's effect with respect to tau's was somewhat expected as A β plaques generalize to many cortical areas early in the disease, while NFT spreading increases rapidly in temporal and parietal regions only (Insel et al., 2020). These pathological progression patterns, measured by PET uptake, inform our individual dynamical models." in the last paragraph of the Discussion. We believe that these lines were not sufficiently clear. Now, we have expanded this in a separate paragraph to better explain the specific A β and tau brain propagation patterns and how that may be reflected on the results (lines 348-373).

Finally, there are two important points to observe when analyzing the results: a) each participant was modeled and optimized individually; no information from the others was utilized. Yet, the group results are consistent and significant relationships with plasma biomarkers, grey matter atrophy and cognitive scores were revealed. b) The main finding is that subjects in advanced disease stages come out, on average, with hyperexcitable neuronal populations as estimated by the method, not that the participants are separated in approximately two clusters. A distribution of obtained excitabilities in two clusters was somewhat expected given the distribution of the PET, as mentioned above. However, of the three possible options (a continuum with no differences whatsoever, higher excitabilities for subjects at the beginning of the AD spectrum or higher excitabilities for subjects at the end of the AD spectrum), the most likely excitability values given the individual PET and rs-fMRI data and assumptions in the model are higher for subjects at the end of the AD spectrum. Coincidentally, animal models, post-mortem studies and indirect approximations of neuronal excitability suggest the same mechanism (see (Busche et al., 2019; Busche & Hyman, 2020; Celone et al., 2006; Lauterborn et al., 2021; Tok et al., 2022; Vossel et al., 2017)). Neuronal excitability cannot be directly and non-invasively measured in the living human brain, thus the importance of data-informed computational approaches to map such a mechanistically relevant quantity.

3. The statistical analysis with multiple linear regression could be more rigorous. For example, what was done to make sure the conditions of a linear system are met? How were the residuals distributed? What about multi-collinearity? What was the effect size in practical terms?

Response:

Thank you for this observation. We utilized this opportunity to improve the reporting of the stats. We have also modified the legend of Table 1 to respond the questions raised by the reviewer.

We verified that in both cases (MoCA and MMSE), the residuals of the multiple linear regression were normally distributed (two-sided one sample Kolmogorov-Smirnov test, $p < 0.001$). The pairwise correlation coefficients between the pairs of variables of interest (the set $\theta^{A\beta}$, θ^{Tau} , $\theta^{A\beta \cdot Tau}$) are all lower than 0.4 and two of them are around 0.2 (small to moderate (Gignac & Szodorai, 2016; Shrestha, 2020)). The effect size had been reported in terms of the R^2 of the model. We have now additionally included the t-statistics for all the variables and the Cohen's d coefficients for the effect sizes corresponding to the variables of interest. Please, see lines 281-291. Thanks

4. The main conclusion stating that the authors "...map pathophysiological influences on neuronal activity, ..." is not backed by data. Also the claim of "... cast new light on microscopical pathological processes that are inaccessible to traditional neuroimaging methods" is a stretch given the study data.

Response:

Thank you for this comment. We appreciate the opportunity to further clarify this aspect on the manuscript. We agree with the reviewer that the use of the word microscopical is confusing as we did not image A β and tau PET accumulation at a microscopic level, thus the results presented in the article are only true for the scale of the anatomical regions in the parcellation we used. Nevertheless, neuronal excitability (estimated at the neural masses in these anatomical regions) is a biophysical quantity that is inaccessible to human in-vivo neuroimaging. The sentence has been re-written to say: Our findings confirm previous observations and cast new light on pathological processes that are inaccessible to in-vivo human neuroimaging, as relationships with neuronal excitability in AD. Regarding the first sentence of the Discussion "We developed an integrative biophysical framework to map pathophysiological influences on neuronal activity, with application to AD", it is expressing the relationship that is established through equations (1) and (2) and its biological meaning. In the models, the pathophysiological factors perturb the firing threshold parameters of the differential equations that describe the evolution of neuronal activity. A dynamical effect is measured: neuronal activity simulated under nominal baseline parameters is different than neuronal activity in a model where the regional firing thresholds depend on the subject's A β and tau PET SUVRs. Fitting these equations to the BOLD data allows us to reconstruct the most likely excitability patterns for the subjects in the cohort. Thus, effectively, maps of A β Tau and A β *Tau influences on neuronal activity can be built: each term in equation (1) through the obtained ($\theta_j^{A\beta}$, θ_j^{Tau} or $\theta_j^{A\beta \cdot Tau}$).

5. This is probably due to my lack of knowledge of this particular method, but I could not entirely follow the section "Personalized integrative AD neuronal activity decoder". The firing threshold of participant j in region k is a subject-specific linear function of the pathological markers. The relation between the amount of pathology and the threshold is assumed to be the same across the brain within the same subject, but different across participants. While I can neither understand the details nor have a clear intuition, I don't feel competent to criticize the use of the method.

Response:

Thank you for this observation. The "Personalized integrative AD neuronal activity decoder" section explains the assumptions in our model and how it was all integrated within the neural mass models for neuronal activity and their transformation into BOLD signals. We studied evidence from many works attempting to assess neuronal excitability changes in AD. These studies include in-vivo animal experiments, computational simulations, ex-vivo and indirect in-vivo human measurements. There seem to exist important mechanistic neuronal excitability processes in the disease. However, the many methodological differences across studies and the high invasiveness that impedes to directly measure this quantity in the living human brain constitute a major obstacle towards clarifying excitability-related phenomena (Maestú et al., 2021).

To completely dissect the assumptions in the model, we need to first understand what neuronal excitability is and the possible changes it may experience. A single neuron may fire, on average, a certain number of action potentials per unit of time. Several factors may shift the neuron's firing threshold. Refs 4-10 in the manuscript show how A β and tau deposits interact with neuronal populations to modify their excitability properties; most of this evidence arising from cell culture and animal models. The activation function of a group of similar neurons in a specific location in the brain follows a sigmoid-like pattern (see for example (Eeckman & Freeman, 1991) for a derivation and (Ermentrout, 1998) for general discussions on neuronal models). This constitutes, in fact, the neural mass approximation. As with a single neuron, the neural mass could be firing at a higher or lower rate than in normal baseline conditions. These intrinsic properties of the neuronal populations are given by the conditions in which they operate (A β and tau accumulations, in our AD model) and best modeled through the regional firing threshold parameter. Supplementary Fig. 9b shows how the activation function of the local regional population could be shifted to right (firing less at a given input current, hypoexcitable) or left (firing more at a given input current, hyperexcitable) based on the overall value of the threshold θ

In an approximation that avoids overfitting by adjusting too many parameters, we assume that an intra-brain influence parameter exists for each pathological factor. Looking at the rows in Supplementary Fig 11 a,b, most subjects present a quasi-homogeneous intra-brain pattern of A β and tau involvement in the considered regions, which partly justifies the approximation. It is also worth noticing that those patterns change from subject to subject, explaining why subject-specific sets should be obtained. In the literature, similar assumptions have been made, especially in computational simulation studies. For example, kindly observe equation (12) in (Deco et al., 2018) where a global scaling parameter for each subject multiplies the density of serotonin receptors from PET. The parameter that is consequently obtained acts as a gain in the local neuronal response functions. We also find a subject-specific influence approximation in a work by (Stefanovski et al., 2019) et al modeling neuronal activity in AD. The influence function is not linear in this case, but sigmoidal of the A β burden, which is also chosen ad-hoc. Looking to estimate the contributions of three pathological factors from the data in our work, we adopted a simple linear function of the pathological markers. In doing so, the direction of their influence and magnitude are easily interpretable. The populations can be hyper- or hypoexcitable from a baseline value and the farther they are from this value applied to all the subjects, the greater the hyper or hypoexcitable effect. For example, see Table 1 for an analysis on how each factor's neuronal activity influences predict cognitive scores (the conclusion is that hyperexcitability by A β and A β *Tau, i.e. activation functions shifted to the left, occurs together with cognitive deterioration).

6. Are the authors sure that there are no circular arguments in their analyses? Consider the following thought experiment: All participants have a similar neural activation pattern. Some

have, however, more A β across brain regions and are classified as A β +. Would you expect to see a group difference in Fig. 3?

Response:

We appreciate the opportunity to further discuss the analyses and results regarding this interesting question. It is important to note that differences in terms of functional (rs-fMRI) indicators in the AD spectrum are much more subtle than pathological accumulation levels (Supplementary Table 5). While all regions present statistically significant higher A β and tau deposition in AD, compared to CU subjects in our analyzed cohort, only 12 regions present fALFF differences (which is in line with previous evidence (Yang et al., 2018, 2020)). Other features as functional connectivity (Damoiseaux, 2012; Vemuri et al., 2012) are ill-defined for structural parcellations and differences appear in some default mode network connections only, for example.

Thus, the pathological accumulations of A β and tau do play an important role in producing the close to bimodal distributions of Fig. 3. However, as shown in the response to question 2, not all A β + subjects are in one end of the distribution and the A β - in another. The same is true for Braak 0 and Braak > 0 subjects. This is where the interaction of A β and tau, and the subtle differences in neuronal activation patterns -as measured through the fMRI indicators- seem to come into play. From a clinical perspective, following the subjects who are A β and tau negative but still clustered with the more advanced diseased participants could be insightful in determining whether they transition to disease or not, provided that the data is recorded in studies. As it was previously stated, the separation of subjects into two major groups with parameter identification was somewhat expected given the underlying distribution of the data. However, it is the association of these two clusters with disease stages what we report as striking finding. Various and biologically different, independently measured variables as plasma biomarkers and cognitive scores support a link between the hyperexcitability values identified in this study and AD.

To further confirm this result, we also computed voxel-based morphometry (VBM) values for all the subjects in the study and ROIs in the parcellation. VBM correctly characterizes between groups' regional volume and tissue concentration differences from structural MRI (Tissot et al., 2021; Wang et al., 2015). It is important to highlight that structural MRIs were not used in any step of our models' individualized optimization. Performing analyses like those of previous Fig 4, we observed an analogous behavior: the lower the average VBM values, the more excitable the populations of the participants. Note that a low VBM value indicates brain atrophy, which constitutes an ultimate sign and deleterious consequence of AD. The behavior is even more striking when we focus on regions that prominently showcase neurodegeneration in AD as the hippocampus, entorhinal cortex, posterior cingulate gyrus, amygdala, parahippocampal gyrus and fusiform (see also Supplementary Table 3). These new analyses reaffirming the mechanistic discoveries of this work have been added to section Results, Neuronal hyperexcitability relates to high levels of plasma AD biomarkers and grey matter atrophy.

To conclude, there exists a distribution in the data that somewhat conditions the existence of a distribution in the estimated quantities. This is unavoidable if we want to model Alzheimer's disease and its related spectrum. A β plaques and tau neurofibrillary tangles are hallmark AD signs with a great effect on neuronal activity (Busche et al., 2019; Busche & Hyman, 2020; Celone et al., 2006; Lauterborn et al., 2021; Maestú et al., 2021; Tok et al., 2022; Vossel et al., 2017). That, however, does not entail circularity in the analyses, because each subject is modeled, and her/his parameters are estimated, separately from the others. Each personalized dynamical model is informed by the pathological accumulation and activity indicators of the subject only, and parameter identification is performed in a completely unsupervised manner. A posteriori, when we gather all the subject-specific results and put them in perspective, utilizing the subject's clinical information, is that we observe differences in the estimated quantities. Importantly, these differences appear in a way that puts AD-related hyperexcitability together with cognitive deterioration, grey matter atrophy and high soluble tau levels. Thus, in a completely unsupervised way, we have reconstructed a parameter that cannot be otherwise measured in-vivo in humans and detected the general mechanistic relationship between neuronal activity, pathological protein accumulation and other cognitive and biophysical indicators in AD.

We have made corresponding clarifications in the Discussion also in response to question 2 above (lines 348-373) Thank you.

Reviewer #2 (Remarks to the Author):

Dear authors,

Thank you for your paper, I hope that you find the following points useful.

The present paper presents a computational model fitted on an individual basis to multi-modal data (gathered and processed by the authors) of subjects in varying stages of Alzheimer's disease. The model in question is a network (connectome) of Wilson-Cowan oscillators and the fitted parameters correspond to each node's/region's excitability. The fitted parameters suggest that A β -positivity concurs with overall hyperexcitability. Linear regression analysis also shows that A β (and A β -tau cross interaction) is significantly correlated with worse cognitive performance per MMSE scores.

The extent of multi-modality in the data used to fit the neural-mass whole-brain models is impressive and, to my knowledge, a novel aspect of the work. However, the use of whole-brain models to infer neuronal parameters from data is not novel, and is, at times, suggested to be novel by the authors (see below). The linear regression analysis showing a positive correlation between A β -induced hyperexcitability and cognitive decline is a novel addition (to my knowledge) to investigating excitability in Alzheimer's with whole-brain modeling. The parameter estimation aspect of the work is not discussed sufficiently for the reader to be able to deem the methodology reliable or not. The E/MEG results are also unclear.

In short, the amount and variety of data collection integrated with whole-brain modeling is impressive, however, the parameter estimation methodology needs to be improved and elaborated. The results are very interesting and will be a great addition to the literature. However, it is my opinion that additional work is needed on behalf of the authors to add sufficient rigor and reproducibility to the parameter estimation for the paper to be publishable.

General response to Reviewer #2:

Thank you for the thorough and helpful comments. We appreciate the reviewer's kind words regarding the relevance of the results, multimodality of the data and the integration with whole-brain modeling. In this revised version, we have provided complimentary information in the Methods, Results and Discussion to address the points raised by the reviewer. For example, we have re-written the subsection Parameter estimation to further describe the optimization process. Other specific details appear in the Supplementary Materials (e.g., parameter values, an additional figure explaining the dynamical system analysis, etc.).

Please, find a detailed point-by-point response in what follows specifying the reasoning and changes in the manuscript.

1. The usage of the term: activity decoder. As far as I understand, the authors are introducing the term activity decoder to refer to inferring neuronal activity by estimating parameters of a computational model. In lines 40-42, it appears to the reader that this is novel. In line 269, the activity decoder is described as a "hybrid approach". In lines 364-366, the impression of the reader is that the activity decoder is a novel methodological contribution that can infer biological parameters. The fitting of computational models to infer real-world parameters is an established methodology in computational neuroscience and the computational modeling of Alzheimer's disease. It is my strong suggestion that this term be avoided. It should be noted, however, that I commend the authors' wish to communicate computational neuroscience to a wider audience.

Response:

We thank the reviewer for observing our focus on biological mechanisms, and for motivating us to further clarify this important methodological aspect.

First, we apologize for the error of lines 40-42. With the goal of avoiding repetition in the sentence, we did not put "AD" or "AD Personalized Pathophysiological" before the term "activity decoder". This was a mistake, producing confusion. We appreciate the suggestion and have corrected the sentence. AD activity decoder is the term that was most used throughout the paper and is really what we intended to communicate. The neural mass model was primed to study pathophysiological activity in AD, by considering the influences of A β and tau, and importantly, their synergistic interaction, to the excitability parameters. Neuronal excitability is not accessible to non-invasive in-vivo measurements and yet deemed mechanistically relevant (Maestú et al., 2021).

In the literature, many parameter estimation approaches exist. Notably, the paper by (Sotero et al., 2009) estimated many parameters from BOLD signals utilizing the robust local linearization technique for nonlinear differential equations. Unlike our approach, with many brain regions, resting-state fMRI and disease information incorporated into the model, Sotero et al dealt with one region, task-related fMRI and was not focused on studying disease. Parameter estimation in such conditions (far fewer differential equations and clear task-related signals) is relatively "easier". A more recent trend is simulating disease-related signals and comparing them to empirical (typically) BOLD functional connectivity to retain the set of parameters that best resembles the real data (e.g. (Deco et al., 2018)). However, generally these approaches do a parameter search -evaluating a cost function at specific points- rather than optimization in a continuous interval. Unless the cost function is monotonical or has a single minimum -which might seldom happen in real data- one risks missing the optimum due to the resolution of the search. Other studies that we discussed in our paper (e.g. refs 25, 28, 10, 22) did simulate AD-related processes in high-dimensional models. However, these are only simulations, not parameter identification from the data. We also appreciate the recommendation to discuss the papers of question 4. One of such studies performs computationally robust parameter

estimation of a whole-brain neural mass model, although the causal role of A β and tau deposits in AD pathophysiological neuronal activity cannot be assessed through the rest of the methodology. Please see the response to question 4 for more details.

Likewise, the term hybrid approach was embedded in a paragraph referring to mechanistic and data driven models. To be precise, our method seeks to unveil disease mechanisms with a model that is informed by real data. We have also made edits to clarify this (please see lines 294-311). The model as is could be equally applied to study other diseases where neuronal excitability is a relevant parameter (by simply considering its relevant disease factors). For this iteration, however, the AD activity decoder includes the most clinically relevant pathophysiological factors that are currently imaged, thus the methodological contribution for the study of AD

2. The term "computational model" and "mathematical model" is absent in the paper although the paper deals with parameter estimation of a computational/mathematical model. I would prefer the content of the paper (parameter estimation of whole-brain model for data) to be clear from the title and abstract. Which model (whole-brain Wilson-Cowan) is used should also be clear from the abstract.

Response:

Thank you for highlighting the need for clarifying the proposed computational model. Based on the feedback and seeking to communicate the methodology and main results through the title, we are now proposing the following "Personalized whole-brain neural mass models reveal combined A β and tau hyperexcitable influences in Alzheimer's disease." We have also made a change to the abstract to respond to this point; it now reads "Our computational model built on whole brain Wilson-Cowan oscillators aims to evaluate the direct impact of toxic protein deposition on neuronal activity". Additionally, the term "computational" has been specified wherever necessary (e.g. lines 240, 419, 457) Other sentences like "Here, we develop a personalized whole-brain neural mass model integrating multilevel, multifactorial AD pathophysiological profiles..." in the last paragraph of the Introduction were not modified as the term neural mass model is already referring to a computational model.

3. Here are three papers that investigate neuronal excitability in Alzheimer's disease. I believe the results and methodology from the reviewed paper should be compared with these three papers, as they use a similar computational model to make statements on neuronal excitability in terms of A β and tau:

Altered excitatory and inhibitory neuronal subpopulation parameters are distinctly associated with tau and A β in Alzheimer's disease

Kamalini G Ranasinghe^{1*}, Parul Verma², Chang Cai², Xihe Xie², Kiwamu Kudo^{2,3}, Xiao Gao², Hannah Lerner¹, Danielle Mizuiri², Amelia Strom¹, Leonardo Iaccarino¹,

Renaud La Joie¹, Bruce L Miller¹, Maria Luisa Gorno-Tempini¹, Katherine P Rankin¹,
William J Jagust⁴, Keith Vossel^{1,5}, Gil D Rabinovici^{1,2}, Ashish Raj²,
Srikantan S Nagarajan²

A multi-scale model explains oscillatory slowing and neuronal hyperactivity in Alzheimer's disease

Christoffer G. Alexandersen¹, Willem de Haan², Christian Bick^{1,3,4,†} and Alain Goriely^{1,†}

Tau Pathology induces Whole-Brain Subcritical Brain Dynamics across the spectrum of Alzheimer's Disease

Xenia Kobeleva^{1,2} Petra Ritter³ Gustavo Patow⁴ Gustavo Deco⁵

Response:

Thank you for bringing our attention to these very relevant and interesting works. These papers fall in different categories that were referenced in our manuscript. For example, the second study ("A multi-scale model...") constitutes a simulation study with A β and tau propagation. Computational approaches reproducing real data features but not inferring underlying biological mechanisms were discussed in the second last paragraph of the introduction. As such, we have commented this work therein. The third publication ("Tau Pathology...") is a poster presentation. From what we understand, simulations of the computational model were compared to empirical functional connectivity as in the paper by (Deco et al., 2018) that we discussed as an important contribution to state-of-the-art whole brain models. The text of the results section suggest that high tau burdens associate with noisier oscillations in a Hopf bifurcation system. We have referenced the poster in the first paragraph of the Discussion that treated similar approaches. The first paper ("Altered excitatory...") estimates parameters of a whole-brain neural mass model and then correlates the results with A β and tau PET SUVrs. This study is ambitious and interesting because performs whole-brain parameter estimation. In fact, it constitutes a new category in our referencing and it has been added to the Discussion (also first paragraph). The main limitation in our humble opinion is that the correlation analysis does not inform about the causal role that A β and tau could have on the neuronal activity. One way to measure that effect is perturbing the relevant biophysical parameters in the neural mass model with the A β and tau accumulation levels, recovering the parameters most likely producing the observed neuronal activity and analyzing the trends in such parameters to unveil disease mechanisms.

4. In Fig. 3, it is not clear what "Neuronal Excitability" is with respect to the fitted parameters. It would be very interesting to see how tau and A β and their interaction themselves affect the subjects, as opposed to their sum. Which is contributing to the hyperexcitability? Is tau inducing hypoexcitability or hyperexcitability? What about their cross-interaction?

Response:

Thank you for this comment and the opportunity to discuss more in-depth the factor-specific contributions. In the Supplementary Material, we had included a figure (Fig. 1) with all the parameters that were estimated. Please, find the figure and caption in that document.

In the cohort of 136 participants, 75 subjects had $\theta A\beta < 0$, 75 $\theta \tau < 0$ and 80 $\theta A\beta \cdot \tau < 0$. Please, also note that negative values of these parameters mean hyperexcitable effects, as explained in the Methods. The effects are more striking when clinical diagnosis and $A\beta/\tau$ status are considered. For example, in the AD group (all $A\beta$ and τ positive), 13 of 16 subjects had estimated $\theta A\beta < 0$, 10 $\theta \tau < 0$ and 9 $\theta A\beta \cdot \tau < 0$.

In this manuscript, the individual factor influences were best studied in terms of their effects on cognition (Table 1), where we observe that the $A\beta$ and $A\beta \cdot \tau$ influences were significant predictors of MMSE and MoCA scores in the cohort ($p < 0.05$). The coefficients of these terms in the fitted linear models were positive in all cases. This means that negative influences of the pathophysiological factors –yielding hyperexcitability– are generally linked to low cognitive scores. This finding also constitutes evidence supporting the discovered increased neuronal excitability with advanced disease stages. In consequent research that is under consideration elsewhere we further delve into the factorial influences and identify the molecular signatures of their specific alterations.

We have further commented Supplementary Fig 1 in the main text, based on the reviewer's concern and this response (lines 134-144). Thank you for the suggestion.

5. In lines 290-291, it is suggested that the present paper is providing in-vivo evidence of hyperactivity. It is my opinion that inferring parameters with a computational model is not an "in-vivo" method but is more aptly thought of as "in-silico". I do not believe the authors meant to say this, but the sentence can be misleading.

Response:

Thank you for the observation. To better clarify the overall idea, we have changed previous lines 290-291 to now say: "Although proxy measurements, post-mortem studies and animal models have suggested neuronal hyperactivity mechanisms in AD, no direct quantification of *in-vivo* neuronal excitability existed thus far in humans" which better communicates the meaning of in vivo/human/quantifying excitability values as opposed to post-mortem, animal model or indirect proxy measurements of neuronal excitability. The following sentence ("In this study, by assuming a toxic protein influence model ($A\beta$, τ , $A\beta \cdot \tau$) we inferred neuronal excitability values...") then continues explaining the contribution of this work.

6. In lines 301-302, the meaning of "biomarker capabilities of our estimated excitability parameters" is not sufficiently clear to me.

Response:

Yes, we appreciate the opportunity to clarify this phrase. The phrase was embedded in a paragraph where we discussed the consistency of the findings. In the sentence where it appeared, we explained how the estimated neuronal excitability values significantly associate with robust plasma markers of AD pathophysiology (blood phosphorylated tau and glial fibrillary acidic protein), that were independently obtained in the cohort. In what follows, we discuss the results of the regression analysis of Table 1, where the $A\beta$ and $A\beta$ *tau subject-specific influences significantly predict the cognitive performance of the participants. We observe the same trend: higher estimated excitability associates with advanced AD states. Thus, we were trying to express that in the absence of some of these particular measurements, inferred excitability values could be equally informative. Further clinical validation experiments are nevertheless required to completely establish these biomarker capabilities. To avoid confusion, we have removed the clause and further explain the potential implications of the results elsewhere.

7. Line 469: Where are the generic Wilson-Cowan parameters taken from?

Response:

The generic parameters were obtained from a set of studies that has utilized the Wilson-Cowan model either in their original one single E-I module or whole-brain simulations. The references were provided in Supplementary Table 6. We thank the reviewer for noticing that the references were not reiterated in the main text in line 469, which has been corrected in the revision (line 559). Additionally, a dynamical system analysis was performed (see answer to question 10) to confirm that these sets of parameters produced oscillatory activity.

8. It should be clear from the Results Section that the excitability in question is the neuronal excitability of excitatory neurons and not inhibitory neurons.

Response:

Thank you for the suggestion. We have made the appropriate changes. Please, see lines 124-125

9. In the Methods Section on the computational model, I recommend presenting the computational model first and then immediately explaining the variables and parameters afterward. This is common practice and is generally deemed to be clearer to read.

Response:

Thank you for this recommendation. All parameter values and their interpretation are provided in Supplementary Tables 6 and 7. The parameters and variables that are relevant to the analyses

presented in this paper, and general presentation of the model, are provided in the main text. There were three main reasons for selecting this writing approach:

- 1) Our group is heterogeneous, consisting of medical practitioners, engineers, biologists, psychologists and computational scientists. When writing was concluded, all authors expressed satisfaction with the level of detail that the text had and the complimentary information that was provided. This could indicate that the findings are accessible to a broader audience.
- 2) Similar practices are found in the computational neuroscience literature, depending on the goals of the paper. For example, one of the papers that is referenced (Deco et al., 2018) omits several parameter values as they are found in previous work by Stephan et al. Additionally, another seminal work, by (Valdes-Sosa et al., 2009) in this case, utilizes the model of Sotero et al for the generation of the BOLD signal (the same one that we utilized in this paper) and the model is presented in Appendix C, not the main text.
- 3) Both the Wilson-Cowan model (Wilson & Cowan, 1972) and the MHM by (Sotero & Trujillo-Barreto, 2007) are highly-regarded models that have been utilized and validated many times, thus we found that definitions of generic parameters in the Supplementary Text were proper for this work (as they will considerably increase the length of the Methods section) while the focus of the main text should be on the integration of the two models, the biophysical parameters that are estimated and the mechanistic relevance of the findings.

Nevertheless, we appreciate the editor and reviewers' opinions based on this explanation of the rationale behind the writing of this part.

10. In line 466, it is unclear what kind of dynamical systems analysis was performed. What is meant by plausible electrophysiological oscillations and BOLD signals? In particular, what frequency are the Wilson-Cowan oscillators running at? Are the WC parameters set globally across all subjects?

Response:

Thank you for the question. We appreciate the opportunity to add to the paper a supplementary figure (#10) that we had produced and where the dynamical system analysis was illustrated and the terms "plausible electrophysiological oscillations and BOLD signals" were better explained.

In a nutshell, a classical bifurcation analysis of the isolated WC module with the parameters of Supplementary Table 6 was performed. The objective was to select a working point for the external inputs to the regions in the connected model. For some inputs, the system does not oscillate (see the Hopf bifurcation points in the figure). The initial conditions (question 13) were also chosen within the darker area in the diagram (stable limit cycle, only E variable shown). Dynamical system analyses are customary in the computational neuroscience literature (see for example (Deco et al., 2009; Spiegler et al., 2010)). Several possible values for the global connectivity parameter existed, nevertheless. Thus, we simulated many whole-brain systems (assuming no $A\beta$ or tau burdens, but with the effective excitatory sigmoidal threshold explored in a broad interval). We noticed that for some combinations of the global coupling parameter and excitatory sigmoidal threshold very regular and non-realistic BOLD signals were produced (see all regional signals in the figure). Finally, we set a global coupling parameter of 2 because for that value we had a large interval where the pathophysiological influence parameters could exist. Similar approaches exist in the literature to select global coupling constants (Daffertshofer & van Wijk, 2011; Stefanovski et al., 2019) The caption of the figure (Supplementary Materials) contains the detailed explanation for the readers.

All parameters but the pathophysiological influence parameters affecting neuronal excitability (parameters to estimate) and the individual data ($A\beta$ and tau PETs and rs-fMRI, the latter used

for the cost function) were equally set across participants. This is also common practice in computational neuroscience (Deco et al., 2018; Valdes-Sosa et al., 2009), especially in disease modeling when it is known or hypothesized how disease affects neuronal activity (i.e., which specific biophysical parameters could be affected). The frequencies of the isolated WC oscillators are given by their time constants, specifically $\tau_I = 0.02$ s and $\tau_E = 0.01$ s (see also Supplementary Table 6)

Additionally, we have acknowledged in the Discussion section (previous lines 294-297, now lines 348-362) that the results are conditioned by the selection of the WC and MHM parameters. Many studies justify the values of these parameters (Supplementary Tables 6-7) and the methodology of the dynamical system analyses that we employed to select the free ones. It is, however, the a-posteriori analyzed differences between subjects and clinical groups in terms of neuronal excitability, with all other parameters kept the same, what makes biological sense and what we tried to interpret in this study.

11. In line 422, it should be made clear in the Results that the connectome used for simulations is averaged across subjects. Particularly, as the word “personalized” model is being used, it might mislead readers into thinking the connectomes are used per individual.

Response:

Thank you for the opportunity to further clarify this aspect. We have specified that the connectivity matrix is average in the legend of Fig 1 in the Results, where the anatomical connectivity was mentioned. Additionally, in Fig 1, the connectome was shown in panel B (the lower part), where the neuro-dynamical model and BOLD transformations -which were also maintained the same for all subjects- are shown (as opposed to part A, where the individual data is shown). We believe that this, together with the previous explanation in the Methods and identification as a possible limitation in the Discussion will make it clear to the readers.

12. With regards to the E/MEG simulations: the Wilson-Cowan model typically operates at a single frequency (PSD will reveal a single peak). It should be clear what frequency an isolated Wilson-Cowan model operates at with the parameters set by the authors. Analyzing the frequency band outside of this frequency does not really make sense (unless it can be demonstrated that the PSD shows multiple peaks). As such, I am not confident in the electrophysiological simulations and the claim that the model captures both phenomena seen in theta and alpha bands before I see a more detailed analysis.

Response:

Thank you for this question. The parameters of the WC oscillators appear in Supplementary Table 6. The time constants were also reiterated in the response to question 10 ($\tau_I = 0.02$ s and

$\tau_E = 0.01$ s). It must be noted that these parameters are kept the same for all regions and subjects. However, in the interconnected model, the excitability of the excitatory populations is allowed to change, based on the local and subject-specific A β and tau pathologies burdens.

Importantly, in the WC formulation, the EEG signals are proportional to the regional excitatory input current (Meijer et al., 2015), which is obtained through the following expression indicating the superposition of all inputs arriving to the excitatory population (region k):

$$x_{E,k} = C_{EE}E_k - C_{IE}I_k + P + \frac{\eta}{N} \sum_{l=1, l \neq k}^N C_{lk}E_l$$

These are the quantities that were used in the analyses leading to Fig 2 (see also Methods, subsections Electrophysiological model and Interpreting the pathophysiological effects on neuronal activity). Exemplary reconstructed cortical sources of resting-state EEG are shown in Fig 1c (the purple signals at the top). The following figure shows the spectra of all regional excitatory input currents reconstructed after identification of the pathophysiological influences for one specific subject in the cohort. Please, note the power in the alpha band, around 10Hz. Also note that a couple of the regions have higher power in the theta band, which creates a more pronounced visual effect, though the alpha content is much less than theta, in general. The other subjects have similar spectra.

Please, also note that the results in Fig. 2 refer to the standardized power in each band, using the mean and s.d. of powers in that band. The figure is not suggesting that the theta and alpha bands of the proxy EEG signals had the same power across the subjects. What the figure is showing is that within one band (either theta or alpha) the estimated power statistically changes with clinical groups.

13. How the computational model is solved is not presented. It would be assuring to see a neural mass signal compared to a BOLD signal (it would also clarify what is meant by setting the parameters such that the signal is comparable to BOLD and MEG signals), to see that the integration has been done successfully. It should also be clear how the initial conditions have been set.

Response:

We appreciate the opportunity to clarify these aspects. From last to first mentioned issue:

- 1) The initial conditions of the Wilson-Cowan system were set as $X_{0I} = 0.01$ and $X_{0E} = 0.075$ for all the neural masses. These are in the basin of attraction of the limit cycle (see also response to question 10) (Spiegler et al., 2010). The initial conditions of the MHM model for the generation of the BOLD signal follow Sotero et al. The initial conditions have been added to Supplementary Tables 6 and 7. Additionally, a transient time of 20s was dropped for every analysis that included the signals (e.g. the maximization of the similarity with the real signals) so that the stable oscillatory behavior was established
- 2) The connectivity scaling parameters of the model were set so that all signals possibly existing in the examined intervals of the unknown parameters were qualitatively similar to real BOLD and E/MEG signals (the quantitative resemblance is established through the optimization algorithm). As it is more deeply explained in the response to question 10, we fixed the connectivity parameters so that the dynamical models produced oscillatory behavior with observed noise/chaos, signal modulation, etc, as in real neurophysiological signals and following the methodology of (Daffertshofer & van Wijk, 2011; Stefanovski et al., 2019).

- 3) Three typical BOLD signals that were reconstructed after parameter optimization are shown in Fig. 1 for the subject whose data served to generate this figure. Each of these segments appear next to the real BOLD signal for the same brain region and subject. The whole signals are close to 10 min of duration which is not shown due to space constraints.

14. Parameter estimation. The objective function used to estimate the parameters using MATLAB's surrogate optimization method is not clear. A mathematical definition of the objective function should be provided.

Response:

Thanks. As mentioned in the Methods, Parameter estimation, the objective function is the correlation distance between the regional, simulated (at each parameter optimization trial) BOLD fALFF values and the real BOLD fALFF values, for each subject. In other words, one is looking for a set of parameters that minimizes the distance -or, maximizes the similarity- between the simulated and real indicators. We have included a mathematical definition to the section Parameter estimation (lines 631-638). We have tried to make this definition as simple and readable as possible, since the notation may be confusing.

15. Parameter estimation: Which parameters are being optimized over, is not entirely clear. Is θ_0 being optimized as well?

Response:

No, θ_0 is not optimized. The set $\theta_j^{A\beta}$, θ_j^{Tau} , $\theta_j^{A\beta \cdot Tau}$ is optimized (please see subsection Parameter Optimization). Here, we assume that θ only changes due to $A\beta$ and τ accumulations. There are, of course, many other factors that will make θ to differ from one region to another. We have acknowledged this in previous lines 307-310 and 343-350, now lines 374-377 and 412-418, in the context of modeling neurodegeneration. The purpose of this work was to establish the $A\beta$ and τ mechanistic effects on neuronal excitability. Thus, only those parameters were estimated.

16. Assuming all θ (4 parameters) are being optimized per subject, it is not trivially true that any optimization method can be successful in finding optimal parameters, and it depends particularly on the objective function. In line 546, the authors say that it is "guaranteed" that an optimal solution is found. No optimization method can guarantee the identification of a global maximum.

Response:

Thank you for this comment. We appreciate the opportunity to further clarify how parameter optimization was performed. The objective function was defined as the correlation distance between the regional, simulated (at each parameter optimization trial) BOLD fALFF values and the real BOLD fALFF values, for each subject (question 14). The objective of the optimization was to make this distance as small as possible. A zero value would mean that when a region has high real fALFF value, so does the simulated signal, and so on, finding a perfect alignment between the two. Since the magnitude of a simulated signal likely won't exactly be the magnitude of the real signal (there are unmodeled factors and unknown specific parameters) we preferred this approach over fitting signal vs signal, using Euclidean distances between fALFFs, etc. It also follows evidence (Yang et al., 2018, 2020)) of specific fALFF increases and decreases in certain areas with disease progression, which would be best captured by correlation distances.

The problem of finding global optima is indeed complex. There exists a vast, yet still incomplete literature addressing the many issues it presents (Arora et al., 1995). We made decisions given the characteristics of our problem and following best practices. For example, surrogate optimization was picked over other available algorithms (e.g. genetic algorithm, pattern search, multistart) as it works best with objective functions that take a long time to evaluate and is suited to deal with bounds, constraints, nonsmooth functions, etc. Additionally, we imposed constraints to the parameters motivated by the dynamical properties of the dynamical system (question 10), which are in turn linked to the biophysical data that was fed to the model. Imposing physical constraints to the optimization problem is generally recommended to increase the chances of obtaining the real parameters, roughly speaking. Finally, instead of performing just one surrogate optimization run with the default number of iterations, we evaluated 20 different random trial point runs (different sequences) and extended each run until they all converged, i.e. a positive stopping flag was returned. This happened in less than 2,000 iterations for all subjects. Then, we selected the parameter set with the lowest objective function value. We observed that several sequences were converging to the same point as well. For 13 subjects with various clinical and pathological statuses (question 19) we performed confirmatory additional surrogate optimization runs (i.e. other random evaluations sequences) in a desktop computer, which yielded the same result than in the computing cluster (the Digital Research Alliance of Canada) where all subjects were calculated due to the high computational requirements of the problem.

There's another important point to consider: all the subjects had the same definition of the cost function, but the optimizations only depend on the individual real data, i.e. A β -PET, tau-PET and rs-fMRI since all other parameters and conditions are kept the same. Thus, once the optimization algorithm has found the most likely parameter set (with the lowest value of the cost function) and we put all the subjects in perspective (clinical groups, A β and tau status, etc) we observe different biological trends. For the purpose of this work, it is the mechanistic differences between groups what's relevant, rather than the individual estimated factor-specific contributions. Nevertheless, such individual values must be taken into account for designing

effective therapeutic interventions, as we highlight in lines 79-81 and 401-403. Please, also see the re-writing of subsection Methods, Parameter estimation addressing all the parameter optimization-related questions and the new elements added to the Discussion (lines 631-652 and 419-442) to respond to the reviewer's concerns.

17. Previous work in inferring parameters of whole-brain models, typically will maximize the Pearson correlation between simulated and experimental fMRI resting-state functional connectivity. It is established in the literature that we should expect Pearson correlations of about 0.4. Without a discussion of how these optimization methods compare with previous work is it difficult to ascertain whether the global optimization methodology is trustworthy or not. (of course, MATLAB's surrogate method works fine, but the question is whether the particular optimization problem can be solved to a satisfactory degree, and this has to be demonstrated, as it is not trivially true).

Response:

Thank you for bringing up this important point. The design of the cost function was a decision to which we devoted a rather considerable part of the work. Ultimately, utilizing fALFFs markers was motivated by its simplicity and yet power to effectively discern between disease stages in the AD spectrum (Iturria-Medina et al., 2018; Yang et al., 2018, 2020). AD-characteristic low frequency fluctuation content variations have been observed and validated for many years, virtually since resting-state fMRI research took off. In particular, fractional markers (lff content divided by the whole interval) are more sensitive to disease progression than the band amplitude or any similar construction (Yang et al., 2018; Zou et al., 2008).

The focus of this work is on the biological mechanisms, reporting differences between disease stages, e.g., how advanced stages relate to hyperactivity. We are also advocating for disease-specific models and optimization cost functions. Functional connectivity differences in the default mode network are well-documented in aging, mild cognitive impairment, and AD (Damoiseaux, 2012). This fact will make its use adequate in our research. One important aspect to consider is how functional connectivity is defined and calculated, however. As Vemuri, Jones and Jack discussed in 2012 (Vemuri et al., 2012): anatomical regions of interest may not always correspond to the functional organization of networks. Do we use a functional brain atlas instead of a structural one? In that case, how do we define structural connections between regions, which propagate the A β and tau effects in our model? Even if favoring the suboptimal computation of functional connectivity in a structurally defined brain parcellation, many options exist. Although Pearson correlation is the most popular (and has been traditionally used as one-fits-all tool in whole-brain optimization regardless of its specificity to the disease), other approaches as partial correlation, or covariance matrix graph theoretical analyses are also valid. On top of that, not only the connections could be informative, the network features as well (Vemuri et al., 2012). If we focused on the connections over the structural atlas, there would exist $N_{\text{regions}}*(N_{\text{regions}}-1)/2$ of those (vs N_{regions} features in our current formulation), which

increases the chances of random correlations driving the similarity between the simulated and real signals. If, otherwise, we focused on the whole network, which properties do we select, what's the right number? One may argue that the extensive results in the AD functional connectivity realm are a byproduct of all the available options that functional connectivity has.

To reiterate what was above said, fALFF measures were AD-specific and simple. The average objective function value was 0.56. Since this denotes the correlation distance between the fALFFs generated with the best parameters and the real fALFFs, then the average correlation between the two in our entire cohort was $1 - 0.56 = 0.44$ which is roughly the expected functional connectivity Pearson correlation value mentioned in the question. We have added lines 436-442 to the Discussion to express the rationale for the fALFF-based objective function chosen over functional correlation metrics.

18. After reading the documentation on MATLAB's surrogate method, it is not clear to me what is meant by lines 544-545 that "no new feasible points are found". This should be elaborated, as it seems stopping criteria are usually defined by objective function values and total iterations per MATLAB's documentation.

Response:

Thank you for this observation. We would like to further clarify the stopping criteria. Please, also see the answers to questions 14 and 16 and the corresponding changes we have made to the Methods, Parameter estimation subsection (lines 631-652) to respond to the reviewers' concerns about the optimization algorithm.

We did not modify Matlab's default ObjectiveLimit value of "-Inf". In reality, our objective function has a physical limit of 0 as it corresponds to a distance measure. Surrogate optimization was then tasked to find the best point, ie, $\theta^{A\beta}$, θ^{Tau} , $\theta^{A\beta \cdot Tau}$ that minimized the objective function. However, there were constraints in the optimization problem. Each of the parameters had lower and upper bounds of -0.05 and 0.05. Their combined effect was also bounded to the [-0.05,0.05] interval. This is what Matlab writes as $A \cdot x \leq b$ on their help page. The constraints are given by the physical properties of the problem to be solved: the dynamical system analysis and the A β and Tau PET values being normalized to the [0,1] interval (see the responses to questions 10 and 16 for further details). Thus, what is meant by "feasible points" is points that satisfy these conditions.

Ideally, we would have preferred a "1: The objective function value is less than options.ObjectiveLimit." exit flag, with a zero-ObjectiveLimit. But this is highly unrealistic in any optimization problem other than textbook examples. We did not expect a perfect similarity between an in-silico generated fALFF vector and the subject's real fALFF, partly because there are biophysical processes that are not modeled and because the problem is rather computationally complex. When surrogate optimization stopped because "The number of function evaluations exceeds options.MaxFunctionEvaluations" (exitflag = 0), we simply

extended the surrogate iterations as it was possible that the best point was not yet found. It was "3: Feasible point found. Solver stopped because too few new feasible points were found to continue." the reason why the optimization process finished. This means that the best of all points satisfying the constraints (feasible point) was returned. As previously explained (question 16), we also run several surrogate optimizations with different random evaluation sequences in our quest for the global optimum.

19. It should also be checked whether the surrogate method returns the same parameter values when rerun with different parameter settings (stricter stopping criteria) to demonstrate that it is converging (and converging to the same value per run). It should also be verified that the optimization method identifies the same optimal parameters for different initial conditions. Ideally, this should be demonstrated with a figure or, at least, be discussed in the text.

Response:

Thank you for this comment and allowing us to add further details to the explanation of the optimization process. As mentioned in the response to question 18, the optimization algorithm stopped when a point satisfying the constraints was found and not enough resting feasible points existed to continue evaluating (<https://www.mathworks.com/help/gads/surrogateopt.html>). Before that, as any optimization algorithm does, surrogate optimization keeps track of all the evaluated points and retains the set returning the lowest value of the optimization function. We continued all runs until the algorithm converged with a positive exit flag. Limiting the number of optimization iterations would likely make the algorithm stop before it has reached the likely global minimum. Likewise, it is best to have the objective function limit as the default $-\text{Inf}$ so one guarantees that the method will continue to perform evaluations trying to decrease the value of the objective function, thus approaching the global minimum.

Regarding the initial conditions for the optimization, or random evaluation trials with different seeds in the surrogate optimization algorithm, we assessed several of those sequences to finally retain the one with the lowest objective function value. This procedure has been more detailed in the subsection Parameter estimation now and we believe it constitutes a good practice to increase the chances of finding the absolute global minimum. Additionally, we selected 13 subjects at random and performed another extra set of 20 runs per subject, all different random trial evaluations by surrogate optimization to confirm the results that we were obtaining in the platforms of the Digital Research Alliance of Canada where all the subjects were run given the high time (~1.5 days to complete 2,000 iterations) and memory (10 GB of RAM) computational requirements of the high-dimensional dynamical system.

Regarding the initial conditions of the dynamical system, following best practices in computational neuroscience oscillation models (Sotero & Trujillo-Barreto, 2008; Stefanovski et al., 2019), the initial simulation points were dropped to avoid transient behavior. Additionally,

the cost function is built with indicators emanating from the frequency domain that must be rather robust to changes in initial conditions (please, see (Spiegler et al., 2010) for a discussion on how the frequency of limit cycles only slightly changes at a certain distance from a Hopf bifurcation and the bifurcation diagram of Supplementary Fig 10). Finally, the results that are shown largely refer to the comparison of clinical groups and disease biomarkers with all subjects being run in equal conditions (dynamical model, initial conditions, etc). While it is true that the specific initial conditions are unknown and they were not estimated from the data as in (Sotero et al., 2009), this fact shall not affect the interpretation of the results and the obtained mechanistic insights. We have added the utilization of equal initial conditions to the note in the Discussion section that referred to the constraints of the model (lines 358-362) and also now mentioned the dropping of the transient time in the Methods (line 630). The issues and applied strategies are explained in lines 431-436. Thank you for the suggestion.

Reviewer #3 (Remarks to the Author):

The manuscript by Sanchez-Rodriguez and colleagues proposes a model that links A β -B and tau effects on functional MRI in Alzheimer's disease. The authors use A β -B and tau PET data to inform the a large-scale computational model that generates simulated the fractional amplitudes of low-frequency fluctuations (fALFF) of BOLD signals. The main motivation of the manuscript and the idea of using PET maps to infer the alterations in excitability during AD progression is novel and highly interesting. Nevertheless, the manuscript does not provide convincing evidence that the model offers any meaningful information that explains the observed empirical data.

General response to Reviewer #3:

Thank you for the valuable comments on the manuscript. To clarify the points raised by the reviewer, we edited the first section of the Results, the Methods and Supplementary Information, to provide additional details. A new Discussion paragraph with expanded additional existing information was written to help the readers interpret the results.

Among the new analyses, we provide further evidence to support the mechanistic decoding value of the whole-brain computational models informed by A β and tau PET and rs-fMRI indicators. We computed voxel-based morphometry (VBM) values for all the subjects in the study and ROIs in the parcellation. VBM correctly characterizes between groups' regional volume and tissue concentration differences from structural MRI (Tissot et al., 2021; Wang et al., 2015). Performing analyses similar to those of previous Fig 4, we once again observed increased excitability with disease progression: the lower the average VBM values, the more excitable the populations of the participants were. Note that a low VBM value indicates brain atrophy - ultimate sign and deleterious consequence of AD. The behavior is even more striking when we focus on regions that prominently showcase neurodegeneration in AD as the hippocampus, entorhinal cortex, posterior cingulate gyrus, amygdala, parahippocampal gyrus and fusiform. Taken together, the results in the manuscript support a strong relationship between hyperexcitability and disease progression in AD.

Major comment 1: The most important concern derives from the fact that the authors did not include any analyses regarding the PET SUVR values in the manuscript. It appears in Supplementary Figure 8 that after the normalization the subjects that are classified as A β -positive have substantially higher values than A β -negative ones, as expected. However, without knowing these data-driven results it is not possible to evaluate whether the proposed computational model adds any useful information to the observed differences in the empirical

data. Being more specific it is very likely that the results regarding neuronal excitabilities, theta and alpha power, and plasma tau correlations simply reflect the substantial differences in PET SUVR values. In such case we may not draw any conclusions that the model provides any useful information beyond the data-driven results. It is also clear in Supplementary Figure 8 that the differences in fALFF values, if any, are not as substantial as PET SUVR values. As a result any model or any model parameter would produce similar results to compensate for the differences in PET SUVR values if provided rather than explaining the subtle differences in the observable (i.e. fALFF). To test this the authors might replicate the results with other biophysically-relevant cortical maps or spatial-autocorrelation matched random maps with differences comparable to empirical PET SUVR values. For example, lack of positive results with microglial activation would also be explained simply by not having the same empirical differences in this particular measure. Similarly, other than excitability, any other model parameter such as unspecific local inputs or global coupling would manifest a similar contrast between groups if provided by the average PET SUVR values. Therefore, the only novel information provided by the model is that the changes in excitability simultaneously explain the alterations in theta and alpha power. However, that could be shown by using a generic model without a necessity to fit to the empirical data focusing on additional insights regarding the relationship between excitability and theta/alpha power. Therefore, to support the claims and the conclusions of the manuscript one would expect to see the group differences in raw PET SUVR scores which appears to be the main result of the study and the relationships between these PET SUVRs and fALFF (and preferably functional measure) profiles before and after fitting the model. Then systematically introduce the model that can explain the differences in empirically observed measures (i.e. fALFF and/or functional connectivity), preferably providing mechanistic insights. I believe the authors already have proper data with empirical results that could produce such study.

Response:

Thank you for this comment and the opportunity to further clarify these important aspects. In what follows we provide a dissected response and information about the additional clarifications, edits and new analyses that we performed on the revised manuscript.

We utilized this opportunity to improve the ways in which the group differences in the PET and rs-fMRI data were reported in the paper. Previously, we had performed t-tests for the regional values between the AD and CU groups. The results were provided in Supplementary Table 5 and discussed in previous lines 317-320, highlighting the regions for which significant differences existed. We have now performed statistical analyses similar to those that were carried on to produce Fig 3 (ANCOVA post-hoc t-tests, age and sex as covariates). Supplementary Table 5 was modified accordingly.

Once more, all A β and tau PET SUVRs were different for the AD and CU groups in all regions. The PET SUVRs in the model are baseline values. We do not model changes (before and after fitting) to the PET-measured pathological burden as this study was cross-sectional, not longitudinal. The same goes to the rs-fMRI and their fALFFs. It would have been very interesting

to see the evolution of the estimated pathophysiological influence parameters with imaging times, and model pathology deposition changes too (at a much slower time scale). However, we did not possess more than baseline evaluations of all imaging modalities for a considerable number of subjects.

Regional differences in the data were more subtle for fALFF (12 regions) -sex and age-adjusted (Supplementary Table 5). These observed differences in our data agree with previous findings (Yang et al., 2018, 2020). In the response to Minor Comment 1, we extensively discuss the rationale for selecting fALFF indicators over functional connectivity. The personalized models that we utilized explain the differences in the empirically observed fALFF because they have been fitted to do so. Correlation distances are utilized because not all contributions to the (BOLD) signals can be modeled. For example, the neural mass oscillators have preponderant frequency peaks, and the neuronal activation, when transformed to BOLD signals do not show the full range of biological content (0.01-0.25 Hz) (Sotero & Trujillo-Barreto, 2008). However, a maximized correlation of the spatial distribution of simulated and real fALFF content (0.01-0.08 Hz/the entire biological band) guarantees that the best parameters values were obtained for each subject in the conditions of the experiment. In the now added Supplementary Fig. 2 (please find in the Supplementary Materials) we show the observed and parameter-estimation-reconstructed BOLD indicators for all the subjects in the cohort and regions in the brain parcellation, which follows a linear relationship (this figure has been referenced in lines 140-144).

As we discussed in previous lines 307-327, the inability of models to identify microglial activation neuronal excitability influences is a result of paramount importance. Although a body

of evidence indicates a major role by neuroinflammation in AD and it is expected that chronic inflammation exists towards the end of the AD spectrum (Kwon & Koh, 2020; Pascoal et al., 2021; Shen et al., 2018), PET-tracers are not sensitive to microglial activation types (Shen et al., 2018), which jeopardizes the correct in-vivo characterization of this factor. When such unspecific data – 30 regions exhibiting AD vs CU microglial activation differences as compared to all in A β and tau PET, see also Supplementary Table 5- was entered to the individual models, no excitability differences were detected. This fact highlights the capacity of the model to produce reliable results and mechanistic insights when disease-specific data is inputted. Please, also note that parameter identification is performed individually, with each personalized model being completely blind to the data (PET, rs-fMRI) of the other subjects. As the parameters that are not relevant to neuronal excitability disease-induced changes were kept constant, we were able to statistically investigate mechanistic differences a-posteriori (i.e., after individual parameter identification and reconstruction of underlying neuronal excitability values and signals).

We have expanded the lines that were devoted to this key element of the methodology in the previous version of the manuscript (please see new lines 348-373) and also further explained how the results are in fact conditioned by the distribution of the pathological data that the model receives. This is expected, both through common sense and given the results of previous disease models in the literature ((Deco et al., 2018), for example). It was exactly the A β and tau causal influence what we wanted to test. For this purpose, we introduced pathological perturbations to the parameter that represents the model's biophysical quantity "neuronal excitability". Other parameters would not measure regional-specific firing properties. The personalized model, when analyzed in perspective, confirm several observations in animal models, and some indirect evidence stemming from post-mortem and computational studies in humans. Neuronal excitability does not stay invulnerable to A β and tau separate and synergistic effects as their pathological burden increases, neither it decreases with AD progression (which could have been possible in a binary A β + or A β - distribution of data, for example). Our results strongly suggest that neuronal excitability increases, on average, with the disease, and this is the first time, to our knowledge that such conclusion can be drawn in in-vivo data-informed human studies.

Importantly, this version of the manuscript contains additional evidence (Results, subsection Neuronal hyperexcitability relates to high levels of plasma AD biomarkers and grey matter atrophy) supporting the mechanistic relevance of neuronal excitability increases with disease progression. It is known that AD produces accelerated brain atrophy. Various hypotheses suggest that this occurs because of toxic A β and/or tau accumulation (Hampel et al., 2021; Tissot et al., 2021). In any case, atrophy, particularly at regions as the fusiform, parahippocampal gyrus, hippocampus, entorhinal cortex, amygdala and posterior cingulate gyrus, constitutes an ultimate marker of the disease (Wang et al., 2015). Thus, we calculated voxel-based morphometry atrophy measures for all the regions and subjects and examined the relationship between these values and the estimated neuronal excitabilities. It is important to highlight that structural MRIs were not used in any step of our models' individualized optimization. The results starkly indicate

that neuronal excitability increases with decreased grey matter volume (please, see Fig. 4 and Supplementary Table 3), particularly in the above-mentioned brain areas. Biologically speaking, this finding may suggest that the human brain tries to maintain certain functional normalcy (by increasing firing of its existing neuronal populations) to withstand the effects of the disease (the loss of neurons). Taken together, the characterization of the relationships between the estimated neuronal excitability values in the cohort and the participants' cognitive scores, plasma biomarkers and grey matter volumes, in addition to the observed trends in the reconstructed electroencephalographic signals, indicate the emergence of neuronal excitability increases with disease progression as an important mechanistic event. These results also reaffirm the model's capacity to uncover key hidden elements in the pathogenesis of neurodegenerative disorders.

Major Comment 2: Similar to the first point, the results in figure 4 seems to show the group differences in PET SUVR values rather than excitability. In other words, the use of correlation coefficient to describe the relationship between excitability/SUVR and plasma AD biomarkers is misleading because the main effect is driven by the group differences. Nevertheless, the figure also suggests a positive relationship with AD biomarkers in the hyper-excitable subjects. The authors could demonstrate these relationships using a more adequate analyses.

Response:

Thank you for the question and allowing us to clarify what the objective of the analyses in Fig. 4 was.

AD is currently regarded as a complex and multifactorial disorder. Decades of research trying to identify a single triggering factor for its associated neurodegeneration haven't provided a clear answer. Except for dominantly inherited AD, which accounts for less than 5% of the cases (Calabrò et al., 2021), the subject-specific cascade of events may arise from different causes and follow different degeneration trajectories (Iturria-Medina et al., 2018). However, previous research has shown that in AD (or pathologically advanced MCI) all the following effects coexist (Ashton et al., 2021; Babiloni et al., 2013; Jack et al., 2018; Ossenkoppele et al., 2022; Wang et al., 2015; Yang et al., 2018):

- elevated misfolded protein accumulation (evaluated through PET),
- abnormal plasma phosphorylated tau marker values,
- grey matter atrophy (evaluated through MRI voxel-based morphometry or neuropathological evaluations),
- affected brain function (measured via markers as the low-frequency fluctuations content of the resting-state fMRI signal and alpha or theta EEG contents), and
- cognitive deterioration (measured with MMSE or MoCA scores)

In-vivo human neuronal excitability cannot be measured non-invasively but indirect measurements, post-mortem and animal experiments suggested that changes occurred with the disease and that A β and tau are involved in it (Busche et al., 2019; Busche & Hyman, 2020; Celone et al., 2006; Lauterborn et al., 2021; Tok et al., 2022; Vessel et al., 2017). That's exactly what we wanted to test in this work.

In a model that receives A β and tau accumulation levels as the only possible cause of neuronal excitability perturbations, a neuronal excitability distribution somewhat aligning with the PET distribution was expected. This was statistically tested in subsection "The estimated neuronal excitabilities increase with clinical states and disease progression" and it is shown in Fig. 3 and Supplementary Fig 3.

In the analyses leading to Fig 4 -now modified to additionally show the relationships with grey matter atrophy: the more the atrophy, the more the firing at AD characteristically affected brain areas- we did not seek to repeat the A β and tau PET information. As previously mentioned, the A β and tau information is embedded in the obtained subject-specific neuronal excitability values (Supplementary Fig 4). Also, A β and tau PET are important components of AD diagnosis, but as it was above highlighted, it is very likely that A β and tau positive subjects also present cognitive and functional decay, brain atrophy, etc.

The purpose of the analyses yielding Fig 4 was to determine if the estimated neuronal excitability values associated with AD biomarkers that were not included in our pathological

affectation model (equation 1). This analysis was performed to confirm that the estimated neuronal excitabilities correctly captured the disease's features. The answer is: yes. All evidence suggests that the in-vivo AD human brain increases neuronal excitability (overall and in the most prominently affected areas) in the presence of A β and tau pathology. Likewise, the results of the following subsection ("Synergistic A β and tau interactions strongly relate to cognitive performance"), establish the relationship with cognitive scores (in that case, the analysis is performed in a factor by factor way, underscoring the contributions of A β and A β and tau's synergistic interaction term to cognitive deterioration).

We have modified the text of the subsection "Neuronal hyperexcitability relates to high levels of plasma AD biomarkers and grey matter atrophy" to better state its specific goal (lines 236-242). Additionally, we have commented the new atrophy results in the Discussion (lines 337-343) and added a paragraph to further explain to the readers that the separation according to A β and tau-positivity groups is obvious and the relationship with AD biomarkers in the direction "more AD, more hyperexcitation" is the significant result (lines 348-373).

Minor Comment 1: The rationale behind choosing fALFF to fit the model is not clear in the manuscript. In methods section authors claim that this indicator has shown high sensibility to disease progression. However, there is a much extensive literature on functional connectivity related to the disease progression. Furthermore, there might be differences in functional and fALFF measures regarding their involvement in AD as well as the model dynamics. Therefore authors should at least elaborate their rationale to choose this specific measure.

Response:

Thank you for bringing up this important point. The design of the cost function was a decision to which we devoted a rather considerable part of the work. Ultimately, utilizing fALFFs markers was motivated by its simplicity and yet power to effectively discern between disease stages in the AD spectrum (Iturria-Medina et al., 2018; Yang et al., 2018, 2020). AD-characteristic low frequency fluctuation (Lff) content variations have been observed and validated for many years, virtually since resting-state fMRI research took off. In particular, fractional markers (Lff content divided by the whole interval) are more sensitive to disease progression than the band amplitude or any similar construction (Yang et al., 2018; Zou et al., 2008).

The focus of this work is on the biological mechanisms, reporting differences between disease stages, e.g., how advanced stages relate to hyperactivity. We are also advocating for disease-specific models and optimization cost functions. Functional connectivity differences in the default mode network are well-documented in aging, mild cognitive impairment, and AD (Damoiseaux, 2012). This fact will make its use adequate in our research. One important aspect to consider is how functional connectivity is defined and calculated, however. As Vemuri, Jones and Jack discussed in 2012 (Vemuri et al., 2012): anatomical regions of interest may not always correspond to the functional organization of networks. Do we use a functional brain atlas

instead of a structural one? In that case, how do we define structural connections between regions, which propagate the A β and tau effects in our model? Even if favoring the suboptimal computation of functional connectivity in a structurally defined brain parcellation, many options exist. Although Pearson correlation is the most popular (and has been traditionally used as one-fits-all tool in whole-brain optimization regardless of its specificity to the disease), other approaches as partial correlation, or covariance matrix graph theoretical analyses are also valid. On top of that, not only the connections could be informative, the network features as well (Vemuri et al., 2012). If we focused on the connections over the structural atlas, there would exist $N_{\text{regions}}*(N_{\text{regions}}-1)/2$ of those (vs N_{regions} features in our current formulation), which increases the chances of random correlations driving the similarity between the simulated and real signals. If, otherwise, we focused on the whole network, which properties do we select, what's the right number? One may argue that the extensive results in the AD functional connectivity realm are a byproduct of all the available options that functional connectivity has.

To reiterate what was above said, fALFF measures were AD-specific and simple. We have added lines 436-442 to the Discussion to express the rationale for the objective function chosen over functional correlation metrics.

Minor Comment 2: The main text lacks a lot of relevant details to understand and interpret the results such as which parcellation is used, which computational model is used, how the model fit to the data, why fALFF is chosen...etc. Most of the information was found in methods section. However, as a convention it is better to very briefly introduce these details in the main text that are crucial to understand and interpret the results. Moreover, the manuscript lacks any details and results regarding the model fitting procedure (such as the model fit values, number of iterations, whether there were random initializations...etc), which makes it difficult to evaluate the results.

Response:

Thank you for the comment. We have made appropriate changes to further clarify these aspects.

Regarding the first part of the question, we have modified the Results subsection Modeling pathophysiological impacts on whole-brain neuronal activity to state the specific computational model, parcellation and briefly explain model optimization, with the fALFF-based cost function (see also the response to Minor Comment 1 above). Whenever appropriate, the reader is directed to the Methods section for further details. The changes are highlighted and appear over lines 115-133. Thank you

The details of the model fitting (second part of the question) were expanded over what appeared in the Methods and Supplementary Information. We have re-written the subsection Parameter Estimation to further explain the procedure (lines 631-652). Please, find the changes highlighted. The Supplementary Pseudocode was also further edited. In a nutshell, we ran 20 different random trials evaluations for each subject and extended each run until they all

converged, i.e. a positive stopping flag was returned. This happened in less than 2,000 iterations, for all subjects. Then, we selected the parameter set with the lowest objective function value. We observed that several sequences were converging to the same point as well. For several subjects with various clinical and pathological statuses we performed confirmatory additional surrogate optimization runs (i.e. other random evaluations sequences) in a desktop computer, which yielded the same result than in the computing cluster (the Digital Research Alliance of Canada) where all subjects were calculated due to the high computational requirements of the problem.

We adopted recommended model optimization strategies with the purpose of robustly fit our highly complex biological system including 1) surrogate optimization was picked over other available algorithms (e.g. genetic algorithm, pattern search, multistart) as it works best with time-consuming objective functions. 2) imposing physical constraints to the parameters motivated by the dynamical properties of the dynamical system (e.g. restraining the parameter search range to that where oscillatory behavior exists, see also Supplementary Fig 10), 3) evaluating multiple runs of the optimization problem with random initializations. We have commented issues associated with parameter optimization and the adopted solutions in the Discussion (lines 419-442). Additionally, we expanded on the selection of the cost function (see also the responses to Major Comment 1 and Minor Comment 1 above and the revised subsection Parameter estimation). The average correlation between the fALFFs generated with the best parameters and the real fALFFs across the participants was 0.44 (see also lines 140-144). Thank you

References

- Arora, J. S., Elwakeil, O. A., Chahande, A. I., & Hsieh, C. C. (1995). Global optimization methods for engineering applications: a review. In *Structural Optimization* (Vol. 9). Springer-Verlag. <https://link.springer.com/article/10.1007/BF01743964>
- Ashton, N. J., Karikari, T. K., Rodriguez, J. L., Benedet, A. L., Snellman, A., Pascoal, T. A., Gauthier, S., Rosa-Neto, P., Jack, C. R., Petersen, R. C., Mielke, M. M., Chatterjee, P., Martins, R. N., Thambisetty, M., Varma, V. R., Resnick, S. M., Fox, N. C., O'Connor, A., Vrillon, A., ... Blennow, K. (2021). Plasma p-tau₂₃₁ in the Alzheimer's disease continuum: A multi-cohort evaluation of diagnostic performance, detection of A β pathology and preclinical application. *Alzheimer's & Dementia*, *17*(S5). <https://doi.org/10.1002/alz.056186>
- Babiloni, C., Lizio, R., Del Percio, C., Marzano, N., Soricelli, A., Salvatore, E., Ferri, R., Cosentino, F. I. I., Tedeschi, G., Montella, P., Marino, S., De Salvo, S., Rodriguez, G., Nobili, F., Vernieri, F., Ursini, F., Mundi, C., Richardson, J. C., Frisoni, G. B., & Rossini, P. M. (2013). Cortical Sources of Resting State EEG Rhythms are Sensitive to the Progression of Early Stage Alzheimer's Disease. *Journal of Alzheimer's Disease*, *34*(4), 1015–1035. <https://doi.org/10.3233/JAD-121750>
- Bero, A. W., Yan, P., Roh, J. H., Cirrito, J. R., Stewart, F. R., Raichle, M. E., Lee, J. M., & Holtzman, D. M. (2011). Neuronal activity regulates the regional vulnerability to amyloid- β 2 deposition. *Nature Neuroscience*, *14*(6), 750–756. <https://doi.org/10.1038/nn.2801>
- Busche, M. A., Chen, X., Henning, H. A., Reichwald, J., Staufenbiel, M., Sakmann, B., & Konnerth, A. (2012). Critical role of soluble amyloid- β for early hippocampal hyperactivity in a mouse model of Alzheimer's disease. *Proceedings of the National Academy of Sciences of the United States of America*, *109*(22), 8740–8745. <https://doi.org/10.1073/pnas.1206171109>
- Busche, M. A., & Hyman, B. T. (2020). Synergy between amyloid- β and tau in Alzheimer's disease. *Nature Neuroscience*, *23*(10), 1183–1193. <https://doi.org/10.1038/s41593-020-0687-6>
- Busche, M. A., Wegmann, S., Dujardin, S., Commins, C., Schiantarelli, J., Klickstein, N., Kamath, T. V., Carlson, G. A., Nelken, I., & Hyman, B. T. (2019). Tau impairs neural circuits, dominating amyloid- β effects, in Alzheimer models in vivo. *Nature Neuroscience*, *22*(1), 57–64. <https://doi.org/10.1038/s41593-018-0289-8>
- Calabrò, M., Rinaldi, C., Santoro, G., & Crisafulli, C. (2021). The biological pathways of Alzheimer disease: a review. *AIMS Neuroscience*, *8*(1), 86–132. <https://doi.org/10.3934/Neuroscience.2021005>
- Celone, K. A., Calhoun, V. D., Dickerson, B. C., Atri, A., Chua, E. F., Miller, S. L., DePeau, K., Rentz, D. M., Selkoe, D. J., Blacker, D., Albert, M. S., & Sperling, R. A. (2006). Alterations in memory networks in mild cognitive impairment and Alzheimer's disease: An independent

component analysis. *Journal of Neuroscience*, 26(40), 10222–10231.
<https://doi.org/10.1523/JNEUROSCI.2250-06.2006>

Daffertshofer, A., & van Wijk, B. C. M. (2011). On the Influence of Amplitude on the Connectivity between Phases. *Frontiers in Neuroinformatics*, 5(July), 6.
<https://doi.org/10.3389/fninf.2011.00006>

Damoiseaux, J. S. (2012). Resting-state fMRI as a biomarker for Alzheimer's disease. *Alzheimer's Research and Therapy*, 4(3). <https://doi.org/10.1186/alzrt106>

Deco, G., Cruzat, J., Cabral, J., Knudsen, G. M., Carhart-Harris, R. L., Whybrow, P. C., Logothetis, N. K., & Kringelbach, M. L. (2018). Whole-Brain Multimodal Neuroimaging Model Using Serotonin Receptor Maps Explains Non-linear Functional Effects of LSD. *Current Biology*, 28(19), 3065-3074.e6. <https://doi.org/10.1016/j.cub.2018.07.083>

Deco, G., Jirsa, V., McIntosh, A. R., Sporns, O., & Ko, R. (2009). Key role of coupling, delay, and noise in resting brain fluctuations. *PNAS*, 106(25). <https://doi.org/10.1073/pnas.0901831106>

Eeckman, F. H., & Freeman, W. J. (1991). Asymmetric sigmoid non-linearity in the rat olfactory system. *Brain Research*, 557(1–2), 13–21. [https://doi.org/10.1016/0006-8993\(91\)90110-H](https://doi.org/10.1016/0006-8993(91)90110-H)

Ermentrout, B. (1998). Neural networks as spatio-temporal pattern-forming systems. *Reports on Progress in Physics*, 61(4), 353–430. <https://doi.org/10.1088/0034-4885/61/4/002>

Gignac, G. E., & Szodorai, E. T. (2016). Effect size guidelines for individual differences researchers. *Personality and Individual Differences*, 102, 74–78.
<https://doi.org/10.1016/j.paid.2016.06.069>

Hempel, H., Hardy, J., Blennow, K., Chen, C., Perry, G., Kim, S. H., Villemagne, V. L., Aisen, P., Vendruscolo, M., Iwatsubo, T., Masters, C. L., Cho, M., Lannfelt, L., Cummings, J. L., & Vergallo, A. (2021). The Amyloid- β Pathway in Alzheimer's Disease. In *Molecular Psychiatry* (Vol. 26, Issue 10, pp. 5481–5503). Springer Nature. <https://doi.org/10.1038/s41380-021-01249-0>

Insel, P. S., Mormino, E. C., Aisen, P. S., Thompson, W. K., & Donohue, M. C. (2020). Neuroanatomical spread of amyloid β and tau in Alzheimer's disease: implications for primary prevention. *Brain Communications*, 2(1), 1–11.
<https://doi.org/10.1093/braincomms/fcaa007>

Iturria-Medina, Y., Carbonell, F. M., & Evans, A. C. (2018). Multimodal imaging-based therapeutic fingerprints for optimizing personalized interventions: Application to neurodegeneration. *NeuroImage*, 179(May), 40–50. <https://doi.org/10.1016/j.neuroimage.2018.06.028>

Jack, C. R., Bennett, D. A., Blennow, K., Carrillo, M. C., Dunn, B., Haeberlein, S. B., Holtzman, D. M., Jagust, W., Jessen, F., Karlawish, J., Liu, E., Molinuevo, J. L., Montine, T., Phelps, C., Rankin, K. P., Rowe, C. C., Scheltens, P., Siemers, E., Snyder, H. M., ... Silverberg, N. (2018). NIA-AA

Research Framework: Toward a biological definition of Alzheimer's disease. In *Alzheimer's and Dementia* (Vol. 14, Issue 4, pp. 535–562). Elsevier Inc.
<https://doi.org/10.1016/j.jalz.2018.02.018>

Kwon, H. S., & Koh, S. H. (2020). Neuroinflammation in neurodegenerative disorders: the roles of microglia and astrocytes. In *Translational Neurodegeneration* (Vol. 9, Issue 1). BioMed Central Ltd. <https://doi.org/10.1186/s40035-020-00221-2>

Lauterborn, J. C., Scaduto, P., Cox, C. D., Schulmann, A., Lynch, G., Gall, C. M., Keene, C. D., & Limon, A. (2021). Increased excitatory to inhibitory synaptic ratio in parietal cortex samples from individuals with Alzheimer's disease. *Nature Communications*, *12*(1), 2603.
<https://doi.org/10.1038/s41467-021-22742-8>

Maestú, F., de Haan, W., Busche, M. A., & DeFelipe, J. (2021). Neuronal Excitation/Inhibition imbalance: a core element of a translational perspective on Alzheimer pathophysiology. *Ageing Research Reviews*, *69*, 101372. <https://doi.org/10.1016/j.arr.2021.101372>

Meijer, H. G. E., Eissa, T. L., Kiewiet, B., Neuman, J. F., Schevon, C. A., Emerson, R. G., Goodman, R. R., McKhann, G. M., Marcuccilli, C. J., Tryba, A. K., Cowan, J. D., van Gils, S. A., & van Drongelen, W. (2015). Modeling focal epileptic activity in the Wilson-cowan model with depolarization block. *Journal of Mathematical Neuroscience*, *5*, 7.
<https://doi.org/10.1186/s13408-015-0019-4>

Ossenkoppele, R., Pichet Binette, A., Groot, C., Smith, R., Strandberg, O., Palmqvist, S., Stomrud, E., Tideman, P., Ohlsson, T., Jögi, J., Johnson, K., Sperling, R., Dore, V., Masters, C. L., Rowe, C., Visser, D., van Berckel, B. N. M., van der Flier, W. M., Baker, S., ... Hansson, O. (2022). Amyloid and tau PET-positive cognitively unimpaired individuals are at high risk for future cognitive decline. *Nature Medicine*, *28*(11), 2381–2387. <https://doi.org/10.1038/s41591-022-02049-x>

Pascoal, T. A., Benedet, A. L., Ashton, N. J., Kang, M. S., Therriault, J., Chamoun, M., Savard, M., Lussier, F. Z., Tissot, C., Karikari, T. K., Ottoy, J., Mathotaarachchi, S., Stevenson, J., Massarweh, G., Schöll, M., de Leon, M. J., Soucy, J. P., Edison, P., Blennow, K., ... Rosa-Neto, P. (2021). Microglial activation and tau propagate jointly across Braak stages. *Nature Medicine*, *27*(9), 1592–1599. <https://doi.org/10.1038/s41591-021-01456-w>

Rodriguez, G. A., Barrett, G. M., Duff, K. E., & Hussaini, S. A. (2020). Chemogenetic attenuation of neuronal activity in the entorhinal cortex reduces A β and tau pathology in the hippocampus. *PLoS Biology*, *18*(8). <https://doi.org/10.1371/JOURNAL.PBIO.3000851>

Serrano-Pozo, A., Frosch, M. P., Masliah, E., & Hyman, B. T. (2011). Neuropathological alterations in Alzheimer disease. *Cold Spring Harbor Perspectives in Medicine*, *1*(1), 1–24.
<https://doi.org/10.1101/cshperspect.a006189>

- Shen, Z., Bao, X., & Wang, R. (2018). Clinical PET imaging of microglial activation: Implications for microglial therapeutics in Alzheimer's disease. In *Frontiers in Aging Neuroscience* (Vol. 10, Issue OCT). Frontiers Media S.A. <https://doi.org/10.3389/fnagi.2018.00314>
- Shrestha, N. (2020). Detecting Multicollinearity in Regression Analysis. *American Journal of Applied Mathematics and Statistics*, 8(2), 39–42. <https://doi.org/10.12691/ajams-8-2-1>
- Sotero, R. C., & Trujillo-Barreto, N. J. (2007). Modelling the role of excitatory and inhibitory neuronal activity in the generation of the BOLD signal. *NeuroImage*, 35(1), 149–165. <https://doi.org/10.1016/j.neuroimage.2006.10.027>
- Sotero, R. C., & Trujillo-Barreto, N. J. (2008). Biophysical model for integrating neuronal activity, EEG, fMRI and metabolism. *NeuroImage*, 39, 290–309. <https://doi.org/10.1016/j.neuroimage.2007.08.001>
- Sotero, R. C., Trujillo-Barreto, N. J., Jiménez, J. C., Carbonell, F., & Rodríguez-Rojas, R. (2009). Identification and comparison of stochastic metabolic/hemodynamic models (sMHM) for the generation of the BOLD signal. *Journal of Computational Neuroscience*, 26(2), 251–269. <https://doi.org/10.1007/s10827-008-0109-3>
- Spiegler, A., Kiebel, S. J., Atay, F. M., & Knösche, T. R. (2010). Bifurcation analysis of neural mass models: Impact of extrinsic inputs and dendritic time constants. *NeuroImage*, 52(3), 1041–1058. <https://doi.org/10.1016/j.neuroimage.2009.12.081>
- Stefanovski, L., Triebkorn, P., Spiegler, A., Diaz-Cortes, M. A., Solodkin, A., Jirsa, V., McIntosh, A. R., & Ritter, P. (2019). Linking Molecular Pathways and Large-Scale Computational Modeling to Assess Candidate Disease Mechanisms and Pharmacodynamics in Alzheimer's Disease. *Frontiers in Computational Neuroscience*, 13(August), 1–27. <https://doi.org/10.3389/fncom.2019.00054>
- Tissot, C., L. Benedet, A., Therriault, J., Pascoal, T. A., Lussier, F. Z., Saha-Chaudhuri, P., Chamoun, M., Savard, M., Mathotaarachchi, S. S., Bezgin, G., Wang, Y. T., Fernandez Arias, J., Rodriguez, J. L., Snellman, A., Ashton, N. J., Karikari, T. K., Blennow, K., Zetterberg, H., De Villiers-Sidani, E., ... Rosa-Neto, P. (2021). Plasma pTau181 predicts cortical brain atrophy in aging and Alzheimer's disease. *Alzheimer's Research and Therapy*, 13(1). <https://doi.org/10.1186/s13195-021-00802-x>
- Tok, S., Maurin, H., Delay, C., Crauwels, D., Manyakov, N. V., Van Der Elst, W., Moechars, D., & Drinkenburg, W. H. I. M. (2022). Pathological and neurophysiological outcomes of seeding human-derived tau pathology in the APP-KI NL-G-F and NL-NL mouse models of Alzheimer's Disease. *Acta Neuropathologica Communications*, 10(1). <https://doi.org/10.1186/s40478-022-01393-w>

- Valdes-Sosa, P. A., Sanchez-Bornot, J. M., Sotero, R. C., Iturria-Medina, Y., Aleman-Gomez, Y., Bosch-Bayard, J., Carbonell, F., & Ozaki, T. (2009). Model driven EEG/fMRI fusion of brain oscillations. *Human Brain Mapping, 30*(9), 2701–2721. <https://doi.org/10.1002/hbm.20704>
- Vemuri, P., Jones, D. T., & Jack, C. R. (2012). Resting state functional MRI in Alzheimer's Disease. *Alzheimer's Research & Therapy, 4*(1), 2. <https://doi.org/10.1186/alzrt100>
- Vossel, K. A., Tartaglia, M. C., Nygaard, H. B., Zeman, A. Z., & Miller, B. L. (2017). Epileptic activity in Alzheimer's disease: causes and clinical relevance. *The Lancet Neurology, 16*(4), 311–322. [https://doi.org/10.1016/S1474-4422\(17\)30044-3](https://doi.org/10.1016/S1474-4422(17)30044-3)
- Wang, W. Y., Yu, J. T., Liu, Y., Yin, R. H., Wang, H. F., Wang, J., Tan, L., Radua, J., & Tan, L. (2015). Voxel-based meta-analysis of grey matter changes in Alzheimer's disease. *Translational Neurodegeneration, 4*(1). <https://doi.org/10.1186/s40035-015-0027-z>
- Wilson, H. R., & Cowan, J. D. (1972). Excitatory and inhibitory interactions in localized populations of model neurons. *Biophysical Journal, 12*(1), 1–24. [https://doi.org/10.1016/S0006-3495\(72\)86068-5](https://doi.org/10.1016/S0006-3495(72)86068-5)
- Yang, L., Yan, Y., Li, Y., Hu, X., Lu, J., Chan, P., Yan, T., & Han, Y. (2020). Frequency-dependent changes in fractional amplitude of low-frequency oscillations in Alzheimer's disease: a resting-state fMRI study. *Brain Imaging and Behavior, 14*(6), 2187–2201. <https://doi.org/10.1007/s11682-019-00169-6>
- Yang, L., Yan, Y., Wang, Y., Hu, X., Lu, J., Chan, P., Yan, T., & Han, Y. (2018). Gradual Disturbances of the Amplitude of Low-Frequency Fluctuations (ALFF) and Fractional ALFF in Alzheimer Spectrum. *Frontiers in Neuroscience, 12*(December), 1–16. <https://doi.org/10.3389/fnins.2018.00975>
- Zou, Q., Zhu, C., Yang, Y., Zuo, X., Long, X., Cao, Q.-J., Wang, Y.-F., & Zang, Y.-F. (2008). An improved approach to detection of amplitude of low-frequency fluctuation (ALFF) for resting-state fMRI: Fractional ALFF. *Journal of Neuroscience Methods, 172*(1), 137–141. <https://doi.org/10.1016/j.jneumeth.2008.04.012>

Reviewers' comments:

Reviewer #1 (Remarks to the Author):

The authors have thoroughly revised the manuscript and addressed all major concerns. I have no further suggestions for improvement.

Reviewer #2 (Remarks to the Author):

Thank you for your thoughtful and thorough replies

I am overall very happy with your paper, congratulations!

I have only one final (and very short) request to the authors:

The bounds of the fitted parameters in the objective function are $[-0.05, 0.05]$. It is, of course, possible that new global minima for these parameters exist in the neighborhood, say, $[-0.07, 0.07]$. The new optima may change sign even.

This has to be brought up in the discussion. The reason for choosing these intervals and the limitation of setting these intervals should be remarked upon in the Discussion. I am aware that this is mentioned in the Methods section, but this is important enough to be included.

Here are other minor comments that you can consider:

#1

I have to admit I am still taken aback by calling the method "AD activity decoder". I do not see how this method is conceptually different than the fitting done in, for example, Ranasinghe et al. 2022. The authors argue that because (and please correct me if I am wrong) θ_0 in the objective function is derived from data the paper herein finds a more causal link than compared to just fitting without θ_0 .

What the authors herein have done is, of course, an improvement to Ranasinghe, but giving the method a name such as "activity decoder" is unnecessary and I think the authors should omit it from the manuscript. What the authors have is a fantastic data fitting of Alzheimer's patients, but it is not a novel methodology in and of itself and does not require a name.

I hope the authors understand that I only have good intentions here: I sincerely think the paper would be easier to read and understand if the "activity decoder" term is left unused.

The authors say

"However, generally these approaches do a parameter search -evaluating a cost function at specific points- rather than optimization in a continuous interval"
and state that such grid searches are not robust as optima can hide between evaluations (resolution

too low). This is not correct when done properly. Grid searches are better and much more robust in general than global optimization methods, it is just that they are too computationally expensive for $n > 2$, typically. The problem with global optimization is you have no way to verify whether you're at a global optima. With grid searches, you can verify this for an interval of parameter values, which is why it is preferred for low-dimensional problems.

#3

I did not mean for you to cite a poster. I was confused, as I had seen a seminar about the work and mistakenly thought it was published. I am sorry about this, as I think it is unfair (and must seem strange) to suggest adding a poster to the literature list.

The paper I was thinking about has been published now if you are interested in reading (I'm not asking you to cite it, just FYI)

<https://link.springer.com/article/10.1186/s13195-023-01349-9>

Also, one paper and one preprint on fitting whole-brain models to Alzheimer's data for your consideration

<https://www.frontiersin.org/articles/10.3389/fnagi.2023.1204134/full>

<https://www.medrxiv.org/content/10.1101/2023.01.11.23284438v1>

Reviewer #3 (Remarks to the Author):

I thank Sanchez-Rodriguez and colleagues for addressing the concerns and substantially improving the manuscript after the revision. The previous version of the manuscript lacked details about the model and it was not clear how the model offered any evidence to back the conclusions up by the authors. It is clear that both the rebuttal and the newer version of the manuscript provide sufficient details of the model and they added new analyses and figures to describe the model. They have also added more rigor into the statistical analyses in the new version. The study relies on very nice data and it seems like there have been rigorous work done. Nevertheless, the rebuttal and the new version do not answer the most critical concerns, especially on establishing the causal relationships and evidence to support the validity of the model. A substantial part of the rebuttal refers to the literature instead of addressing these issues in the manuscript. Indeed, as more information is provided regarding the model, more concerns have been raised in the revised version of the manuscript.

Major Comment 1: The main hypothesis of the paper and how it has been shown is still unclear given the results. If the main hypothesis is to the causal relationship between $A\beta$, tau and excitability, then it obviously not possible in the context of this paper. To test such an hypothesis one should have appropriate data that involves direct access to excitability. As it is provided mostly in the rebuttal letter and the manuscript, previous literature already establishes that link, which is not the result of this manuscript. A more reasonable hypothesis, which the authors likely mean, might be that both $A\beta$ and tau PET SUVR values, and the resting-state fMRI can be a proxy for neural excitability. However, the authors present the excitability values inferred by the model as if they were ground truth empirical values, which is misleading for the readers. To provide evidence for this claim the authors must show that the $A\beta$ and tau PET SUVRs better explain the variations in the observed fMRI data (i.e. the model fit) compared to alternative maps and measures. Instead they compare different models (i.e. microglial activation) using model generated neuronal excitabilities, but not what they explain in the empirical data. As it was also mentioned by the other reviewers, this type of reasoning is circular and does not provide any evidence for the hypothesis. It is clear in Supplementary Figure 11 that the

fALFF values are noisy and evenly distributed across groups, but the almost binary contrast in A β SUVRs is directly manifested in the model generated neuronal excitabilities, which provokes doubt. There are straight forward methods in the literature to resolve this doubt. It is not known what is the correlation between raw SUVRs (A β and tau) and fALFFs. One should expect a high similarity between raw SUVRs and fALFF, and the large-scale model fit should provide significantly higher similarity, meaning that the large-scale model is generating some features of the observed signals due to the dynamics beyond provided input. In other words does the large scale model provide any information or prediction other than what is already present in the empirical data? (also see Major Comment 5 for concerns regarding potential artificial origins of the results.) If not, it is more appropriate and more straight forward to present only the empirical results and describe the relationship between the parameter θ and mean excitatory activity/theta power/alpha power in a toy WC model as a possible mechanism.

Major Comment 2: In the new version of the manuscript, the authors included information regarding the model fitting, however the information provided is not sufficient and it is contradictory. It is also not clear why model fitting results are not in the main text. Model fitting values are essential to the main claims of the paper.

Major Comment 2a: In rebuttal letter and manuscript authors mention that the similarity between model and empirical fALFF was about $r = 0.44$. In supplementary figure 2, they provide a similarity value of $r = 0.33$. Why is the discrepancy between these values?

Major Comment 2b: Given the number of points in the supplementary figure 2, it seems to show the data from each region and subject separately, which unnecessarily inflates the degrees of freedom and make it difficult to interpret. In addition, usually the individual subject fit is an order of magnitude lower than the average fit, therefore the actual model fit is expected to be lower than this value. The authors are expected to provide the average (and/or the distribution) of the best fit values of individual subjects given that the model fitting is done at individual subject level.

Major Comment 2c: They say in the text that the relationship is linear, but in supplementary figure 2 they state that the data is shown in logarithmic scale, so how the relationship can be linear?

Major Comment 2d: Actually, they supposed to have a proper null model to compare with A β and tau, since any other arbitrary map could explain the variability in fALFF values (i.e. is there any other model, such as NMDA or other neurotransmitter, that explains the fALFF better and known to correlate with neural excitability, although it does not generate the contrast across clinical groups?). Microglial activation could only partly address this issue.

Major Comment 3: Related to the comments 1 and 2, the model fit to microglial activation should also be reported and be compared to the proposed model. From supplementary figure 7, it is understood that the authors used A β , tau and interaction term as well as microglial activation in this extended model. If their hypothesis was fulfilled, one would expect no differences in the optimal dynamical regime and the neural excitability values, but the θ values for microglial activation would approach to zero. Indeed, the results in supplementary figure 7 suggests that microglial activation might be explaining fALFF values as much as A β , tau and the interaction term (i.e. might have better model fit). Therefore, authors should show that the model fit for the reduced model (without microglial activation) is indeed better than the extended model (with microglial activation). If it fails, this means that the results merely show under-fitting to A β and tau due to the restrictions of the model on the given parameter space (also see Comment 5).

Major Comment 4: The comment 1 in the initial revision was meant to understand whether almost

binary differences in A β and tau SUVRs values (as in Supplementary Figure 11) alone can explain the results. It would be useful to see the relationship between average SUVRs and average excitability to rule out circularity of the argument. For example the raw A β SUVR values in Supplementary Figure 11 might be perfectly correlated with those in Figure 3. Moreover, as in table 1, a linear model of average A β and tau SUVRs might simply explain the effects presented in Figure 4. If this is the case, it would raise questions about the utility of a whole-brain model if these results can be explained by a simpler linear model.

Major Comment 5: Supplementary figure 10 was quite informative to understand the dynamics of the model. I thank the reviewer who asked this information and the authors for providing it. I suggest plotting also the average model fit (along with the standard deviation), because it is quite important to understand how the optimal model is selected, also because the range for θ seems quite narrow (2.75 – 2.85), raising further concerns. Since A β SUVR and tau values vary across regions, the model fitting algorithm may try to adjust excitability to keep regions with extreme values within this restricted range. Note that the initial values of the difference between minimum and maximum SUVR is larger for A β + individuals. Therefore, the hypo- and hyper-excitability states might appear artificially in A β + individuals because the algorithm may be unable to find a solution without at least one region being outside the restricted range. Authors could diagnose this issue by simply normalizing the SUVRs between 0 and 1 per subject instead of the absolute maximum value to each imaging modality (so removing the SUVR contrasts in Supplementary Figure 11). The results should not change apart from relative optimal θ s, if they are driven by the spatial patterns of SUVRs and the optimization procedure is robust.

Minor Comment 1: The optimal model parameters were not reported anywhere in the manuscript. One of the advantages of using such a model is to study the contribution of θ -0, θ -A β , θ -tau and θ -A β .tau terms. Knowing the optimal parameters is also important to diagnose any potential issues as raised in Major Comment 5 and Major Comment 3.

Minor Comment 2: Supplementary figure 1 does not explain anything. It shows 2D points overlaid on a 3D axis. It is not possible to extract any information from this figure.

Minor Comment 3: In figure 1b, there is an illustration of real BOLD and simulated BOLD, which is confusing because the model fit was not done with BOLD signal itself, but with fALFF values.

Reviewer #1 (Remarks to the Author):

The authors have thoroughly revised the manuscript and addressed all major concerns. I have no further suggestions for improvement.

Response:

Thank you for your suggestions to improve the manuscript. We deeply appreciate the helpful review.

Reviewer #2 (Remarks to the Author):

Thank you for your thoughtful and thorough replies

I am overall very happy with your paper, congratulations!

Response:

We deeply appreciate your comments and suggestions and are pleased with the improvement that they are bringing to the manuscript.

Major comment 1: I have only one final (and very short) request to the authors:

The bounds of the fitted parameters in the objective function are [-0.05, 0.05]. It is, of course, possible that new global minima for these parameters exist in the neighborhood, say, [-0.07, 0.07]. The new optima may change sign even.

This has to be brought up in the discussion. The reason for choosing these intervals and the limitation of setting these intervals should be remarked upon in the Discussion. I am aware that this is mentioned in the Methods section, but this is important enough to be included.

Response:

Thank you. This is an excellent observation. We agree that it deserves further clarification and have made corresponding changes around the lines (356-362 of the current manuscript) where the assumptions on the parameters were discussed:

<<To generate plausible signals and compare results across participants and disease states after the unsupervised estimation of the A β and tau neuronal activity effects, the individual calculations were run under equal experimental conditions (model parameters that are not relevant to the neuronal populations' excitabilities, anatomical connectivities, initial conditions of the dynamical system). These constraints yielded rather stringent parameter optimization bounds ([-0.05,0.05] for each pathophysiological influence and their combined effect). It is possible that new optima existed outside of these intervals in different conditions.>>

Here are other minor comments that you can consider:

Minor comment 1: I have to admit I am still taken aback by calling the method "AD activity decoder". I do not see how this method is conceptually different than the fitting done in, for example, Ranasinghe et al. 2022. The authors argue that because (and please correct me if I am wrong) θ_0 in the objective function is derived from data the paper herein finds a more causal link than compared to just fitting without θ_0 .

What the authors herein have done is, of course, an improvement to Ranasinghe, but giving the method a name such as "activity decoder" is unnecessary and I think the authors should omit it from the manuscript. What the authors have is a fantastic data fitting of Alzheimer's patients, but it is not a novel methodology in and of itself and does not require a name.

I hope the authors understand that I only have good intentions here: I sincerely think the paper would be easier to read and understand if the "activity decoder" term is left unused.

Response:

Thank you. We agree with the reviewer and apologize for not making appropriate changes before. In this revised manuscript version, the term "activity decoder" has been replaced with "neural/neuronal activity models", "neuropathological influence model", "whole-brain estimations" given the specific context in which it appeared.

Minor comment 2: The authors say

"However, generally these approaches do a parameter search -evaluating a cost function at specific points- rather than optimization in a continuous interval"

and state that such grid searches are not robust as optima can hide between evaluations (resolution too low). This is not correct when done properly. Grid searches are better and much more robust in general than global optimization methods, it is just that they are too computationally expensive for $n > 2$, typically. The problem with global optimization is you have no way to verify whether you're at a global optima. With grid searches, you can verify this for an interval of parameter values, which is why it is preferred for low-dimensional problems.

Response:

Thank you. Yes, it is true that it is straight-forward to confirm the existence of an optimum by observing the behavior of the cost function in a certain grid range. This works best in one dimension, as expressed by the reviewer. In our case, we're estimating three parameters, thus the

utilized optimization strategy. We recognize the limitations of the method as in Major comment 1 above. We apologize if we didn't express well in the previous response.

Minor comment 3: I did not mean for you to cite a poster. I was confused, as I had seen a seminar about the work and mistakenly thought it was published. I am sorry about this, as I think it is unfair (and must seem strange) to suggest adding a poster to the literature list.

The paper I was thinking about has been published now if you are interested in reading (I'm not asking you to cite it, just FYI)

<https://link.springer.com/article/10.1186/s13195-023-01349-9>

Also, one paper and one preprint on fitting whole-brain models to Alzheimer's data for your consideration

<https://www.frontiersin.org/articles/10.3389/fnagi.2023.1204134/full>

<https://www.medrxiv.org/content/10.1101/2023.01.11.23284438v1>

Response:

Thank you for bringing our attention to these papers. We are pleased to see how much the field has advanced since we finalized drafting this article some time ago. We hope all these efforts, taken together, can help further our understanding of AD and inform the rest of the research and clinical community to develop disease-modifying interventions.

Reviewer #3 (Remarks to the Author):

I thank Sanchez-Rodriguez and colleagues for addressing the concerns and substantially improving the manuscript after the revision. The previous version of the manuscript lacked details about the model and it was not clear how the model offered any evidence to back the conclusions up by the authors. It is clear that both the rebuttal and the newer version of the manuscript provide sufficient details of the model and they added new analyses and figures to describe the model. They have also added more rigor into the statistical analyses in the new version. The study relies on very nice data and it seems like there have been rigorous work done. Nevertheless, the rebuttal and the new version do not answer the most critical concerns, especially on establishing the causal relationships and evidence to support the validity of the model. A substantial part of the rebuttal refers to the literature instead of addressing these issues in the manuscript. Indeed, as more information is provided regarding the model, more concerns have been raised in the revised version of the manuscript.

Response:

Thank you for the revision. Below we reply to the reviewers' additional comments and suggestions, also making a substantial effort to improve and clarify these aspects in the manuscript.

Major Comment 1: The main hypothesis of the paper and how it has been shown is still unclear given the results. If the main hypothesis is to the causal relationship between A β , tau and excitability, then it obviously not possible in the context of this paper. To test such an hypothesis one should have appropriate data that involves direct access to excitability. As it is provided mostly in the rebuttal letter and the manuscript, previous literature already establishes that link, which is not the result of this manuscript. A more reasonable hypothesis, which the authors likely mean, might be that both A β and tau PET SUVR values, and the resting-state fMRI can be a proxy for neural excitability. However, the authors present the excitability values inferred by the model as if they were ground truth empirical values, which is misleading for the readers. To provide evidence for this claim the authors must show that the A β and tau PET SUVRs better explain the variations in the observed fMRI data (i.e. the model fit) compared to alternative maps and measures. Instead they compare different models (i.e. microglial activation) using model generated neuronal excitabilities, but not what they explain in the empirical data. As it was also mentioned by the other reviewers, this type of reasoning is circular and does not provide any evidence for the hypothesis. It is clear in Supplementary Figure 11 that the fALFF values are noisy and evenly distributed across groups, but the almost binary contrast in A β SUVRs is directly manifested in the model generated neuronal excitabilities, which provokes doubt. There are straight forward methods in the literature to resolve this doubt. It is not known what is the correlation between raw SUVRs (A β and tau) and fALFFs. One should expect a high similarity between raw SUVRs and fALFF, and the large-scale model fit should provide significantly higher similarity, meaning that the large-scale model is generating some features of the observed signals due to the dynamics beyond provided input. In other words does the large scale model provide any information or prediction other than what is already present in the empirical data? (also see Major Comment 5 for concerns regarding potential artificial origins of the results.) If not, it is more appropriate and more straight forward to present only the empirical results and describe the relationship between the parameter θ and mean excitatory activity/theta power/alpha power in a toy WC model as a possible mechanism.

Response:

Thank you. To better reply to this comment, we will separate it in main points:

- We would like to emphasize that previous literature did not establish a causal relationship between A β , tau and excitability in-vivo in human subjects with AD. As we write, <<Although proxy measurements⁵⁴, post-mortem studies¹⁴ and animal models⁴ have suggested neuronal hyperactivity mechanisms in AD, no direct quantification of in-vivo neuronal excitability existed thus far in humans>>. We would be happy to add and discuss any additional reference if it existed.

- Please, note that we do not present the excitability values inferred by the model as if they were ground truth empirical values. Across all the manuscript, we utilized the terms <<reconstructed>>, <<estimated>>, <<model-obtained>>, <<inferred>>, <<identified>>, etc. accompanying the words excitability/excitabilities. In other cases, it is implied in the context that these values are estimates from our data. For example, through lines 206-234 of the current version, variations of the word <<excitability>> appear 9 times without any of the above-mentioned clarifying terms. However, the title of the sub-section clearly reads <<The ESTIMATED neuronal excitabilities...>> and the accompanying figure caption says <<INFERRED neuronal excitability values...>> As it has been also extensively described in the Discussion (see for example the paragraphs of lines 352-381, 407-429 and 430-453 of the current version), these estimates are subject to model assumptions, computational limitations, etc.
In this revision, we have identified two instances where the terms <<reconstructed>>, <<estimated>>, <<model-obtained>>, <<inferred>>, <<identified>> did not preclude <<excitability/excitabilities>> and have corrected it for increased clarity (lines 137 and 271). We are confident that the readers will understand that the inferred excitability values are not the ground truth.
- We have modified the previous lines that explain what causality means in this context to, once again, explicitly explain the purpose of our work (now lines 307-313): <<Instead of assessing associations, we intended to characterize the direct role that A β and tau have in the generation of pathological neuronal activity in AD. In neural mass models, causal effects are straight-forwardly measured by perturbing its relevant biophysical parameters and observing the dynamical changes that occur to the neuronal signals generated under the perturbation^{1,23,29,50}. The subject-specific influences by A β and tau (imaged through in-vivo PET) on neuronal firing were computationally identified from fMRI indicators in this work.>>
- Across sections of the manuscript, we have characterized/explained the predominant effect of A β and how the estimations were performed individually with only the participant's data feeding the model –thus without circularity– but also in a framework that allowed the comparison of results between participants and disease stages (to better understand the disease mechanisms). Although the results were clear from the initial version, we could have done a better job organizing the discussion of these important aspects. This is why, responding to concerns by Referees 1 and 2 in the previous revision -that are invoked in the above question-, we re-wrote parts of the *Discussion* to come up with the paragraph of lines 352-381 (of the current manuscript). We appreciate the feedback.
<< The above interpretation of the results in this article relies on the assumptions of the pathophysiological influence model (see equation 1) and previous evidence of A β and tau's effects in the AD brain. In our individual dynamical models, the subject's A β and tau accumulations are the sole source of information and permitted influences on the generated neuronal activities. To generate plausible signals and compare results across

participants and disease states after the unsupervised estimation of the A β and tau neuronal activity effects, the individual calculations were run under equal experimental conditions (model parameters that are not relevant to the neuronal populations' excitabilities, anatomical connectivities, initial conditions of the dynamical system) . These constraints yielded rather stringent parameter optimization bounds ([-0.05,0.05] for each pathophysiological influence and their combined effect). It is possible that new optima existed outside of these intervals in different conditions. A β and tau's toxic accumulations are believed to play key roles in the processes leading to neuronal degeneration and loss¹⁻³. Their respective progression patterns (measured in-vivo by PET uptake) are different, however, with A β plaques generalizing to many cortical areas early in the disease, while NFT spreading increases rapidly in temporal and parietal regions only⁵⁷. Previous research also suggested that A β -induced hyperexcitability precluded tau accumulation^{58,59}. AD participants, naturally, present higher levels than their counterparts who are A β - and tau-negative and may not be diagnosed as such⁶⁰. Based on these facts, one may explain why A β 's separate contribution and A β and tau's synergistic interaction –and not tau alone, with less whole-brain involvement than A β – were the most significant factors influencing aberrant neuronal activity and cognitive symptomatology in our cohort (Table 1). Likewise, one would have expected a certain separation of neuronal excitability according to A β and tau statuses (Figure 3, Supplementary Figure 3). Existing computational models that assume parameter perturbations by relevant factors as serotonin receptor maps in neuropsychiatric disorders yield equivalent results²⁴. Our objective was to detect the trends in such possible separation conditioned by the underlying A β and tau AD data and, by doing so, to better characterize in-vivo human disease mechanisms. Unequivocal evidence across analyses indicates the existence of a significant neuronal excitability (and functional) change in AD that relates to the disease's physical progression: the more pathology there is, the more the neuronal populations fire. >>

- Regarding the relationship between SUVRs (A β and tau) and fALFFs and the utilization of data-informed individual whole brain models, we shall report that the Pearson correlation coefficient for the A β SUVR and fALFFs (all regions and subjects) was -0.06 and between tau SUVR and fALFFs (all regions and subjects), -0.01 (not statistically significant), which supports why the biophysical analyses and the multivariate transformations in this work (see Methods) were performed instead of the above-mentioned correlation analyses.

Major Comment 2: In the new version of the manuscript, the authors included information regarding the model fitting, however the information provided is not sufficient and it is contradictory. It is also not clear why model fitting results are not in the main text. Model fitting values are essential to the main claims of the paper.

Response:

Thank you. We would like to clarify that the model fitting values were already provided in the main text (please see lines 143-144 of this version). We apologize for not previously stating the standard deviation of the optimization costs (correlation between observed fALFF markers and the best-fit in-silico analogous quantities) which has been corrected in this version (line 144). The relationship between the real fMRI indicators and the best reconstructed ones from the parameter optimization process was shown in two complementary -but not opposing- ways: 1) through the average of all optimization costs across subjects (in the main text) and 2) through the analysis of the relationship between all regional values for all participants (Supplementary Figure 2 and discussed in the main text). More details are provided in the response to the comments 2a-d below.

Major Comment 2a: In rebuttal letter and manuscript authors mention that the similarity between model and empirical fALFF was about $r = 0.44$. In supplementary figure 2, they provide a similarity value of $r = 0.33$. Why is the discrepancy between these values?

Response:

Please, note that in lines 140-144 of the previous version we wrote: <<The AVERAGE correlation between observed fALFF markers and the best-fit in-silico analogous quantities was 0.44 ACROSS PARTICIPANTS. For ALL REGIONS AND SUBJECTS (8712 data points), real and in-silico neuronal activity indicators generated through our pathological influence model follow a linear relationship (Supplementary Fig. 2)>> (which is the 0.33 value). These are two complimentary ways to inform about the results of the individual model fitting: the first one focuses more on the participant-wise behavior and simply consists of averaging all the obtained optimization costs (1-the optimization cost, i.e., the lower the optimization cost, the more similar the model-fitted and real fALFF are, see Methods, Parameter Estimation). The second approach focuses on the regional behavior across all subjects. This justifies the investigation of regional profiles as in the results of Figure 4 f-g, which validate the results of the model through AD biomarkers that were not considered for the parameter estimations.

Major Comment 2b: Given the number of points in the supplementary figure 2, it seems to show the data from each region and subject separately, which unnecessarily inflates the degrees of freedom and make it difficult to interpret. In addition, usually the individual subject fit is an order of magnitude lower than the average fit, therefore the actual model fit is expected to be lower than this value. The authors are expected to provide the average (and/or the distribution) of the best fit values of individual subjects given that the model fitting is done at individual subject level.

Response:

Thank you. As stated above, the average value was provided in the previous version, which is accompanied by the standard deviation of the distribution of values in this revision (lines 143-144 of the current manuscript). We have explained the rationale for Supplementary Figure 2 in the response to Major Comment 2a above. We have not found evidence for the individual subject fit being an order of magnitude lower than the average fit. If we understand correctly, the “average fit” value will be calculated as the sum of all individual “subject fit” values divided by the number of subjects, which necessarily entails that some subjects will have higher values than the average, and others, less (unless all the subjects have the same value). Additionally, we do not have reasons to expect differences of an order of magnitude (i.e., 10 times lower or higher).

Major Comment 2c: They say in the text that the relationship is linear, but in supplementary figure 2 they state that the data is shown in logarithmic scale, so how the relationship can be linear?

Response:

Thank you. The reported values are the result of a linear Pearson correlation analysis, as explained in the Supplementary Fig. 2 caption and in lines 144-146 of the current manuscript. As it was also mentioned in the caption of Supplementary Figure 2 in the previous version, we transformed the data into logarithmic scale for better visualization of the pattern. Logarithmic functions are one-to-one. That means that the function maps distinct elements of its domain to distinct elements; that is, $x_1 \neq x_2$ implies $f(x_1) \neq f(x_2)$. We have removed the logarithmic scale in this revision.

Major Comment 2d: Actually, they supposed to have a proper null model to compare with $A\beta$ and tau, since any other arbitrary map could explain the variability in fALFF values (i.e. is there any other model, such as NMDA or other neurotransmitter, that explains the fALFF better and known to correlate with neural excitability, although it does not generate the contrast across clinical groups?). Microglial activation could only partly address this issue.

Response:

Thank you for this interesting suggestion. Yes, it would have been additionally informative to utilize other distributions in the analyses. However, notice that for our cohort or any other publicly available AD cohort, NMDA PET data is not available (nor any other neurotransmitter imaging) for the participants. Accordingly, our data include AD-relevant PET/MRI modalities (e.g., $A\beta$ -tau PET, and structural MRI), but not neurotransmitters PET. Although freely-available templates exist for many proteins, genes and neurotransmitters distributions, they constitute only averages over very small populations. Including such maps would only make sense if the individual distributions were known for exactly the same subjects, so to study the effects of these factors and their produced individual variability.

Regarding the microglial activation data, we have explained in the manuscript (lines 382-406 of the present version) how this sort of PET imaging requires further improvement. Specifically, it does not distinguish detrimental from neuroprotective types, a fact that seems to deeply compromise its interpretation. We provide further details in the response to Major comment 3 below, but it is worth noting that the empirical confirmation of the models in terms of AD biomarkers that were not inputted into the model (Figure 4), relationships with the subjects' cognitive scores (Table 1) or reconstructed electrophysiological alterations (Figure 2), do not stand when microglial activation effects were assumed to influence neuronal activity too. The microglial activation data does not predict fALFF per se (0.04 Pearson correlation coefficient for all regions and subjects, no statistically significant relationship), as did not A β or tau SUVRs alone (see response to Major comment 1 above).

It is fundamental to observe the nonlinear dynamical transformations that produce the in-silico resting-state fMRI signals from which the fitting to the real fALFF indicators is done (section Methods, Personalized integrative AD neuronal activity model). It would have been very interesting to test if other data distributions could directly predict fALFF, why also providing mechanistic insights into variables that are hidden to non-invasive human in-vivo neuroimaging as neuronal excitability. We explained that straight-forward modifications could be made to our framework provided that the data is available (for example, lines 421-429 of the present manuscript) and we expect that new disease-relevant distributions are tested in the near future.

Major Comment 3: Related to the comments 1 and 2, the model fit to microglial activation should also be reported and be compared to the proposed model. From supplementary figure 7, it is understood that the authors used A β , tau and interaction term as well as microglial activation in this extended model. If their hypothesis was fulfilled, one would expect no differences in the optimal dynamical regime and the neural excitability values, but the θ values for microglial activation would approach to zero. Indeed, the results in supplementary figure 7 suggests that microglial activation might be explaining fALFF values as much as A β , tau and the interaction term (i.e. might have better model fit). Therefore, authors should show that the model fit for the reduced model (without microglial activation) is indeed better than the extended model (with microglial activation). If it fails, this means that the results merely show under-fitting to A β and tau due to the restrictions of the model on the given parameter space (also see Comment 5).

Response:

Thank you. We have not found evidence for underfitting in the main results (i.e., models with A β , tau and A β -tau individual influences on neuronal excitability). The model optimization results predict cognitive decline in the participant cohort (Table 1), clinical states (Figure 3 and Supplementary Figure 3), the increase in the concentration of plasma biomarkers with the disease (Figure 4a-d), grey matter atrophy (Fig 4e-h in brain-wide relationships and specific

regions, particularly within the medial temporal lobe, that show degeneration in AD). These are all clear hallmarks of the disease that were assessed by clinicians and/or obtained through established techniques for each participant (see Methods, Image processing and Plasma biomarkers) and that did not inform our computational models. Additionally, we reconstructed electrophysiological signals with the obtained parameters through the nonlinear dynamical transformations of Methods, Personalized integrative AD neuronal activity model, and they showed the classical sign of AD alterations reported in the literature (Figure 2).

The microglial activation results were included in the manuscript as a further element to underscore the need for developing better PET tracers for neuroinflammation, which we discuss in lines 382-406. The results explain that this particular measure does not present the same empirical differences as A β and tau SUVRs values. While the average correlation between the best-fit reconstructed fALFFs and the real resting state fMRI indicators was slightly higher in the model <> (0.50 ± 0.07 vs 0.44 ± 0.10 correlation in the model without the microglial activation term) this is likely indicative of the noise observed in the microglial activation data and thus overfitting in the models. In Supplementary Figures 7-8 we showed that no significant differences in neuronal excitability existed between the clinical groups in this microglial activation PET-informed model. Supplementary Table 4 showed that cognitive impairment was predicted by the A β -tau interaction term in this model with no significant effect by the microglial activation. In the newly added Supplementary Figure 9 (see below and Supplementary Materials for more details) we also present illustrative results for electrophysiological quantities, plasma biomarkers and grey matter atrophy. In all cases, the relationships observed in the <> influence model did not appear whatsoever or were significantly affected when the microglia-PET information is added, which further supports its noisy/low-quality attribute (as also discussed in publications by Shen et al., 2018 and Nutma et al., 2023 that have been cited in the manuscript).

We shall also not assume that the functional data (fALFF indicators) used to fit our nonlinear, personalized dynamical neuronal activity models can be directly predicted from the input data (as demonstrated in the response to Major Comments 1 and 2d above). Please, see the updated Discussion -lines 382-406, with changes highlighted in orange- for the final writing of the elements presented in this answer.

<< Beyond AD-related protein deposition, our method can also investigate the influence of other critical factors. It has been hypothesized, and to some extent observed^{7,12,13}, that microglial activation (a probable marker for neuroinflammation^{61,62}) affects excitability and neuronal activity in AD. Consequently, we performed a set of complimentary experiments where we recreated the obtained results in a model that also considers deviations to neuronal excitability due to microglial activation –measured with 18kDa Translocator Protein PET. Despite the slightly better fit in terms of resembling the real resting state fMRI indicators (0.50 ± 0.07 vs 0.44 ± 0.10 correlation in the model without the microglial activation term), we did not find substantial neuronal excitability or spectral electrophysiological separation between clinical groups when the microglial activation factor was considered, nor the estimations were confirmed by the participants' plasma and grey matter atrophy markers (Supplementary Figures 7-9). Moreover, the synergistic interaction of Aβ and tau was the factor that better predicted cognitive impairment, with no significant effect by the microglial activation term (Supplementary Table 4). We attribute these results to model overfitting and/or technical limitations associated with the acquisition of microglial activation. Unlike the Aβ and tau PET SUVRs data, which showed extended statistically significant differences across all brain regions for CU and AD participants (ANCOVA post-hoc t-tests with age and sex as covariates, $p < 0.05$), microglial activation images exhibited differences in only 30 regions (i.e., approx. 45%; Supplementary Table 5). Microglial

activation is thought to have a neuroprotective character (M2-phenotype) at early disease stages^{12,13}. On the other hand, excessive activation of microglia seemingly becomes detrimental in clinical AD (M1-phenotype) by releasing pro-inflammatory cytokines that may exacerbate AD progression^{12,13,62}. Nevertheless, modern neuroinflammation PET tracers are not specific to these two different phenotypes as no consistent targets have been discovered¹³. Thus, our extended results albeit being relatively uninformative in terms of AD-affectations to neuronal excitability, capture intrinsic microglial activation PET mapping insufficiencies⁶³. >>

Major Comment 4: The comment 1 in the initial revision was meant to understand whether almost binary differences in A β and tau SUVRs values (as in Supplementary Figure 11) alone can explain the results. It would be useful to see the relationship between average SUVRs and average excitability to rule out circularity of the argument. For example the raw A β SUVR values in Supplementary Figure 11 might be perfectly correlated with those in Figure 3. Moreover, as in table 1, a linear model of average A β and tau SUVRs might simply explain the effects presented in Figure 4. If this is the case, it would raise questions about the utility of a whole-brain model if these results can be explained by a simpler linear model.

Response:

Thank you. We have explained the biophysical meaning of the magnitude neuronal excitability (see section Methods, Personalized integrative AD neuronal activity model, lines 571-590 in the current manuscript). This quantity has a dynamical/functional definition (firing pulses per unit of time) and is non-observable by current (non-invasive) imaging techniques. Personalized whole-brain models were necessary to estimate its values and perform consequent analyses as those of Figure 4. Only after the individual brain neuronal excitability maps have been identified, we can look at relationships with this quantity, thus reemphasizing the need for the whole-brain model.

In lines 352-381 we acknowledged the effect that the amyloid-beta and tau PET SUVR distributions have on neuronal excitability, which is a key scientific question we sought to respond (this paragraph was written into the Discussion in the last revision by expanding on information that existed over other paragraphs). In this specific study, we are interested in added excitability influences/contributions by amyloid-beta and tau given their key role in AD, thus the model considers (and infer) the effect of these neuropathological factors over typical/baseline neuronal excitability. Motivated by our scientific questions, we focus on quantifying the added contributions (over normal physiological baseline) of amyloid-beta and tau to neuronal activity. This means that the model in fact considers the relationship between intrinsic brain oscillations and neuronal excitability given amyloid and tau, while the estimated parameters for amyloid-tau are optimized in a 'competitive' way: they are only meaningful when amyloid and tau are contributing to explain data variance beyond the normal physiological state.

Figure 4 demonstrates that the estimated neuronal excitabilities predict Alzheimer's biomarkers (plasma and tissue atrophy) that did not feed the individual whole-brain dynamical models. In the manuscript, we reference 7 whole-brain Alzheimer's models that existed at the moment of writing or that were kindly suggested by Referee 2. Of those, only two (Ranasinghe et al and Kobeleva et al) perform some form of parameter identification. The rest are simulation studies. While most of these articles (4 simulation studies) reproduce observed electrophysiological (E/MEG) features of AD (which we do in Figure 2 with data-inferred quantities) and one of them (Zimmermann et al) explores correlation with cognitive performance (which we do in Table 1 through data-inferred quantities), research had not produced, to this moment, the kind of biological validation that we observe in Figure 4 for the estimations in our personalized models.

Major Comment 5: Supplementary figure 10 was quite informative to understand the dynamics of the model. I thank the reviewer who asked this information and the authors for providing it. I suggest plotting also the average model fit (along with the standard deviation), because it is quite important to understand how the optimal model is selected, also because the range for θ seems quite narrow (2.75 – 2.85), raising further concerns. Since A β SUVR and tau values vary across regions, the model fitting algorithm may try to adjust excitability to keep regions with extreme values within this restricted range. Note that the initial values of the difference between minimum and maximum SUVR is larger for A β + individuals. Therefore, the hypo- and hyper-excitable states might appear artificially in A β + individuals because the algorithm may be unable to find a solution without at least one region being outside the restricted range. Authors could diagnose this issue by simply normalizing the SUVRs between 0 and 1 per subject instead of the absolute maximum value to each imaging modality (so removing the SUVR contrasts in Supplementary Figure 11). The results should not change apart from relative optimal θ s, if they are driven by the spatial patterns of SUVRs and the optimization procedure is robust.

Response:

Thank you. Importantly, all the subjects' SUVR values are normalized to the same interval in order to be able to compare results across subjects and disease states. This is common practice in studies utilizing neuroimaging data, particularly those focusing on understanding disease mechanisms. Notice that a normalization at the individual level would destroy the physiological meaning of the images and make difficult the across-subjects comparison/interpretation of the results. We have further described the choosing of the normalization strategy in lines 588-590 of the current manuscript: <<each PET modality's SUVRs normalized to the [0,1] interval (by dividing by the maximum value across subjects and regions⁹², Supplementary Figure 12), to preserve the dynamical properties of the generated signals and compare values across subjects and conditions>>.

Additionally, the average value and the standard deviation for the optimization results across subjects are given in lines 143-144 (i.e., in the first section of the Results). Supplementary Figure

10 shows the conditions in which the parameter estimation was performed and was not meant to refer to any results. We have made corresponding changes in the Discussion (lines 356-362 of the current manuscript) to better explain assumptions on the parameters and optimization intervals.

<<To generate plausible signals and compare results across participants and disease states after the unsupervised estimation of the $A\beta$ and tau neuronal activity effects, the individual calculations were run under equal experimental conditions (model parameters that are not relevant to the neuronal populations' excitabilities, anatomical connectivities, initial conditions of the dynamical system). These constraints yielded rather stringent parameter optimization bounds ([-0.05,0.05] for each pathophysiological influence and their combined effect). It is possible that new optima existed outside of these intervals in different conditions.>>

Minor Comment 1: The optimal model parameters were not reported anywhere in the manuscript. One of the advantages of using such a model is to study the contribution of θ -0, θ - $A\beta$, θ -tau and θ - $A\beta$.tau terms. Knowing the optimal parameters is also important to diagnose any potential issues as raised in Major Comment 5 and Major Comment 3.

Response:

Thank you. Supplementary Figure 1 shows the model parameters (see next question aka Minor Comment 2). This figure existed from the initial submission and was referred to in the first subsection of the results. We discuss the parameter values over lines 136-146 of the current manuscript.

In this work, the individual factor influences were best studied in terms of their effects on cognition (Table 1), where we observe that the $A\beta$ and $A\beta$ -tau influences were significant predictors of MMSE and MoCA scores in the cohort ($p < 0.05$). The coefficients of these terms in the fitted linear models were positive in all cases. This means that negative influences of the pathophysiological factors –yielding hyperexcitability– are generally linked to low cognitive scores. This finding also constitutes evidence supporting the discovered increased neuronal excitability with advanced disease stages. In consequent research that is under consideration elsewhere we further delve into the factorial influences and identify the molecular signatures of their specific alterations. The specific parameter values for each subject would be of tremendous importance to design effective therapeutic interventions: for example, a subject with a high tau parameter perhaps may receive a clinical recommendation of a therapy that controls this effect. While the information is available, we decided to focus on overall mechanistic insights here. We explain the reasoning, in general terms, in the last paragraph of the Discussion.

Minor Comment 2: Supplementary figure 1 does not explain anything. It shows 2D points overlaid on a 3D axis. It is not possible to extract any information from this figure.

Response:

Thank you. Supplementary Figure 1 is a three-dimensional plot. On a piece of paper, points in the space are seen in two dimensions with perspective. The labels and ticks indicate what the axes are representing. To help improve the perspective, we have added a grid to the plot in this revision.

Minor Comment 3: In figure 1b, there is an illustration of real BOLD and simulated BOLD, which is confusing because the model fit not done with BOLD signal itself, but with fALFF values.

Response:

Thank you. The text presenting Fig. 1 explains that it was fALFF values what was utilized for model optimization (lines 123-126). This part was modified in the last version to further justify the utilization of fALFF indicators. Additionally, we provide all the fALFF values in Supplementary Figure 2 which is referred to after Fig. 1. It is important to provide the generated neural mass signal compared to a BOLD signal somewhere in the manuscript to show that the integration of the Wilson-Cowan and hemodynamic/metabolic model was successfully done.

Reviewers' comments:

Reviewer #3 (Remarks to the Author):

The authors addressed many issues in the revised manuscript by additional information and clarifying the discussions. However, the authors still do not resolve the major concern that have been raised from the first review. In the rebuttal letter the authors report that there is no correlation between tau, A β , microglial activation SUVRs and fALFF values. First, this is a very critical information to interpret the results that must be in the main text. Second, this also reaffirms the suspicion that the regional variations of these SUVR maps might not be contributing to the model fit at all, which is against a key claim of the manuscript. Moreover, in the manuscript it is reported that the extended model (with microglial activation) show no difference in excitability values. As the extended model also includes tau and A β , although it is likely that adding extra information cause some overfitting, it should not remove the effect that is already observed. All these directs to possible methodological issues in the modeling framework, which the authors are expected to rule out. As a reviewer it is not possible to figure out what that might be given the information in the manuscript. I suggested in the earlier review that the authors could normalize SUVR values by subject to avoid artifactual results due to the restricted parameter space. I disagree authors' argument in the rebuttal letter that this would change the physiological meaning. This argument might hold for the statistical analysis but it should not change the physiological meaning in the model. Indeed, if the model fitting is robust at individual subject level and tau/A β maps are responsible for the results, the algorithm should find a similar optimum fit with only adjusted θ -values to compensate for the normalization. Alternatively the authors could shuffle the SUVRs across regions for each subject and refit the model. If these random maps leads to similar excitability differences, this would show that the inferred excitabilities do not reflect the regional variations in tau and A β SUVRs.

Reviewer #3 (Remarks to the Author):

The authors addressed many issues in the revised manuscript by additional information and clarifying the discussions. However, the authors still do not resolve the major concern that have been raised from the first review. In the rebuttal letter the authors report that there is no correlation between tau, A β , microglial activation SUVRs and fALFF values. First, this is a very critical information to interpret the results that must be in the main text. Second, this also reaffirms the suspicion that the regional variations of these SUVR maps might not be contributing to the model fit at all, which is against a key claim of the manuscript. Moreover, in the manuscript it is reported that the extended model (with microglial activation) show no difference in excitability values. As the extended model also includes tau and A β , although it is likely that adding extra information cause some overfitting, it should not remove the effect that is already observed. All these directs to possible methodological issues in the modeling framework, which the authors are expected to rule out. As a reviewer it is not possible to figure out what that might be given the information in the manuscript. I suggested in the earlier review that the authors could normalize SUVR values by subject to avoid artifactual results due to the restricted parameter space. I disagree authors' argument in the rebuttal letter that this would change the physiological meaning. This argument might hold for the statistical analysis but it should not change the physiological meaning in the model. Indeed, if the model fitting is robust at individual subject level and tau/A β maps are responsible for the results, the algorithm should find a similar optimum fit with only adjusted θ -values to compensate for the normalization. Alternatively the authors could shuffle the SUVRs across regions for each subject and refit the model. If these random maps leads to similar excitability differences, this would show that the inferred excitabilities do not reflect the regional variations in tau and A β SUVRs.

Reply:

We appreciate the opportunity to further discuss and clarify these aspects. To better reply to this comment, we separated it into two main points:

1) Statistically and mathematically, the correct way of evaluating if adding specific data contributes or not to improving a given model is via the field of 'statistical model comparison'. Essentially, the basic model is evaluated without and with the additional data (e.g., without and with PET values), and specific statistics are subsequently evaluated to compare the explanation of response data (e.g., fMRI indicators) under both conditions and adjusting for number of parameters.

In our study, to test if adding A β and tau SUVR distributions improved or not the explanation of the neurofunctional data, we conducted subject-specific F-tests and summarized them. As the reviewer may know, the F-tests provide a quantitative (not subjective) answer to determine the better model fitting to a given data accounting for the increase in number of parameters,

particularly when an extended model (i.e. with PET) is nested within a restricted model (i.e. without PET).

As shown in Supplementary Figure 7, and discussed in the manuscript (page 13, 2nd paragraph, lines 356-364), the **results clearly indicate significant F-statistics ($p < 0.05$) for most of the participants** (81.8% of participants), while subjects with non-significant F-statistics are predominantly A β - and tau-negative (meaning that PET values were not informative in their case, because they are healthy). Importantly, the obtained F-statistics also tend to increase with disease progression, i.e., **with higher A β and tau accumulations, the stronger their statistical impact on explaining the fMRI signals.**

Altogether, **this statistical analysis further confirms that the regional variations of SUVR maps do contribute to the model fit of fMRI**, capturing the pathophysiological influence by misfolded proteins in brain activity under AD. These effects were suggested in the literature without previous confirmation in the living human brain.

Supplementary Figure 7 | Statistical contribution of regional A β and tau accumulations to explain the observed fMRI-derived neuronal activity patterns. Participants were sub-grouped according to their clinical diagnosis (CU, MCI, AD) and A β -positivity (a) or Braak staging (b). For each participant, statistical comparison was established through subject-wise F-tests between models with and without A β , tau and A β -tau maps (4 and 1 parameters, respectively). The critical threshold (black dashed lines) corresponds to statistically significant F-statistics ($p < 0.05$), i.e. improvement due to A β , tau and A β -tau influences on neuronal excitability, accounting for the increase in adjustable model parameters.

2) As noted above, the correct way to evaluate the added value of new data on models is via 'statistical model comparison', keeping the models exactly as they are with and without the added data and compare performance while adjusting for additional parameters. Please notice that:

- all used SUVR values are **by definition already standardized and normalized**,
- Standardized uptake value ratio (SUVR) is the **most common quantitative method used to make regional comparisons within a subject as well as between subjects**,
- in the field of neurodegeneration, **SUVR values are always normalized at the population level to keep the disease's information, and they are never modified at the individual level** as suggested. Essentially, individual modifications would negatively remove the information distinguishing between controls and diseased subjects, affecting not only the statistical comparisons across groups but also the biological meaning of the input information on the models (since CU would be equal to MCI and equal to AD, which is wrong).
- subsequently, to avoid incorrect statistical and biological management of our data, in this study we will keep following the field's guidelines (SUVR standardization and normalization at population level) and will avoid individual data modification that would break individual-individual relationships. As indicated, performance indicators are tested via the F-tests or the many other analyses performed (e.g., Fig. 4, Table 1).